# Orbital hybridization of donor and acceptor to enhance the conductivity of mixed-stack complexes

Tomoko Fujino [1] ✉, Ryohei Kameyama[1], Kota Onozuka [1], Kazuki Matsuo[1], Shun Dekura[1], Tatsuya Miyamoto[2], Zijing Guo[2], Hiroshi Okamoto [2], Toshikazu Nakamura [3], Kazuyoshi Yoshimi[1], Shunsuke Kitou [2], Taka-hisa Arima [2,4], Hiroyasu Sato[5], Kaoru Yamamoto [6], Akira Takahashi[7], Hiroshi Sawa [8], Yuiga Nakamura[9] & Hatsumi Mori [1] ✉

Mixed-stack complexes which comprise columns of alternating donors and acceptors are organic conductors with typically poor electrical conductivity because they are either in a neutral or highly ionic state. This indicates that conductive carriers are insufficient or are mainly localized. In this study, mixed-stack complexes that uniquely exist at the neutral−ionic boundary were synthesized by combining donors (bis(3,4-ethylenedichalcogenothiophene)) and acceptors (fluorinated tetracyanoquinodimethanes) with similar energy levels and orbital symmetry between the highest occupied molecular orbital of the donor and the lowest unoccupied molecular orbital of the acceptor. Surprisingly, the orbitals were highly hybridized in the single-crystal complexes, enhancing the room-temperature conductivity ($10^{-4}$−$0.1\,\mathrm{S\,cm^{-1}}$) of mixed-stack complexes. Specifically, the maximum conductivity was the highest reported for single-crystal mixed-stack complexes under ambient pressures. The unique electronic structures at the neutral−ionic boundary exhibited structural perturbations between their electron-itinerant and localized states, causing abrupt temperature-dependent changes in their electrical, optical, dielectric, and magnetic properties.

Organic conductors that are lightweight, flexible, and have excellent molecular designability are widely used as essential materials in modern organic electronic devices. The most industrially successful materials are polymer conductors owing to their high conductivities and ease of synthesis[1]. However, analyzing their atomic-level structures and conduction mechanisms is still challenging because of the inhomogeneity of polymers with different chain lengths. To address this issue, researchers are investigating molecular conductors with precisely defined molecular weights[2–6]. The crystal-based structures help to understand the structure−conductivity relationships and address the conduction mechanism. Charge-transfer complexes[3–6] are typical molecular conductors and are categorized as either

[1]The Institute for Solid State Physics, The University of Tokyo, 5-1-5 Kashiwanoha, Kashiwa, Chiba 277-8581, Japan. [2]Department of Advanced Materials Science, The University of Tokyo, 5-1-5 Kashiwanoha, Kashiwa, Chiba 277-8561, Japan. [3]Institute for Molecular Science, 38 Nishigo-Naka, Myodaiji, Okazaki, Aichi 444-8585, Japan. [4]RIKEN Center for Emergent Matter Science (CEMS), Wako 351-0198, Japan. [5]Rigaku Corporation, 3-9-12 Matsubara, Akishima, Tokyo 196-8666, Japan. [6]Department of Physics, Okayama University of Science, 1-1 Ridaicho, Kita-ku, Okayama 700-0005, Japan. [7]Graduate School of Engineering, Nagoya Institute of Technology, Gokiso-cho, Showa-ku, Nagoya, Aichi 466-8555, Japan. [8]Department of Applied Physics, Nagoya University, Furo-cho, Chikusa-ku, Nagoya 464-8603, Japan. [9]Japan Synchrotron Radiation Research Institute (JASRI), SPring-8, 1-1-1, Kouto, Sayo-cho, Sayo-gun, Hyogo 679-5198, Japan. ✉e-mail: fujino@issp.u-tokyo.ac.jp; hmori@issp.u-tokyo.ac.jp

"segregated-stack complexes," in which donors (**D**) and acceptors (**A**) are separately stacked (**DDDD…/AAAA…**) or "(alternatingly) mixed-stack complexes," in which donors and acceptors are alternately stacked (**DADADADA….**; Fig. 1a). The difference in the stacking forms

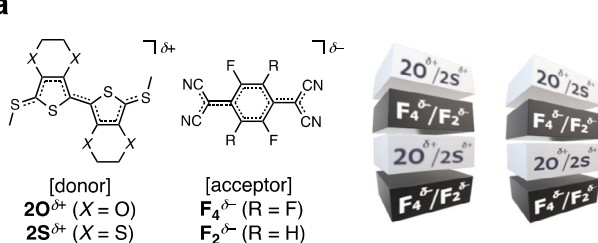

[donor]
**2O**$^{\delta+}$ (X = O)
**2S**$^{\delta+}$ (X = S)

[acceptor]
**F$_4$**$^{\delta-}$ (R = F)
**F$_2$**$^{\delta-}$ (R = H)

**b**

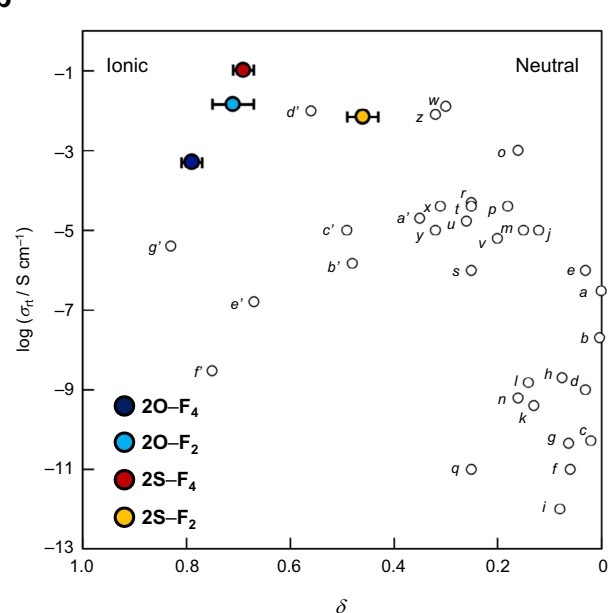

**Fig. 1 | Structures and electrical conductivities of mixed-stack complexes.**
**a** Molecular structures of donors (**2X**; X = O, S) and acceptors (**F$_n$**; n = 4, 2) in the mixed-stack complexes **2X**–**F$_n$**. An illustration of the 1D packing mode of **2X**–**F$_n$** is shown in the right. **b** $\sigma_{rt}$ of 1D mixed-stack complexes as a function of charge-transfer degree from donor to acceptor ($\delta$) determined for single-crystal structures. a: Benzothieno[3,2-b]benzothiophene (BTBT)–**F$_4$**[15]. b: 1,2-Di(2-thienyl)ethylene (DTE)–7,7,8,8-tetracyano-p-quinodimethane (TCNQ)[5]. c: Naphthalene–TCNQ[5]. d: Perylene–TCNQ[5]. e: Dithieno[3,2-b:2′,3′-d]thiophene (DTT)–TCNQ[5]. f: Anthracene–TCNQ[5]. g: trans-Stilbene–TCNQ[5]. h: trans-2,2′,5,5′-Tetramethoxystilbene–TCNQ[5]. i: Pyrene–TCNQ[5]. j: Bis(ethylenedithio)tetrathiafulvalene (BEDT-TTF)–TCNQ[5]. k: 2,3,7,8-Tetramethoxydibenzo[b,e][1,4]thiaselenine (Vn$_2$SSe)–TCNQ[6]. l: 3,3′,5,5′-Tetramethylbenzidine (TMB)–p-chloranil (CA)[16]. m: Tetramethyltetraselenafulvalene (TMTSF)–TCNQ[6]. n: 2,3,7,8-Tetramethoxyselenanthrene (Vn$_2$Se$_2$)–TCNQ[6]. o: 1,6-Diaminopyrene (DAP)–p-bromanil (BA)[22]. p: 4,4′-Dimethyltetrathiafulvalene (DMTTF)–BA[20]. q: Coronene–TCNQ[6]. r: Dibenzotetrathiafulvalene (DB-TTF)–TCNQ[5]. s: Tetrathiafulvalene (TTF)–TCNQ[17]. t: DMTTF–CA[20]. u: 5,10-Dihydro-5-10-dimethylphenazine (DHMP)–TCNQ[6]. v: TTF–CA[18]. w: 2,5-Dichloro-1,4-phenylenediamine (Cl$_2$PD)–2-chloro-5-methyl-7,7,8,8-tetracyano-p-quinodimethane (ClMeTCNQ)[20]. x: 2-Chloro-5-methyl-1,4-benzenediamine (ClMePD)–2,5-diethyl-7,7,8,8-tetracyano-p-quinodimethane (Et$_2$TCNQ)[20]. y: α,α′-Dimethylquaterthiophene (DMQtT)–**F$_4$**[19]. z: ClMePD–2,5-dimethyl-N,N′-dicyanoquino diimine (DMeDCNQI)[20]. a′: Tetrakis(methylthio)tetrathiafulvalene (TTC$_1$-TTF)–2-fluoro-5-methyl-7,7,8,8-tetracyano-p-quinodimethane (FMeTCNQ)[20]. b′: TMB–2,5-dimethyl-7,7,8,8-tetracyano-p-quinodimethane (Me$_2$TCNQ)[16]. c′: TMB–TCNQ[16]. d′: Tetrakis(methyltelluro)tetrathiafulvalene (TTeC$_1$-TTF)–TCNQ[21]. e′: TMB–**F$_2$**[16]. f′: Phenazine–TCNQ[5]. g′: TMB–**F$_4$**[16]. The molecular structures are shown in Supplementary Fig. 1. The $\delta$ values for **2X**–**F$_n$** are shown with error bars (standard deviation).

of complexes composed of the common donors and acceptors dramatically affects the conductivities. Segregated-stack complexes, such as tetrathiafulvalene (TTF) analogs–7,7,8,8-tetracyano-p-quinodimethane (TCNQ) analogs (the molecular structures are shown in Supplementary Fig. 1)[7,8], hexamethylenetetraselenafulvalene–TCNQ[9], and TTF analogs–metal dithiolenes[10], exhibit excellent room-temperature conductivities ($\sigma_{rt}$) of up to $10^3$ S cm$^{-1}$[7,9], although variations on such highly conducting examples are still limited[5,6]. Several hybrid materials between **DD**-stack and **DA**-mixed-stack complexes have shown relatively high $\sigma_{rt}$ ($10^{-2}$ S cm$^{-1}$)[11–14]. In contrast, mixed-stack complexes are typically poor electrical conductors that show low $\sigma_{rt}$ (below $10^{-4}$ S cm$^{-1}$)[15–21] (Fig. 1b), although they are more frequently constructed, possibly owing to the Madelung energy gain between charged donor and acceptor. Generally, mixed-stack complexes are either in a neutral state (charge-transfer degree from donor to acceptor $\delta < 0.5$[4], mostly $0 < \delta < 0.4$[5,6]) with insufficient conductive carriers and weak intermolecular interactions for carrier conduction, or are in a highly ionic state (mostly $\delta > 0.75$)[5,6], wherein the carriers are nearly localized and have low mobility, thereby suppressing conductivity. In the early 2000s, a few neutral complexes showed high conductivity[22,23], possibly due to the structural disorders within the contaminated ionic domains. However, the structural details and mechanisms remained hidden as the CCDC data were unavailable. We hypothesized that the mixed-stack complexes at the neutral–ionic (N–I) boundary ($0.59 < \delta < 0.74$)[5] could have numerous mobile electrons and display excellent electrical conductivity[24]. The hypothesis was supported by the discovery of segregated-stack complexes with exceptional conductivity in the same boundary area[5–10]. Preceding studies[24–26] predicted that charge transfer to form mixed-stack complexes with $\delta$ at the N–I boundary could be realized by engineering appropriate energy gaps between the highest occupied molecular orbital (HOMO) of donors and the lowest unoccupied molecular orbital (LUMO) of acceptors that satisfy the equation $I_D - E_A$ (= $\Delta E_{REDOX}) \approx E_M$, where $I_D$ is the ionization potential of the donor (related to its HOMO of the donor), $E_A$ is the electron affinity of the acceptor (related to its LUMO of the acceptor), and $E_M$ is the electrostatic Madelung energy of the donor–acceptor complex (approximately 0.2 eV; Fig. 2). In addition, a consistent orbital symmetry between the donor HOMO and acceptor LUMO[27] may be necessary to facilitate the hybridization of frontier orbitals and enable efficient charge transfer. Thus far, a few mixed-stack complexes have partly fulfilled these two requirements: 1) similar energy levels between the donor HOMO and the acceptor LUMO, with the appropriate energy gaps of approximately 0.2 eV, and 2) a consistent orbital symmetry between them; for example, tetrakis(methyltelluro)tetrathiafulvalene (TTeC$_1$-TTF)–TCNQ ($\delta = 0.56$; d′ in Fig. 1b)[21] and 3,3′,5,5′-tetramethylbenzidine (TMB)–TCNQ ($\delta = 0.49$; c′ in Fig. 1b)[16,28] close to the N–I boundary exhibited relatively high $\sigma_{rt}$ values of $10^{-2}$ and $10^{-5}$ S cm$^{-1}$, respectively. Under high-pressure conditions (-9 kbar), $\delta$ of TTF–p-chloranil (CA) (v in Fig. 1b) was transiently increased to ≈0.6, exhibiting excellent $\sigma_{rt}$ (7 S cm$^{-1}$)[29], although it exhibits $\delta \approx 0.2$ and $\sigma_{rt} = 10^{-5}$ S cm$^{-1}$ under ambient pressures.

In this study, we designed and synthesized mixed-stack complexes uniquely located at and near the N–I boundary, enhancing the conductivities of mixed-stack complexes. The combination of the oligo(3,4-ethylenedioxythiophene) analog **2O**[30,31] or its oxygen/sulfur-substituted analog **2S** with tetra- or difluorinated TCNQs **F$_n$** fulfills the two requirements for electronic structures (i.e., **2X**–**F$_n$**, X = O, S in Fig. 1a). Surprisingly, the donor HOMO and acceptor LUMO with comparable energy levels and well-matched orbital symmetries were highly hybridized in the complexes, enhancing the $\sigma_{rt}$ values of mixed-stack complexes ($10^{-4}$–0.1 S cm$^{-1}$). The highest $\sigma_{rt}$ value (0.1 S cm$^{-1}$) observed for **2S**–**F$_4$** among the combinations of donors **2X** and acceptors **F$_n$**, which is located at the N–I boundary, is the highest value reported for a structurally defined 1D mixed-stack complex under

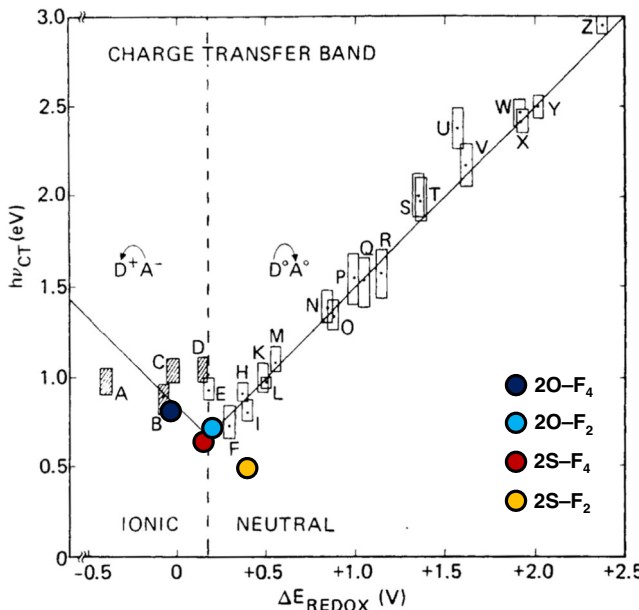

**Fig. 2 | Electronic and optical properties of mixed-stack complexes.** Relationship between $\Delta E_{REDOX}$ and $h\nu_{CT}$ that has two separate linear correlations in the neutral and ionic regions; the two straight lines intersect when $\Delta E_{REDOX} \approx E_M$, resulting in the $h\nu_{CT}$ value reaches to the minimum (approximately 0.6 eV). The vertical dashed line indicates the neutral–ionic boundary (Reprinted and modified from Reference[26], Copyright 1985, from Taylor & Francis). A: *N,N,N',N'*-Tetramethyl-*p*-phenylenediamine (TMPD)–**F$_4$**. B: DHMP–TCNQ. C: TMPD–TCNQ. D: TMPD–CA. E: *N,N,N',N'*-Tetramethyl-1,6-pyrenediamine (TMDAP)–TCNQ. F: TTF–CA. G: TTF–*p*-fluoranil (FA). H: Dibenzene TTF–TCNQ. I: Diethyldimethyltetraselenafulvalene–Et$_2$TCNQ. J: TMDAP–FA. K: TTF–dichlorobenzoquinone. L: Perylene–**F$_4$**. M: Perylene–2,3-dichloro-5,6-dicyano-*p*-benzoquinone. N: Perylene–tetracyanoethylene. O: Perylene–TCNQ. P: TTF–dinitrobenzene. Q: Perylene–CA. R: Pyrene–tetracyanoethylene. S: Pyrene–CA. T: Anthracene–CA. U: Hexamethylbenzene–CA. V: Naphthalene–TCNE. X: Anthracene–pyromellitic dianhydride (PMDA). Y: Anthracene–tetracyanobenzenze. Z: Phenanthrene–PMDA. The molecular structures are shown in Supplementary Fig. 1.

ambient pressure. Furthermore, **2S**–**F$_4$** exhibited a transition in electrical conductivity to a slightly less conductive phase accompanied by significant changes in its optical, dielectric, and magnetic properties. This emphasizes the unique electronic structure of the mixed-stack complex at the N–I boundary that displays a structural perturbation between electron-itinerant and localized states.

## Results and discussion

### Molecular electronic structures

First, **2S** with non-bulky methylthio groups was synthesized in 52% yield after three-step transformations from 2,2′-bi(3,4-ethylenedithiothiophene) **1**[32] (Fig. 3a, Supplementary Note 3, Supplementary Figs. 7, 41, 42, and Supplementary Table 1). The electrochemical properties of **2O**[30] and **2S** were analyzed via cyclic voltammetry (CV). The voltammograms revealed the first oxidation potentials at 0.622 and 0.805 V (*vs.* Ag/AgCl) for **2O** and **2S**, respectively (Supplementary Note 4 and Supplementary Fig. 3), estimating HOMO levels of **2O**/**2S** (−4.94/−5.13 eV) that are comparable to the LUMO levels of **F$_4$**/**F$_2$** (−4.98/−4.74 eV). The gaps between the donor HOMO and acceptor LUMO (i.e., $\Delta E_{REDOX}$ in Fig. 2) are estimated to be in the range of −0.04 to 0.39 eV (Table 1), which is consistent with those predicted by density functional theory (DFT) calculations (Fig. 4a, Supplementary Note 5, Supplementary Figs. 4, 6, Supplementary Tables 6, 7, 9, and 11). The 1$e^-$-oxidized **2O**/**2S** structures show highly symmetrical molecular orbital shapes spread over the entire molecules and consistent horizontally nodal patterns with an average periodicity of 2.0 Å that correspond well with those of 1$e^-$-reduced **F$_4$**/**F$_2$** with an average periodicity of 1.7–1.8 Å (Fig. 4b, Supplementary Fig. 5, Supplementary Tables 8, 10, and 12). Therefore, the combination of **2O**/**2S** as donors and **F$_4$**/**F$_2$** as acceptors ideally has similar energy levels and orbital symmetries between the HOMO of the donor and LUMO of the acceptor, possibly leading to strong hybridization between their orbitals during complexation.

### Single-crystal X-ray diffraction analyses

Motivated by the potentially ideal combination of **2O**/**2S** donors and **F$_4$**/**F$_2$** acceptors in the electronic structures, we investigated their complexation. The donors and acceptors were mixed in dichloromethane or THF, which led to a gradual color change of the solutions

**Fig. 3 | Synthesis of mixed-stack complexes 2$X$–F$_n$. a** Synthesis of donor **2S** by bromination followed by lithiation and methylthiolation. **b** Synthesis of mixed-stack complexes **2$X$–F$_n$** by chemical oxidation of donor (**2O** and **2S**) by acceptors (**F$_4$** and **F$_2$**). The $\delta$ values were determined using Kistenmacher's equation[37]. NBS *N*-bromosuccinimide, *$^n$Bu* *n*-butyl, THF tetrahydrofuran, Me methyl.

## Table 1 | Structural information and physical properties of mixed-stack complexes

| Donor (D)–acceptor (A) | 2O–F₄ | 2O–F₂ | 2S–F₄ | 2S–F₂ |
|---|---|---|---|---|
| Experimental data | | | | |
| $\Delta E_{REDOX}$ ( $= E_{1/2}^{1}$(**D**) $- E_{1/2}^{1}$ (**A**)) (V)[a] | −0.04 | 0.20 | 0.15 | 0.39 |
| **D**–**A** interplanar distance (Å)[b] | 3.361 | 3.329 | 3.406 | 3.398 |
| δ from bond length analyses in **A** | 0.79(2) | 0.71(4) | 0.69(2) | 0.46(3) |
| σ at 300 K ($\sigma_{rt}$; S cm⁻¹) | $4.9 \times 10^{-4}$ | $1.4 \times 10^{-2}$ | 0.10 | $6.9 \times 10^{-3}$ |
| $h\nu_{CT}$ (eV) | 0.83 | 0.73 | 0.64 | 0.50 |
| $E_a$ for high temperature region (eV)[c] | 0.113(1) (290–315 K) | 0.178(1) (259–337 K) | 0.200(1) (288–340 K) | 0.112(1) (288–312 K) |
| $E_a$ for low temperature region (eV)[c] | 0.215(2) (238–258 K) | 0.225(4) (219–231 K) | 0.277(3) (228–273 K) | 0.0902(17) (221–244 K) |
| Calculated data | | | | |
| $E_g$ (eV)[d] | 0.05 | <0.01[e] | <0.01 | 0.02[e] |
| **W** (eV)[d] | 0.86 | 0.89[e] | 0.88 | 0.89[e] |
| $t_{DA}$ (eV)[f] | 0.203 | 0.209[e] | 0.208 | 0.206[e] |
| $U_{eff}$ (**D**) (eV)[f] | 2.31 | 2.28[e] | 1.89 | 1.89[e] |
| $U_{eff}$ (**A**) (eV)[f] | 2.52 | 2.48[e] | 2.19 | 2.22[e] |

[a]Estimated from the cyclic voltammograms (see Supplementary Fig. 3 for details). The values were calibrated using the energy level of ferrocene/ferrocenium (Fc/Fc+ vs. Ag/AgCl/1 M KCl measured under identical conditions) as the reference.
[b]Determined by measuring the distances between the centroid of the ten atoms of the bithiophene in the donor and the mean plane for the eight quinoid carbons in the acceptor in the single-crystal structures.
[c]Determined from ρ–T plots using the Arrhenius equation.
[d]Calculated by OpenMX[38,39] as a sum of dispersions for the bonding and antibonding bands between donor HOMO and acceptor LUMO (i.e., $W_B$ and $W_A$, respectively) and energy gap between bonding and antibonding bands (i.e., $E_g$), if present (i.e., bandwidth $W = W_B + W_A + E_g$).
[e]Structural data with major occupancy were used in the calculations for **2O–F₂** and **2S–F₂**.
[f]Calculated by Q$_{\text{UANTUM}}$ Espresso (QE)[40,41] and RESPACK[42].

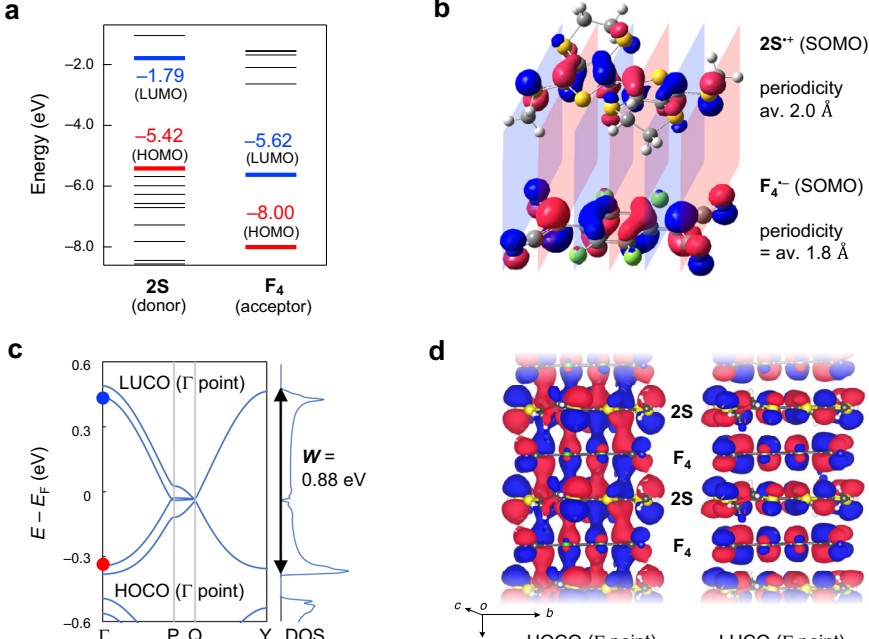

**Fig. 4 | Electronic structure of donor 2S, acceptor F₄, and the mixed-stack complex 2S–F₄ according to theoretical calculations. a** Energy levels of orbitals for neutral **2S** donor and **F₄** acceptor calculated by the Gaussian09 program. The energy levels of the donor's highest occupied molecular orbital (HOMO) and acceptor's lowest unoccupied molecular orbital (LUMO) were comparable, which is appropriate for their strong hybridization. **b** Singly occupied molecular orbitals (SOMOs) of 1e⁻-oxidized **2S** donor and 1e⁻-reduced **F₄** acceptor calculated by Gaussian09 program. These orbitals have horizontally nodal patterns with an average periodicity of 1.8–2.0 Å that correspond well with each other

(Supplementary Fig. 5). **c** Band structure calculated by OpenMX[38,39]. The bandwidth (**W**) value was determined from the calculated density of states (DOS). The highest occupied crystal orbital (HOCO) and lowest unoccupied crystal orbital (LUCO) at the Γ-point are depicted in red and blue dots, respectively. Γ (0,0,0), P (−0.5,0,0.5), Q (−0.5,0.5,0.5), Y (0,0.5,0). **d** Highly hybridized HOCO and LUCO between donor and acceptor at the Γ-point calculated by OpenMX[38,39]. Atoms were colored as follows; white: hydrogen; gray: carbon; blue: nitrogen; red: oxygen; light green: fluorine; yellow: sulfur. Molecular and crystal orbitals with positive and negative phases were colored with magenta and navy, respectively.

from yellow to dark green, indicative of charge transfer from the donor to the acceptor. After slow solvent evaporation over more than three days, dark-green needle-like single crystals of complexes $2O-F_4$, $2O-F_2$, $2S-F_4$, and $2S-F_2$ were isolated (Fig. 3b). These syntheses were highly reproducible and scalable to a yield of several milligrams in a single sequence of operations. The salts had an astonishingly high solubility in organic solvents such as acetonitrile and were stable for at least several weeks under atmospheric conditions.

X-ray diffraction (XRD) analyses determined that all single-crystal structures belonged to the $P2_1/n$ space group with inversion centers at the center of gravity of the donors and acceptors and the glide plane symmetry on the 1/2 translation operation along the $(a + c)/2$ direction (Supplementary Note 6, Supplementary Figs. 9, 10, 12, 14, and Supplementary Table 2). The $2O$ and $2S$ donors in single-crystal complexes are nearly planar (Supplementary Fig. 11), similar to the charge-transfer salts of $2O$ and $2S$ (i.e., $2O \cdot BF_4$[30] and $2S \cdot BF_4$; Supplementary Figs. 2, 8, and Supplementary Table 1). The fluorine atoms in $2O-F_2$ and $2S-F_2$ exhibited positional disorder, with an occupancy of 94:6 and 86:14, respectively (Supplementary Fig. 9b, d). The donors and acceptors are uniformly and alternatingly π-stacked in columns along the $a$-axis. The donor–acceptor interplanar distances depend on the molecular size of the donor (i.e., $2O < 2S$) and acceptor (i.e., $F_2 < F_4$); the distances increased in the order $2O-F_2$ (3.329 Å) < $2O-F_4$ (3.361 Å) < $2S-F_2$ (3.398 Å) < $2S-F_4$ (3.406 Å) (Table 1). Significant intracolumnar short contacts were not observed, suggesting the existence of 1D electronic structures favorable for strong intermolecular interactions. Notably, X-ray diffuse scattering was observed, reminiscent of structural fluctuation (Supplementary Note 6 and Supplementary Fig. 14).

## Charge-transfer degree of mixed-stack complexes

The $\delta$ values of the complexes were determined using the conventional bond length analyses of the single-crystal structures for donors and acceptors[6] in the mixed-stack complexes. An increase in the $\delta$ value of the donor from 0 (i.e., a neutral form with a benzenoid structure) to 1 (a radical cation form with a resonance structure) weakens bond alternation (i.e., C–C bonds $b$ and $d$ and C–S bonds $g$ are shortened, while C=C bonds $a$ and $c$ are lengthened; Fig. 5a, b, c, e, f), as is apparent from the single crystals for neutral $2X$ ($X = O$[30] or S, $\delta = 0$; Supplementary Fig. 7 and Supplementary Table 1) and its 1e⁻-oxidized form $2X \cdot BF_4$[30] ($\delta = 1$; Supplementary Fig. 8 and Supplementary Table 1). All bond length alterations of the $2O$ and $2S$ donors in the mixed-stack complexes are located between those of $2X$ and $2X \cdot BF_4$, indicative of intermediate $\delta$ values for the complexes between 0 and 1. Commonly, $F_4$ complexes exhibited weaker bond length alternation than $F_2$ complexes, indicating that the $F_4$ complexes have higher $\delta$ values than the $F_2$ complexes. Likewise, the acceptor molecules $F_n$ exhibited weakening of the bond alternation (i.e., C–C bonds $i$, $k$, and $m$ are shortened, while C=C bonds $j$ and $l$ are lengthened; Fig. 5a, d, g) upon reduction from neutral[33,34] ($\delta = 0$) to their 1e⁻-reduced forms[35,36] ($\delta = 1$), suggesting the intermediate $\delta$. Notably, the $2O$ complexes exhibited weaker bond length alternation than the $2S$ complexes, suggesting that the $2O$ complexes have higher $\delta$ values than the $2S$ complexes. These tendencies are consistent with the $\delta$ values determined using Kistenmacher's equation[37], which assumes that the bond length changes linearly as $\delta$ increases from 0 to 1. The $\delta$ values of $2O-F_4$, $2O-F_2$, $2S-F_4$, and $2S-F_2$ were determined to be 0.79(2), 0.71(4), 0.69(2), and 0.46(3), respectively (Table 1). These values are close to the N–I boundary $(0.59 < \delta < 0.74)$, whereas that of $2S-F_2$ lies at the nearly neutral state, as predicted by the comparable energy levels of the donor HOMO and acceptor LUMO by CV analyses (Supplementary Fig. 3) and DFT calculations (Fig. 4a and Supplementary Fig. 6).

## Electronic structures of mixed-stack complexes

Based on the single-crystal structures of $2X-F_n$ as the average structures, we then obtained the band structures using first-principles

calculations (OpenMX software; Supplementary Note 7)[38,39], wherein the contributions from Coulomb repulsion between the carriers (discussed later) are ignored. All complexes showed band structures with highly dispersed bonding and antibonding orbital-derived bands formed by the donor HOMO and acceptor LUMO (Fig. 4c and Supplementary Fig. 18). The bands were exclusively dispersed along the $a$-axis (i.e., Γ–X corresponding to the π-stacking direction). The band structures had small or negligible energy gaps ($E_g$) in proximity to the Fermi level (0.02–0.05 eV for $2O-F_4$ and $2S-F_2$ and <0.01 eV for $2O-F_2$ and $2S-F_4$; Table 1 and Supplementary Fig. 18). The density of states (DOS) was calculated to determine the bandwidth ($W$) of the complexes in the major occupancies, the values of which were 0.86 eV ($2O-F_4$), 0.89 eV ($2O-F_2$), 0.88 eV ($2S-F_4$), and 0.89 eV ($2S-F_2$) (Table 1, Fig. 4c, and Supplementary Fig. 18), where $W = W^B + W^A + E_g$; $W^B$ and $W^A$ are the band dispersions of the bonding and antibonding orbitals, respectively. Among typical organic conductors[2], high $W$ values may be characteristic of 1D electronic structures, especially given the comparable energy levels of the donor HOMO and acceptor LUMO and their well-matched orbital symmetries, indicating strong intracolumnar interactions. We next quantified the transfer integrals between a donor and the six neighboring molecules in the single-crystal structures using combined QUANTUM ESPRESSO (QE)[40,41] and RESPACK[42] (Supplementary Fig. 27, and Supplementary Table 3). These calculations also identified prominent intracolumnar donor–acceptor interactions ($t_{DA} \approx 0.21$ eV in Table 1; $t_1$ and $t_4$ in Supplementary Table 3 and Supplementary Fig. 27), corresponding to $W$ values based on the 1D tight-binding model, and negligible intercolumnar donor–acceptor and donor–donor interactions for all complexes. These findings support their 1D electronic structures, which were also suggested by the band structures. The $W$ and intracolumnar $t_{DA}$ values follow nearly consistent trends: $F_4$ complexes < $F_2$ complexes, possibly because the donor–acceptor interplanar distances are shorter for $F_2$ complexes (as shown in Table 1). Despite the longer interplanar distances, the trend $2O$ complexes < $2S$ complexes is primarily due to the presence of multiple S atoms with significant orbitals in the donor.

Notably, we have made a significant discovery using OpenMX calculations[38,39], which identified that the HOMO of the donor and LUMO of the acceptor are highly hybridized. The calculated highest occupied crystal orbital (HOCO) and lowest unoccupied crystal orbital (LUCO) at the Γ point of $2X-F_n$ appeared equivalently on the donor and acceptor, respectively (Fig. 4d and Supplementary Fig. 19). This contrasts with neutral complexes[3], where HOCOs and LUCOs were localized on the donors and acceptors, respectively. This finding supports that $2X-F_n$ complexes around the N–I boundary have different electronic structures from previously reported neutral or ionic mixed-stack complexes[5,6,15–23], leading to significant $W$ values based on their 1D electronic structures and unique electronic structures as discussed later.

## Polarized reflectivity measurements

The polarized reflectivity spectra further support the observation that the $\delta$ values of $2X-F_n$ are close to the N–I boundary. The spectra obtained for the π-stacking direction of the single crystals have peak energies based on the charge-transfer band ($h\nu_{CT}$) at 0.50 eV ($2S-F_2$) < 0.64 eV ($2S-F_4$) < 0.73 eV ($2O-F_2$) < 0.83 eV ($2O-F_4$) (Table 1 and Fig. 6a; complexes are located exactly at the N–I boundary, whereas $2S-F_2$ is nearly neutral). The spectrum of $2S-F_2$ has a shoulder peak at approximately 0.64 eV. The shape of this spectrum reflects the neutral-to-N–I boundary state[43] ($\delta = 0.46(3)$) or donor–acceptor π-dimerized form[43,44] (discussed later), possibly causing the plot for $2S-F_2$ to deviate from the predicted value by $\delta$ in Fig. 2. The significant positional disordering of fluorine atoms in $2S-F_2$ may also affect the shape of the spectrum.

Given that the strong hybridization between the donor HOMO and acceptor LUMO in the band structures with nearly negligible $E_g$,

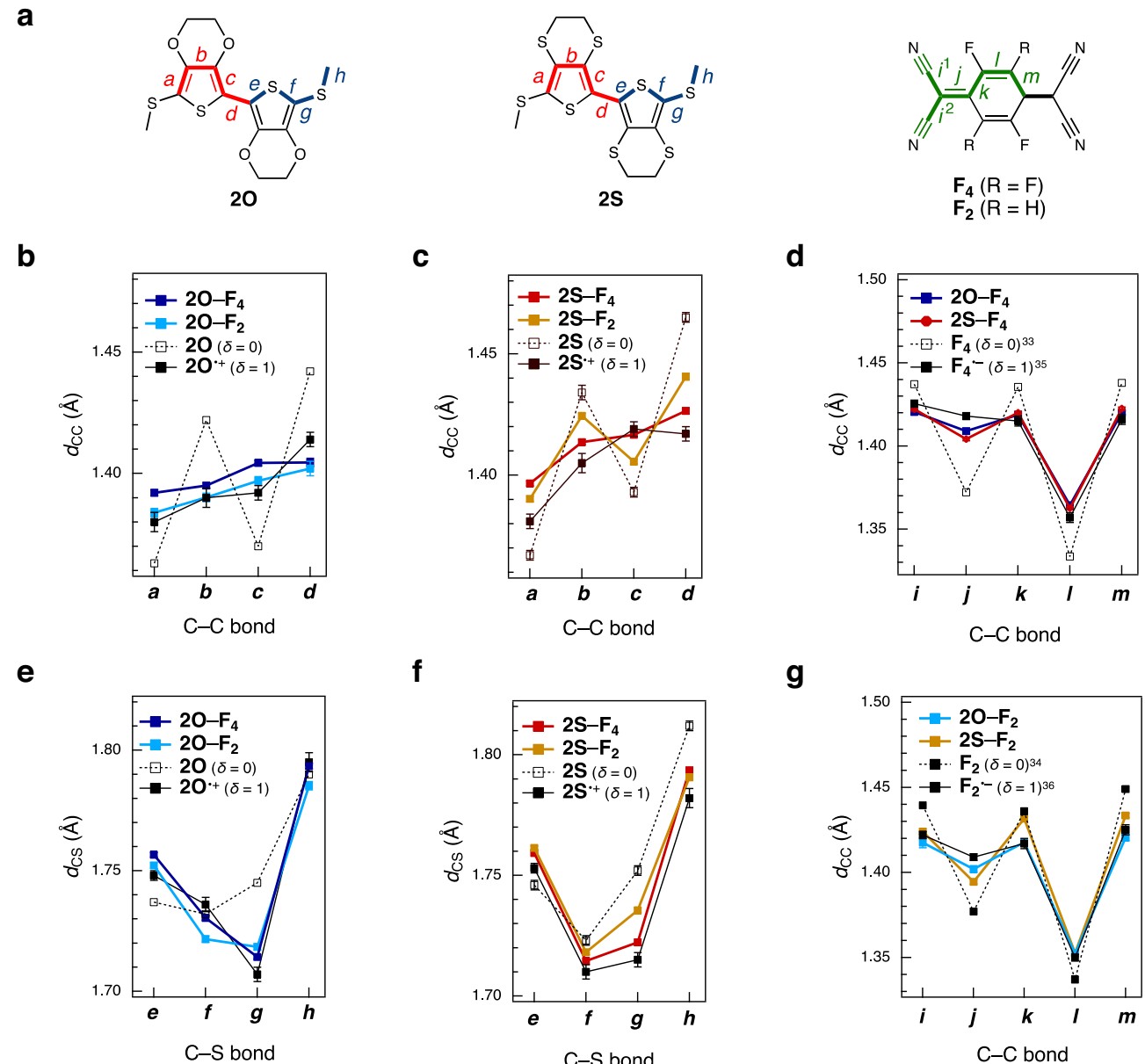

**Fig. 5 | Bond length analysis of mixed-stack complexes. a** Bond labels of **2O**, **2S**, **F₄**, and **F₂**. **b**–**g** Comparison of C–C and C–S bond lengths ($d_{CC}$ and $d_{CS}$, respectively) of donors and acceptors in single crystals of mixed-stack complexes, with error bars (standard deviation). The letters along the *x*-axes of the plots correspond to the bonds depicted in **a**. As a reference, the bond lengths in single crystals of neutral donors or acceptors (i.e., $\delta = 0$) are shown with dotted lines and $1e^-$-oxidized donors or $1e^-$-reduced acceptors (i.e., $\delta = 1$) are shown with solid lines. Observed $d_{CC}$ of **2O**[30] (**b**), $d_{CC}$ of **2S** (**c**), $d_{CC}$ of **F₄**[33,35] (**d**), $d_{CS}$ of **2O**[30] (**e**), $d_{CS}$ of **2S** (**f**), and $d_{CC}$ of **F₂**[34,36] (**g**). $i = (i^1 + i^2)/2$.

we considered the electronic structures of **2X**–**Fₙ** to be like homogeneous donor- or acceptor-stacked charge-transfer salts, with the average structure[2]. Under this consideration, the observed $hv_{CT}$ may correspond to the difference between the bonding and antibonding orbital bands separated by carrier-to-carrier Coulomb repulsions in the solid states ($U_{eff}$), rather than the donor HOMO–acceptor LUMO energy gap for neutral complexes. Accordingly, the lower $hv_{CT}$ for the **2S** complexes than that for the **2O** complexes may correspond to lower $U_{eff}$ for the **2S** complexes. Quantum calculations provided further insights into electronic structures. The $U_{eff}$ values for the complexes were quantified for donors and acceptors (i.e., $U_{eff}$ (**D**) and $U_{eff}$ (**A**), respectively) by combining first-principles calculations (QE[40,41]/RESPACK[42] packages) based on the average single-crystal structures. These calculations identified the far lower $U_{eff}$ (**D**) and $U_{eff}$

(**A**) for **2S** complexes compared to those of **2O** complexes (Table 1, Supplementary Figs. 20–26, and 28–33), emphasizing that the combined effects of smaller $\delta$ value and the larger π-conjugate area of the donor in the complexes may favorably contribute to lowering $U_{eff}$. The cooperative reduction in $U_{eff}$ (**D**) and $U_{eff}$ (**A**) supports the significant orbital hybridization of donors and acceptors in single crystals. According to these calculations, the **2S** complexes have slightly larger $W$ and far lower $U_{eff}$ than the **2O** complexes, conferring superior electrical conductivities[45–48] upon the **2O** complexes. However, it is not negligible that nearly neutral **2S**–**F₂** has insufficient conductive carriers and positional disorders of fluorine atoms, which may impact the electronic structures that contribute to the conductivity (Supplementary Figs. 18–20, 26, 27, 33, and Supplementary Table 3).

**a** 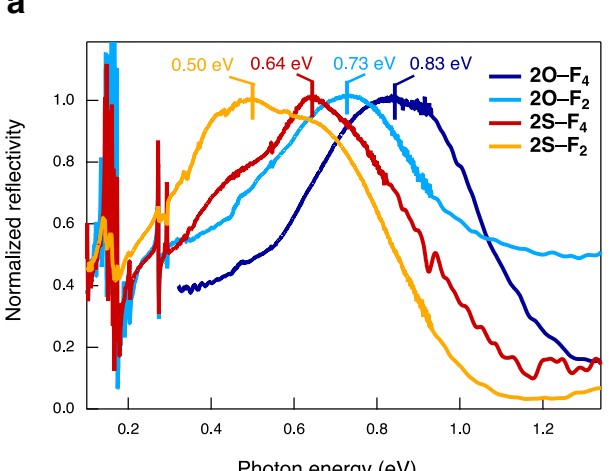

**b** 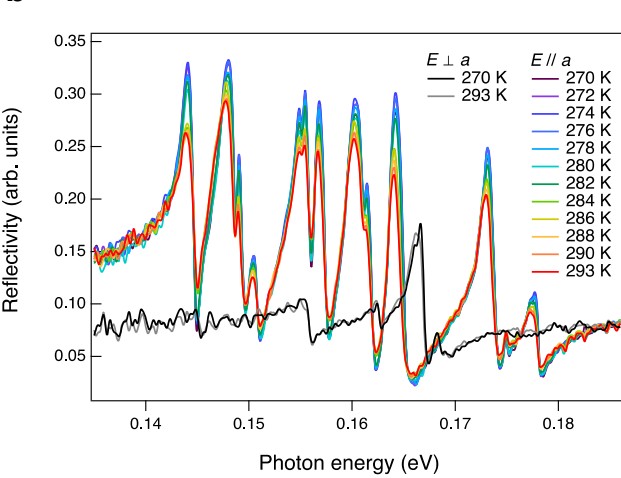

**Fig. 6 | Polarized reflectivity of mixed-stack complexes. a** Spectra of mixed-stack complexes obtained in the electric field of light parallel to the π-stacking direction. Bars indicate the top of the peaks. The spectrum of the relatively thin single-crystal **2O–F₄** exhibits stripe-pattern signals in the low-energy region due to interference from the backside (the raw data is shown in Supplementary Fig. 34a), and thus this part of the spectrum was deleted for clarity. **b** Temperature-dependent reflectivity of **2S–F₄** in the electric field of light parallel (//) and perpendicular (⊥) to the π-stacking directions for the low-energy region.

## Structural fluctuations involved in π-dimerization

It is also notable that the polarized infrared reflectivity spectra of the complexes along the π-stacking direction exhibited multiple sharp peaks in the low-energy region of 0.14–0.19 eV, except for nearly neutral **2S–F₂** (Fig. 6b, Supplementary Figs. 34, and 35a). The optical conductivity spectrum of **2S–F₄** (Supplementary Fig. 35b) derived from the infrared spectrum showed a coincidence of the shapes to the Raman spectrum at 293 K (Supplementary Fig. 35d), with varying intensities. Considering the inversion center of the single-crystal structure in a $P2_1/n$ space group with uniform π-stacked donors and acceptors along the $C2$ glide symmetry, the Raman-active modes should not be visible in IR, while the IR-active modes should not be visible in Raman. The coincidence of the spectral shapes indicates the dynamic fluctuating of donor–acceptor-π-stacking dimerization based on the electron-molecular vibration (EMV) couplings[43,44]. We performed a computational study by numerical simulations using the donor–acceptor dimer model[49], confirming that the EMV coupling effect is responsible for these signals (Supplementary Note 8, Supplementary Fig. 35c, Supplementary Tables 4, and 5). These results suggest that there is inhomogeneity in the one-dimensional molecular stacking, contradicting the uniform columnar structure predicted by single-crystal XRD (Supplementary Table 2 and Supplementary Fig. 10). This inhomogeneity likely indicates dynamic fluctuations from the average structure, which is supported by the X-ray diffuse scattering observed in XRD at 300 K (Supplementary Fig. 14).

## Electrical conductivity measurements

Motivated by the proximity of the $\delta$ value to the N–I boundary, high **W**, and low **U$_{eff}$** for **2S** complexes, we subsequently investigated the electrical conductivity of single-crystal mixed-stack complexes (Supplementary Note 9). The $\sigma_{rt}$ values (300 K) determined via the direct-current method along the π-stacking direction (the $a$-axis) were superior by a few orders of magnitude to those of previously reported typical mixed-stack complexes with $\sigma_{rt}$ (300 K) below $10^{-4}$ S cm⁻¹: $4.9 \times 10^{-4}$ S cm⁻¹ (**2O–F₄**), $1.4 \times 10^{-2}$ S cm⁻¹ (**2O–F₂**), 0.10 S cm⁻¹ (**2S–F₄**), and $6.9 \times 10^{-3}$ S cm⁻¹ (**2S–F₂**) (Fig. 1b and Table 1) within the ohmic region (Supplementary Fig. 36). Except for the nearly neutral **2S–F₂**, the trend of $\sigma_{rt}$ (**2S–F₄** > **2O–F₂** > **2O–F₄**) indicates that the closeness of these complexes to the N–I boundary dominantly affected the conductivity. The $\sigma_{rt}$ of **2S–F₄** (0.10 S cm⁻¹) is the highest value reported to date for a single crystal of a 1D mixed-stack complex, and

it is an order of magnitude higher than the previously highest value of $1.0 \times 10^{-2}$ S cm⁻¹ [21]. The low **U$_{eff}$** based on the large conjugate area may cooperatively contribute to the high $\sigma_{rt}$ [46–48], unlike those of the other complexes at the N–I boundary (e.g., complexes $e'$ and $f$ in Fig. 1b).

The activation energies ($E_a$) near room temperature were determined from the temperature ($T$) dependence of the resistivity ($\rho = \sigma^{-1}$) using the Arrhenius equation. The $\rho$–$T$ plots are indicative of semiconducting behavior with relatively small $E_a$ values: 0.113(1) eV (**2O–F₄**), 0.178(1) eV (**2O–F₂**), 0.200(1) eV (**2S–F₄**), and 0.112(1) eV (**2S–F₂**) around room temperature (Figs. 7, and 8a, and Table 1), indicating that these complexes are narrow-bandgap semiconductors. The $E_a$ trend was inconsistent with those of $\sigma_{rt}$, the calculated $E_g$, and **U$_{eff}$**. Except for the nearly neutral **2S–F₂**, the $E_a$ values increased: **2O–F₄** < **2O–F₂** < **2S–F₄**. The trend is explained by the proximity of $\delta$ to the N–I boundary which induces the π-dimerization fluctuation between the donor and acceptor; **2S–F₄** at the N–I boundary showed largest increase in $\rho$, and highest $E_a$ among the three complexes.

## Abrupt changes in physical properties of 2S–F₄ around room temperature

The mixed-stack complex with the highest conductivity at the N–I boundary, **2S–F₄**, exhibited a unique transition with a sharp increase in $\rho$ at a transition temperature ($T_c$) of 282 K. The transition was followed by an increase in $E_a$ from 0.200(1) eV at high temperatures (288–340 K) to 0.277(3) eV at low temperatures (228–273 K) without significant hysteresis upon cooling and heating (Figs. 7, 8a, and Table 1). The XRD pattern in $P2_1/n$ space group at 300 K showed significant X-ray diffuse scattering (Supplementary Fig. 14), possibly due to the precursor phenomenon for the π-dimerization, which may result in EMV coupling-based signals at room temperature (Fig. 6b, Supplementary Figs. 34, and 35a). On the other hand, the scattering disappeared (Supplementary Figs. 15 and 16) at 200 K, and the XRD showed superlattice patterns with dimensions of $a \times 2b \times 2c$ (Supplementary Figs. 13 and 17), possibly implying the π-dimerization along the $a$-axis (Supplementary Note 6). The broken symmetry upon the possible π-dimerization of the complex may induce the markedly intensified EMV coupling-based signals (e.g., from 0.20 to 0.25) near 282 K during the cooling process while maintaining the energy (Figs. 6b, 8b, and Supplementary[50] Fig. 35a), as well as the large anomaly in the dielectric constants[50] (the real part of the dielectric constant is shown in Fig. 8c).

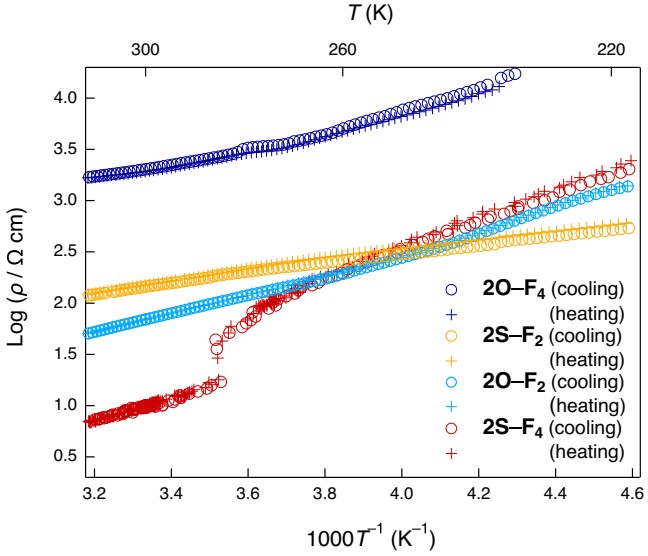

**Fig. 7 | Temperature-dependent electrical conductivities of mixed-stack complexes.** $\rho$–T plots obtained by a four-probe method for cooling (circles) and heating (plus marks) processes are shown.

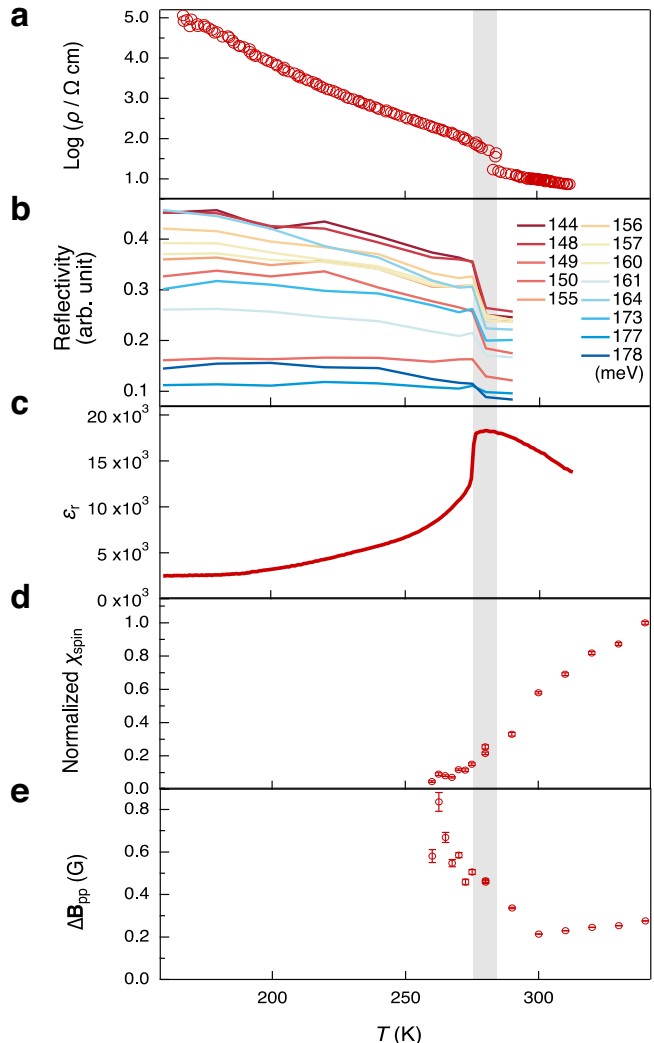

**Fig. 8 | Temperature-dependent physical properties of single-crystal 2S–F₄ upon cooling. a** Electrical resistivity. **b** Polarized reflectivity for photon energies of 144–178 meV in the electric field of light parallel to the π-stacking direction. **c** Real part of the dielectric constant $\varepsilon_r$ at 1.0 MHz. **d** Normalized $\chi_{spin}$ determined from the calculation using the intensity and $\Delta B_{pp}$ of the electron spin resonance (ESR) signals at $g \approx 2.003$. The values are normalized by the value at 340 K, with error bars (standard deviation). **e** $\Delta B_{pp}$ of the ESR signals at $g \approx 2.003$, with error bars (standard deviation). Gray panel indicates the transition temperature around 282 K.

The transition events did not cause any significant hysteresis for the cooling and heating processes.

## Change in the magnetic characteristics at the transition

The magnetic characteristics revealed via temperature-dependent electron spin resonance (ESR) measurements helped us to address the insights into the electronic structures of the mixed-stack complexes (Supplementary Note 10). The spectra of **2S–F₄** exhibited a narrow Lorentzian signal with a peak-to-peak width ($\Delta B_{pp}$) of ≈0.2 G at $g \approx 2.003$, which is typical for π-electrons[2] (Supplementary Figs. 37–39). At 282 K, the magnetic susceptibility ($\chi_{spin}$) quantified from the spectra decreased sharply to reach a nearly nonmagnetic state (Fig. 8d and Supplementary Fig. 39b), suggesting a spin-Peierls-like singlet formation based on the 1D electronic structure[51] during the π-dimerization. This $\chi_{spin}$ decrease was accompanied by an increase in $\Delta B_{pp}$ (Fig. 8e and Supplementary Fig. 39c, d), supporting increased antiferromagnetic interactions in the singlet ground state in the low-temperature region. A decrease in $\chi_{spin}$ and increase in $\Delta B_{pp}$ upon cooling were likewise observed for **2O–F₂** and **2O–F₄** at approximately 240 and 200 K, respectively (Supplementary Fig. 39b–d), which may correspond to the increase in $E_a$ in the electrical conductivity at low temperatures (Figs. 7, 8a, and Table 1). The signals of **2S–F₂** were negligible mainly due to the relatively poor electron spins in the nearly neutral state. These magnetic behaviors were also consistent with the decrease in the magnetic susceptibilities ($\chi$) of the polycrystals of **2S–F₄** and **2O–F₄** around the temperatures using a superconducting quantum interface device (SQUID) magnetometer (Supplementary Note 11 and Supplementary Fig. 40). The $\chi$ values of **2S–F₄** and **2O–F₄** were quantified to be $1.2 \times 10^{-4}$ and $2.6 \times 10^{-4}$ emu mol⁻¹, respectively at 293 K. The $\chi$ values were relatively high compared to those of previously reported mixed-stack complexes[18,50,52], most of which are in nearly nonmagnetic states. These values are comparable to those of typical organic conductors[2], emphasizing the presence of abundant conductive electrons.

Surprisingly, in the case of **2S–F₄**, another broad signal with a $\Delta B_{pp}$ value of ~3 G appeared at $g \approx 2.004$ below 282 K (Supplementary Figs. 37–39), and overlapped with the narrow signal at $g \approx 2.003$. This suggests the appearance of nearly non-correlated spins (e.g., spin solitons) originating from unpaired donors or acceptors during transient partial-π-dimerization between donor and acceptor. This

observation suggests that the conductive mechanism of **2S–F₄** at the N–I boundary may involve spin solitons that exist because of topological excitations[29] that contribute to its high conductivity.

The abrupt change in the electrical, optical, dielectric, and magnetic properties of **2S–F₄** at high $T_c$ (282 K) under ambient pressures (Fig. 8) may reflect structural perturbation, which can be characteristic of the mixed-stack complexes at the N–I boundary. The neutral[20,53] or highly ionic complexes[50] transitioned based on π-dimerization to form a structure with more strongly ionic behavior at lower $T_c$ values of 49–81 K. The complexes around the N–I boundary ($\delta = 0.3$–0.5) had relatively high $T_c$ values of 115–220 K[20,52] under ambient pressures. Under high-pressure conditions (approximately 9 kbar), TTF-CA ($\delta \approx 0.6$) transiently exhibits excellent conductivity (7 S cm⁻¹) and a high $T_c$ at approximately 270 K[29], whereas it has a low $T_c$ of 81 K under ambient pressures ($\delta = 0.2$, $\sigma_{rt} = 10^{-5}$ S cm⁻¹; $\nu$ in Fig. 1b). The high $T_c$ (282 K) and high $\sigma_{rt}$ (0.1 S cm⁻¹) even at ambient pressures of **2S–F₄**, for which $\delta$ is exactly at the N–I boundary ($\delta = 0.69(2)$) with non-hysterical changes in their electronic functionalities such as electrical

conductivity, reflectivity, dielectricity, and magnetism may involve the unique electronic perturbation at the N−I boundary between the electron-itinerant and localized states.

In conclusion, we developed highly conductive 1D mixed-stack complexes for which $\delta$ is close to the N−I boundary by combining a donor HOMO and acceptor LUMO that have similar energy levels and well-matched orbital symmetries. These complexes have highly hybridized orbitals between the donor and acceptor and exhibited high $\sigma_{rt}$ ($10^{-4}$ to $0.1\,S\,cm^{-1}$) under ambient conditions; the top $\sigma_{rt}$ ($0.1\,S\,cm^{-1}$) for **2S**–**F$_4$** at $\delta = 0.69(2)$ is the highest value reported to date for a single-crystal 1D mixed-stack complex. **2S**–**F$_4$** uniquely exhibited a non-hysterical change in its electrical conductivity at high $T_c$ (282 K) under ambient pressure to a less conductive phase below this temperature. This behavior was simultaneously accompanied by abrupt changes in the low-energy EMV coupling-based reflectivity, dielectric constant, and magnetic properties, reminiscent of the π-dimerization fluctuation that spatiotemporally breaks the symmetry and may widen the band gap to form a band-insulating state and increase $E_a$. Furthermore, the appearance of a broad ESR signal suggests non-correlated spins of unpaired donors or acceptors, which may enhance the conductivity of **2S**–**F$_4$** through mechanisms such as topological excitation[29]. In addition, mixed-stack complexes have unique advantages as conductor materials, such as high potential for solution processing and scalable synthesis without special equipment. The high conductivity, high solubility, and air stability of the conductors may provide scope for further investigation of the fundamental physical properties and applications using various external stimuli[29,50,53,54]. The rich variations in the structure-determining factors in oligomer donors, e.g., monomer unit variations, oligomer length[55,56], sequences comprising multiple units[56], and end-capping structures could enable precise tuning of the electronic structures toward highly functional mixed-stack complexes.

## Methods

### Apparatus
Proton ($^1H$) and carbon ($^{13}C$) nuclear magnetic resonance (NMR) spectra were obtained on a JEOL JNM-AL300 ($^1H$ NMR: 300 MHz; $^{13}C$ NMR: 75 MHz) spectrometer. $^1H$ NMR spectra in CDCl$_3$ were referenced internally to tetramethylsilane as a standard. $^{13}C$ NMR spectra in CDCl$_3$ were referenced to the solvent resonance. High-resolution mass spectra (HRMS) were obtained on a JEOL JMS-AX500 with a field desorption (FD) probe in the positive mode using cholesterol as an internal standard. Single-crystal XRD measurements were performed using a Rigaku MercuryII CCD X-ray diffractometer (Mo $K_\alpha$, $\lambda = 0.71073\,Å$) and BL02B1 ($\lambda = 0.30960\,Å$) at a synchrotron facility SPring-8 in Japan. A N$_2$-gas-blowing device was employed for the low-temperature measurements. Cyclic voltammetry was performed using an ALS 610DB electrochemical analyzer. Microscopic mid-infrared (mid-IR) and ultraviolet-visible (UV-vis) measurements were performed on a JASCO FT/IR-6100 and FT/IR-6200 and MSV-5200TSO, both of which are equipped with a JASCO IRT-5000 FT/IR microscope. Raman spectroscopy was performed on a RENISHAW inVia Reflex. Dielectric constant measurements were performed using a custom-built alternating-current (AC) impedance probe with a Solartron Impedance Analyzer SI 1260, a Dielectric Interface 1296, and a Quantum Design SQUID magnetometer (MPMS-XL) as the temperature controller. Resistivity measurements were performed on a Quantum Design Physical Properties Measurement System (PPMS) and HUSO HECS 994C with a high-resistance-low-current electrometer KEYTHLEY 6517B equipped with an ADVANTEST R6142 power supply, an ADVANTEST R7210 scanner, and a KEYTHELY 2001 digital multimeter. X-band continuous wave ESR measurements were performed using a Bruker EMXmicro and EMX spectrometers with a N$_2$ flow variable temperature unit ER4131VT and E500 spectrometer with a He flow cryostat ESR900. Static magnetic susceptibility measurements were performed using a MPMS-XL.

## Synthesis

### Synthesis of 5,5′-bis(methylthio)-2,2′-bi(3,4-ethylenedithiothiophene) 2S.
General synthetic procedure and materials sources are shown in Supplementary Notes 1 and 2. To a solution of 2,2′-bi(3,4-ethylenedithiothiophene)[32] **1** (200 mg, 0.576 mmol) in dichloromethane (20 mL) was dropwise added at 0 °C a solution of N-bromosuccinimide (NBS; 238 mg, 1.34 mmol) in dichloromethane (6 mL), and the reaction mixture was stirred for 6 h. After the addition of a saturated aqueous solution of sodium bicarbonate (100 mL), the mixture was extracted with dichloromethane (3 × 100 mL). The combined organic layer was dried over sodium sulfate and concentrated in vacuo to afford a crude material containing **2** (250 mg) which was used for the subsequent reaction without further purification. Thus, to a solution of the mixture (250 mg) in THF (12 mL) was added $^nBuLi$ (1.6 M in hexane, 1.70 mL, 2.68 mmol) at −80 °C. The reaction mixture was stirred for 2 h and then allowed to warm to ambient temperature. To the mixture was dropwise added dimethyl disulfide (500 μL, 5.63 mmol), and the mixture was stirred at ambient temperature for 3 h. After the removal of volatile materials in vacuo and the subsequent addition of a saturated aqueous solution of sodium bicarbonate (100 mL), the mixture was extracted with dichloromethane (3 × 100 mL). The combined organic solvents were dried over sodium sulfate, concentrated in vacuo, and purified by GPC (eluent: chloroform) to afford **2S** (131 mg, 0.298 mmol) as a yellowish white powder in 52% total yield for three-step transformations from **1**. Physical data of **2S**: $^1H$ NMR (CDCl$_3$, 300 MHz) $\delta$ 2.44 (s, 6H), 3.14–3.23 (m, 4H), 3.25–3.34 (m, 4H); $^{13}C$ NMR (CDCl$_3$, 75 MHz) $\delta$ 20.1, 27.6, 27.7, 126.6, 126.6, 127.8, 130.5; MS (FD) calcd for $C_{14}H_{14}S_8$ [M$^+$] 437.8834, found 437.8856. The structural integrity and purity were identified by the NMR spectra (Supplementary Figs. 41 and 42). Block-like orange single crystals were obtained by the liquid–liquid diffusion method with dichloromethane/hexane (1:1, v/v). The single-crystal XRD analyses revealed the structural details (Supplementary Table 1 and Supplementary Fig. 7), confirming a nearly planar structure. In contrast, other solvent combinations afforded crystal-like solids, while their XRD patterns were too complicated to be resolved, suggesting that neutral **2S** is polymorphic and could have multiple structures, potentially influenced by molecular twisting.

### Typical procedure of mixed-stack complexes.
A donor (**2O**[30] or **2S**, 10 μmol) was placed on one side of an H-shaped cell and an acceptor (**F$_4$** or **F$_2$**, 10 μmol) was placed on the other side of the cell, respectively. To the cell were slowly added THF (8 mL for **2O**–**F$_4$** and **2O**–**F$_2$**) or dichloromethane (8 mL for **2S**–**F$_4$** and **2S**–**F$_2$**), and the mixture was kept at 25 °C in the dark. The slow evaporation of solvents from the mixture over more than three days gave dark green needle-like single crystals which were collected by paper filtration and washed with acetonitrile or 1:1 v/v dichloromethane/hexane (typical size of ~500 × ~40 × ~20 μm$^3$). The single-crystal structures and chemical compositions were identified by single-crystal X-ray structural analyses (Supplementary Figs. 9–17 and Supplementary Table 2).

### CV measurements.
We performed CV in a degassed 0.6 mM solution of **2O**[30] or **2S** in dichloromethane containing 100 mM $^nBu_4N\cdot PF_6$. We used a glassy carbon as the working electrode, a platinum wire as the counter electrode, and a silver–silver chloride electrode (Ag/AgCl in 1 M KCl) as the reference electrode (Supplementary Fig. 3).

### Theoretical calculations

**Energy levels and shapes of orbitals for donors and acceptors.** The energy levels of orbitals for neutral donors (**2O**[30] and **2S**), one-electron-oxidized donors (**2O**$^{\cdot+}$ and **2S**$^{\cdot+}$), neutral acceptors (**F$_4$** and **F$_2$**), and one-electron-reduced acceptors (**F$_4$**$^{\cdot-}$ and **F$_2$**$^{\cdot-}$) were calculated. The calculations were performed using the optimized structures on the Gaussian09 program (Revision D.01, Gaussian, Inc., Wallingford CT, 2016.

https://gaussian.com/) at the DFT level with the (unrestricted) B3LYP functional, the gradient correction of the exchange functional by Becke[57,58] and the correlation functional by Lee, Yang and Parr[59] and the 6−31G(d) split valence plus polarization basis set[60−62].

**Band structure calculations.** All the periodic DFT calculations were performed with the single-crystal structures of mixed-stack complexes as the average structures at 300 K. The calculations were performed by the OpenMX software[38,39], based on optimized localized basis functions and pseudopotentials. The basis functions used were H6.0-s2p1, C6.0-s2p2d1, O6.0-s2p2d1, S7.0-s2p2d1f1, N6.0-s2p2d1, and F6.0-s2p2d1 for hydrogen, carbon, oxygen, sulfur, nitrogen, and fluorine, respectively, wherein the abbreviation of basis functions such as C6.0-s2p2d1, C stands for the atomic symbol, 6.0 the cutoff radius (Bohr) in the generation by the confinement scheme, and s2p2d1 means the employment of two, two, and one optimized radial functions for the s-, p-, and d-orbitals, respectively. The radial functions were optimized by a variational optimization method[38,39]. As valence electrons in the pseudopotentials (PPs), we included 1 s for hydrogen, 2 s and 2p for carbon, oxygen, nitrogen, and fluorine, and 3 s and 3p for sulfur, respectively. All the PPs and pseudo-atomic orbitals (PAOs) we used in the study were taken from the database (2019) in the OpenMX website (https://www.openmx-square.org/), which was benchmarked by the delta gauge method[63]. Real-space grid techniques were used for the numerical integrations and the solution of the Poisson equation using FFT with the energy cutoff of 220 Ryd[64]. We performed Brillouin-zone integrations on a $2 \times 1 \times 2$ $k$-grid; the Fermi-Dirac distribution function at 300 K is employed as a smeared occupation function. We used a generalized gradient approximation (GGA) proposed by Perdew, Burke, and Ernzerhof to the exchange-correlation functional[65]. The real parts of the HOCO and LUCO at the Γ point (0,0,0) were visualized by VESTA[66] (Fig. 4d and Supplementary Fig. 19).

**Calculations of transfer integrals.** The first-principles calculations were performed using the QE package[40,41] with the GGA as the exchange-correlation function[65] based on the single-crystal structure data of $2O-F_4$, $2O-F_2$, $2S-F_4$, and $2S-F_2$ at 300 K. In the calculations, SG15 Optimized Norm-Conserving Vanderbilt (ONCV) pseudopotentials[67] were used as the pseudopotentials, in which 428 bands corresponding to the electron numbers were considered. The cutoff kinetic energies for wave functions, charge densities, and the mesh of the wave numbers were set as 80, 320 Ry, and $5 \times 3 \times 3$, respectively. After the calculations, the maximally localized Wannier functions (MLWFs) were obtained using RESPACK[42], and four bands near the Fermi energy were selected to reproduce the MLWFs (Supplementary Figs. 21−26). Initial coordinates of the MLWFs were located at the center of gravity of crystallographically independent two donors and two acceptors in the unit cell, respectively. Supplementary Fig. 20 shows the energy bands near the Fermi energy obtained by QE[40,41] and the Wannier interpolation. Based on the MLWFs, transfer integrals ($t$ in Supplementary Fig. 27) were calculated as summarized in Table 1 and Supplementary Table 3.

**Calculations of Coulomb interactions.** Effective direct Coulomb interactions for donors and acceptors (i.e., $U_{eff}$ (**D**) and $U_{eff}$ (**A**)) were calculated by RESPACK[42], in which the Coulomb interaction was reduced due to screening effects from bands other than the target band. Using a constrained random phase approximation (cRPA) method, we obtained the screened Coulomb interactions by incorporating screening effects from non-target bands. Through previous reports of such applications to different organic material systems[46−48], we know that the effective model Hamiltonians derived by the cRPA method succeed in evaluating the stability of competing phases in electron-correlated systems. Thus, we estimated the Coulomb interactions by using the cRPA method. The energy cutoff for the dielectric function was set as 5.0 Ry. Static screened direct integrals were calculated as the matrix element of the shielded Coulomb interaction between the Wannier functions on the donors and acceptors, respectively (Supplementary Figs. 21−26)[42]. The values at 0 Å were used as the $U_{eff}$ (**D**) and $U_{eff}$ (**A**) (Table 1 and Supplementary Figs. 28−33).

**Optical reflection spectroscopy measurements.** Steady-state polarized reflectivity spectra of mixed-stack complexes ($2O-F_4$, $2O-F_2$, $2S-F_4$, and $2O-F_2$) in the mid-IR region (0.12−0.97 eV) and UV-vis-near IR region (0.46−1.35 eV) were performed using mid-IR and UV-vis spectrometers equipped with a polarizer by applying an electric field ($E$) along the π-stacking directions (i.e., the $a$-axis) and perpendicular to the directions (Figs. 6, 8b, Supplementary Figs. 34, and 35a, b). Raman spectroscopy of $2S-F_4$ was performed using an excitation wavelength of 532 nm (Supplementary Fig. 35d).

**Dielectric constant measurements.** AC impedance spectroscopy measurements to determine dielectric constant were carried out by the quasi-four-probe method in the frequency range of 1.0 MHz for 160−312 K. A direction parallel to the long axis direction of the crystal shape, corresponding to the π-stacking, was chosen for the AC impedance measurements. Electrodes were made on the two opposite sides of the samples using silver paste and gold wires (15 μm diameter).

**Electrical resistivity measurements.** Electrical resistivity ($\rho$) measurements of the single-crystal mixed-stack complexes were performed by the conventional a four-probe method along the long-axis direction of the crystal shape, corresponding to the π-stacking direction. Samples were prepared by attaching gold wires (15 μm diameter) of a single crystal with a conductive carbon paste. The $\rho$ and conductivity ($\sigma$) values were derived from the following equation: $\rho = \sigma^{-1} = R \times S / L$ ($S$: cross-section, $L$: length). The typical size of samples was $\sim 500 \times \sim 40 \times \sim 20$ μm³.

**ESR measurements.** The X-band ($\sim 9.4$ GHz) continuous wave ESR experiments were performed on single-crystal $2O-F_4$, $2O-F_2$, $2S-F_4$, and $2S-F_2$ (Fig. 8d, e and Supplementary Figs. 37−39). The ESR signals were measured by setting the long axis of the crystal (i.e., the stacking $a$-axis) along the direction parallel to the magnetic field (i.e., $\varphi \sim 0°$ or 180°). The typical size of samples: $\sim 500 \times \sim 40 \times \sim 20$ μm³.

**SQUID measurements.** The static magnetic susceptibility of polycrystals $2O-F_4$ and $2S-F_4$ was measured upon cooling from 300 or 320 K to 2 K by <1 K min⁻¹ applying the static magnetic field of 10,000 O$_e$ (Supplementary Fig. 40). The absence of ferromagnetic impurity in the synthesized samples was confirmed from the obtained $M-H$ curve at 2 K around −55,000 to 55,000 O$_e$, which follows a typical Brillouin function. The obtained magnetic susceptibilities ($\chi_{exp}$) were plotted after subtracting the contribution of the Curie impurity for $S = 1/2$ ($\chi_{cw}$; 0.3% for $2O-F_4$ and 0.2% for $2S-F_4$) and the contribution from core diamagnetism of $2O-F_4$ and $2S-F_4$ ($-3.14 \times 10^{-4}$ and $-3.74 \times 10^{-4}$ emu mol⁻¹, respectively) estimated from Pascal's law[68].

## Data availability
The crystallographic data (CIF files) for the structures reported in this Article have been deposited with the Cambridge Crystallographic Data Centre (CCDC), under deposition numbers 2264341 (**2S**), 2264342 (**2S•BF₄**), 2264325 ($2O-F_4$), 2264326 ($2O-F_2$), 2264327 ($2S-F_4$), and 2264331 ($2S-F_2$). These data can be obtained free of charge via www.ccdc.cam.ac.uk/data_request/cif, or by emailing data_request@ccdc.cam.ac.uk, or by contacting The Cambridge Crystallographic Data Centre, 12 Union Road, Cambridge CB2 1EZ, UK; fax: +44 1223 336033. All experimental data within the article and its Supplementary Information are available from the corresponding authors upon request.

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

## Acknowledgements

Professor Taisuke Ozaki (The University of Tokyo), Professor Takehiko Mori (Tokyo Institute of Technology), Dr. Yusuke Tsutsui (Kyoto University), and Dr. Kazuma Nakamura (Kyushu Institute of Technology) for the helpful discussions related to theoretical calculations; Prof. Hidefumi Akiyama and Dr. Keisei Shibata (The University of Tokyo) for Raman spectroscopy; Ms. Yuka Ito (The University of Tokyo) for technical support in electrical conductivity measurements; Idemitsu Kosan Co., Ltd. for their support throughout the study. We also thank the Supercomputer Center, the Institute for Solid State Physics, and the University of Tokyo for providing their facilities for use. The computation for simulating EMV coupling was performed using Research Center for Computational Science, Okazaki, Japan (Project No. 23-IMS-C276). The synchrotron radiation experiments were performed at SPring-8 with the approval of the Japan Synchrotron Radiation Research Institute (JASRI) (Proposal No. 2023B0304). This work was partially supported by JST PRESTO (JPMJPR22Q8 to T.F.), JSPS Grants-in-Aid for Scientific Research (No. JP20H05206, JP21K05018, JP22H04523 to T.F., JP18H05225, JP21K18597, JP22H00106 to H.M., JP20K15240 to S.D., JP21H04988 to H.O., JP21H01041, JP22K03526 to K.Yoshimi), MEXT Grants-in-Aid for Scientific Research on Innovative Areas "Hydrogenomics" (JP18H05516 to H.M.), a research grant from Iketani Sci. Technol. Foundation, the Naito Foundation, the Kao Foundation for Arts and Science to T.F., and the Noguchi Institute to S.D. Part of this work was conducted at the Institute for Molecular Science, supported by Advanced Research Infrastructure for Materials and Nanotechnology (JPMX1222MS1002) of the Ministry of Education, Culture, Sports, Science and Technology (MEXT), Japan.

## Author contributions

T.F. conceived the project and H.M. directed it. T.F., R.K., K.O., and K.M. performed the synthesis procedures. T.F. and S.D performed electrical conductivity and dielectric constant measurements. T.F., T.M., Z.G., and H.O. performed the optical measurements. T.F., K.O., and T.N. performed magnetic measurements. T.F. and K.Yoshimi performed the theoretical calculations for the band structures and $U_{eff}$. T.F. and H.Sato. performed in-house single-crystal XRD analyses; S.K., T.A., H.Sawa, and Y.N. performed the synchrotron single-crystal XRD analyses. K.Yamamoto., A.T., and T.F. performed simulation for EMV coupling. T.F. wrote the manuscript, and all authors discussed the results and commented on the manuscript.

## Competing interests

The authors declare no competing interests.
