## [Peer Review File · Nature Communications]

Orbital hybridization of donor and acceptor to enhance the conductivity of mixed-stack complexesREVIEWER COMMENTS

Reviewer #1 (Remarks to the Author):

The paper reports the synthesis of two new electron-donor molecules, oligo(3,4-ethylenedioxythiophene (2O) and its sulfur analog (2S), and the preparation of several CT crystals with TCNQ and its F-substitutes analogues. The crystals, exhibiting a mixed stack 1:1 packing, were characterized by a series of physical measurements and computations. They display appreciable room temperature conductivity - that the authors consider the main result, as stressed by the the title itself.

I would say that I remained somewhat disappointed by the paper, starting from the title, where the "topological" word is improperly used and where "highly conducting" is misleading as the conductivity can be considered high for the class of compounds considered, i.e., mixed stack CT crystals. Probably the authors think this kind of hype necessary to publish in the Nature portfolio journals. In fact, the paper contains many sentences in this direction, or obscure sentences of the type "topologically fused" or "highest degree of structural perturbation (p. 14).

All this is probably minor, and could be easily corrected, if the editors share my opinion, and encourage them to do so. As a matter of fact, the paper contains a lot of experimental data, but I think their interpretation is for the most part incorrect, mainly because, despite the high number of reference that the authors have supposedly read, they appear to be not aware of the lot of work, both experimental and theoretical, that has been made since the discovery of neutral-ionic transition back in the eighties. I cannot list all of the problems or lacking references, that would amount to rewrite the paper for the authors, I just state well established points concerning the physics of mixed stack CT complexes.

- a) The many physical properties connected the the degree of charge transfer have a maximum enhancement around the NI boundary, roughly following the equation $\delta(1-\delta)$, and this is true also for e-mv coupling AND for the coupling to the lattice phonons yielding to the Peierls transition (dimerization).
- b) As a matter of fact, all the CT crystals near the NI boundary appear to have a dimerized structure, even if the X-ray structure indicate the contrary: The absence/presence of inversion center is very difficult to ascertain by X-ray, see for instance these two recent papers: Crystals 2018, 8, 158; J. Phys. Chem. C 2021, 125, 25816.
- c) Infrared spectroscopy is much more sensitive than X-ray to the absence of inversion center, thanks to the presence of strong e-mv induced bands with polarization along the stack: As shown by Fig. 5 and Fig. S25, all the CT crystals, except perhaps 2S-F2 are dimerized at room temperature. All the CT crystals then are band semiconductors, and the band structure calculations, that are based on the average structure containing the inversion center, cannot reflect this. There is no need to invoke U_{eff} (page 12 and following) or in any case it has a minor role.
- d) The calculation and "assignment" of the IR spectrum reported in Section S7 and Table S5 are completely meaningless. The authors should consult an expert of vibrational spectroscopy before embarking in such tasks. The computer can only give wrong answers to wrong questions. I don't know from where the symmetry labeling reported in Column 4 of Table 5 is coming from, I would not have allowed one of my student to pass the exam on this basis.
- e) I do not agree with the interpretation of the phase transition of 2F-S4. The uncertainty in the values of δ obtained from X-ray do not allow to establish such an increase of ionicity, nor this fact justifies the small increase (if any... the authors should use an internal standard and measure the areas to confirm the intensity increase of the e-mv bands if Fig. 5b). I believe the transtion is more likely a disorder to order.

In summary, I do not think the paper is suitable for publication in Nature Communications. After the authors have addressed the points above, the paper can be considered for another journal of the Nature portfolio, like Nature Chemistry (after all, the major achievement of the paper is the synthesis of new strong electron donors)

Reviewer #2 (Remarks to the Author):

This work by Fujino, Mori et al. deals with molecular charge-transfer complexes with alternated stacks, usually found in an ionic (I) or neutral (N) states, with in both cases low conductivity, at variance with the very rare examples reported here where mixed-stacks complexes with a charge transfer close to the N-I boundary favors a high conductivity. The main originality of the work is therefore to be found in (i) the ability to engineer CT salts at the N-I boundary by adapting the redox potentials of both partners, (ii) the adaptation of HOMO and LUMO symmetry to favor the best possible overlap within the stacks, using original bis(thiophene) derivatives whose HOMO symmetry differs from that of classical TTFs. These combined approaches allow for the isolation of conducting salts with RT conductivity one order of magnitude higher than those found earlier (and not several orders or magnitude as stated page 13). Another interesting point is that the most conducting material exhibits the largest band gap, a consequence of increased n-dimerization tendency, an observation that perhaps limits the approach toward even more conducting (or metallic) materials.

Besides, the experimental work is complete and the analyses of the data are of excellent quality. The methodology is sound, the experiments well described to be reproduced and the supplementary material very helpful.

Altogether, I recommend publication in Nature Comm. Such materials at the N-I boundary are really hard to find (and only really identified under pressure) and their high conductivity, high solubility and air stability provide opportunities for both fundamental new physics and interesting application in electronics.

As minor modifications, I suggest

- 1) It should be specified (top of page 6) that the synthesis of 2S reported here complements another route to 2S described by the same authors in a recent patent (WO2020262443)
- 2) Page 6, lines 12 and following. The text should detail and give references to the concept of pitch for the node pattern of the frontier orbitals.
- 3) Correct the over-exaggerated statement page 13 last line
- 4) Figure 4 is really hard to read. I suggest that the authors adopt for Fig 4b,c,e,f the same color code used in fig4d,g, i.e. the $\delta=0$ and $\delta=1$ text in black as the related points. In the fig 4 caption, it should be specified that full lines are for $\delta=1$ and dotted lines for $\delta=0$
- 5) In ref 26 correct spelling is: Torrance

Reviewer #3 (Remarks to the Author):

In this work the authors extend their previous investigations on the conducting crystalline materials based on 2,2'-bi(3,4-ethylenedithiophene) and 2,2'-bi(3,4-ethylenedioxythiophene) capped here with methylthio groups, i.e. donors 2S and 2O, respectively. Here they obtained crystalline charge transfer complexes of both donors with the acceptors TCNQ-F4 and TCNQ-F2. Single crystal conductivity measurements of these charge transfer materials, where the donors and acceptors are alternated within 1D stacks, show semiconducting behaviour with relatively high conductivity values (10^{-3} – 0.1 S cm $^{-1}$) for such type of alternated D-A compounds. State-of-the-art characterizations backed up with band structure and DFT calculations of the energy levels of donors and acceptors have been performed, confirming the good match between the electron donor and electron acceptor abilities of the partners, at the threshold of neutral-ionic boundary. The paper reads nicely and the results are of interest for the community. However, in spite of the use of the term "Topological fusion of donor and acceptor" in the title, which is completely misleading as one could think that the donors and the acceptors are chemically fused, while the authors refer to a mixing of the HOMO of the donor and the LUMO of the acceptor in the CT complex, in my opinion there is no striking novelty in the manuscript which could qualify it for the Nature journals portfolio. The conductivity of the CT compounds is certainly higher compared to

other D-A alternated materials but still remains activated. I think some modulation of the conducting properties by light irradiation could maybe add interest to the paper.

Some revisions to be taken into account for a submission in a more specialized journal:

1) The Abstract should be more specific, with the description of the donor and acceptor molecules involved in the study.

2) How are estimated the pitch values in Fig. 3b?

Reviewer #4 (Remarks to the Author):

This is a very relevant and significant contribution by H. Mori and co-workers, reporting a cleverly designed study of a series of charge transfer salts near the border of the neutral to ionic transition. The results clearly show for the first time that near the border of this transition, in spite of the mixed stacking arrangement of donor and acceptor molecules, high electrical conductivity can be achieved due to a favourable combination (topological fusion) of donor and acceptor molecular orbitals.

The class of molecular materials displaying neutral to ionic transitions have been since the early days of molecular conducting materials, key compounds for understanding fundamental aspects of the electronic properties of molecular materials, attracting the attention of a wide scientific community in solid state physics and molecular materials science. These results of this study provide not only new compounds with a breaking record of electrical conductivity among this type of salts, but also pave the way to a new route to prepare highly conducting materials based on neutral species.

This study is well designed, combining a rational choice of new molecular units/compounds with a comprehensive physical characterisation of electronic and magnetic properties complemented by theoretical electronic structure quantum calculations. The main conclusions are overall well supported by the experimental results and theoretical calculations using state of the art techniques.

In spite of not being a native English speaker I feel that the manuscript would benefit from a throughout revision of the English style. Some points of the phrasing are not entirely clear.

Any way in view of the relevance and significance of the results I consider this work as certainly deserving publication in Nature Communications after some minor revisions and authors addressing the following secondary aspects.

One point that does not becomes clear in the discussion is the role of disorder in different compounds. Some of the structures present variable degrees of disorder that should not be neglected, as in 2S-F2 where there is Fluorine occupational disorder. Possible disorder effects should be taken into account when comparing with other compounds. As mentioned in Fig S13 calculations were done considering only the geometry of the largest occupancy. The authors should make more clear and discuss here possible effects of disorder and for instance in the theoretical calculations consider possible differences for other geometries.

Another point to be taken into account are the values for electrical conductivity that were obtained either by 4-probe or only 2-probe techniques. The 2 probe values underestimate the intrinsic conductivity. This seems ignored in comparing the different compounds (first line of page 14 of the manuscript) although it probably does not significantly change conclusions.

The following are some minor points that authors should also address:

Page 2 Abstract, first line : "Mixed-stack complexes which comprise alternating layers of donors and acceptors are ..." . Here layers should be omitted and instead it is suggested "Mixed-stack complexes which comprise columns of alternating donors and acceptor molecules are ...".

Page 3, line 14, where it reads "... , although they are more frequently constructed, possibly owing to the Madelung energy gain between charged donor and acceptor in the intermediates.". Here the word "intermediate" has no clear meaning.

Page 5, line 5, where it reads "...fulfills the two requirements for electronic structures" Here the requirements are not clear. Please specify. ,

Page 7, lines 13-14, please define angles theta 1 and theta 2

Page 16, line 4, wher it reads "... helped us to address the underlying mechanism" Here which mechanism is not clear. Please specify.

RESPONSE TO REVIEWERS' COMMENTS

Our reply to the Comments from Reviewer #1

We sincerely appreciate the favorable comments and important suggestions. The comments are shown in blue, and our responses are shown in black. The revised portions in the text and Supplementary information are highlighted in yellow.

The paper reports the synthesis of two new electron-donor molecules, oligo(3,4-ethylenedioxythiophene (2O) and its sulfur analog (2S), and the preparation of several CT crystals with TCNQ and its F-substitutes analogues. The crystals, exhibiting a mixed stack 1:1 packing, were characterized by a series of physical measurements and computations. They display appreciable room temperature conductivity - that the authors consider the main result, as stressed by the title itself.

I would say that I remained somewhat disappointed by the paper, starting from the title, where the "topological" word is improperly used and where "highly conducting" is misleading as the conductivity can be considered high for the class of compounds considered, i.e., mixed stack CT crystals. Probably the authors think this kind of hype necessary to publish in the Nature portfolio journals. In fact, the paper contains many sentences in this direction, or obscure sentences of the type "topologically fused" or "highest degree of structural perturbation (p. 14).

We thank this reviewer's valuable comments. As per this reviewer's suggestions, we corrected the inappropriate words as follows.

(Title in text)

Orbital hybridization ~~Topological fusion~~ of donor and acceptor to enhance the conductivity of ~~form highly conducting~~ mixed-stack complexes

(Abstract in text)

Surprisingly, the orbitals were **highly hybridized ~~topologically fused~~** in the single-crystal complexes, **~~endowing them with high~~ enhancing the room-temperature conductivity (10^{-3} – 0.1 S cm $^{-1}$) of mixed-stack complexes.**

(p. 5 in text)

In this study, we designed and synthesized **highly ~~conducting~~** mixed-stack complexes uniquely located at and near the N–I boundary, **enhancing the conductivities of mixed-stack complexes.**

(p. 6 in text)

Surprisingly, the similar donor HOMO and acceptor LUMO with comparable energy levels and well-matched orbital symmetries were **highly hybridized ~~topologically fused~~** in the complexes, **enhancing ~~endowing~~ the σ_{rt} values of mixed-stack complexes with top-class σ_{rt} (10^{-3} – 0.1 S cm $^{-1}$).** The highest σ_{rt} value (0.1 S cm $^{-1}$) observed for **2S–F $_4$ among the combinations of donors 2X and acceptors F $_n$,** which is located at the N–I boundary is the highest value reported for a structurally defined 1D mixed-stack complex under ambient pressure.

(p. 12 in text)

Notably, we have made a significant discovery using OpenMX calculations,^{38,39} which identified that the HOMO of the donor and LUMO of the acceptor are **highly hybridized ~~topologically fused for the first time.~~**

(p. 13 in text)

Given that the strong hybridization between the donor HOMO and acceptor LUMO **to form ~~topologically fused orbitals~~** in the band structures with nearly negligible E_g , ...

(p. 17 in text)

..., 2S-F4 has the highest degree of structural perturbation at the N-I boundary, showed largest increase in ρ , and highest E_a among the three complexes.

(p. 21 in text)

These complexes have highly hybridized topologically fused orbitals between the donor and acceptor and exhibited high σ_{π} (10^{-3} to 0.1 S cm^{-1}) under ambient conditions; ...

(p. 50 in text; Caption for Figure 3d)

d, Highly hybridized Topologically fused HOCO and LUCO between donor and acceptor at the Γ -point calculated by OpenMX.^{38,39}

All this is probably minor, and could be easily corrected, if the editors share my opinion, and encourage them to do so. As a matter of fact, the paper contains a lot of experimental data, but I think their interpretation is for the most part incorrect, mainly because, despite the high number of reference that the authors have supposedly read, they appear to be not aware of the lot of work, both experimental and theoretical, that has been made since the discovery of neutral-ionic transition back in the eighties. I cannot list all of the problems or lacking references, that would amount to rewrite the paper for the authors, I just state well established points concerning the physics of mixed stack CT complexes.

a) The many physical properties connected the the degree of charge transfer have a maximum enhancement around the NI boundary, roughly following the equation $\delta(1-\delta)$, and this is true also for e-mv coupling AND for the coupling to the lattice phonons yielding to the Peierls transition (dimerization).

b) As a matter of fact, all the CT crystals near the NI boundary appear to have a dimerized structure, even if the X-ray structure indicate the contrary: The absence/presence of inversion center is very difficult to ascertain by X-ray, see for instance these two recent papers: Crystals 2018, 8, 158; J. Phys. Chem. C 2021, 125, 25816.

c) Infrared spectroscopy is much more sensitive than X-ray to the absence of inversion center, thanks to the presence of strong e-mv induced bands with polarization along the stack: As shown by Fig. 5 and Fig. S25, all the CT crystals, except perhaps 2S-F2 are dimerized at room temperature. All the CT crystals then are band semiconductors, and the band structure calculations, that are based on the average structure containing the inversion center, cannot reflect this. There is no need to invoke U_{eff} (page 12 and following) or in any case it has a minor role.

d) The calculation and "assignment" of the IR spectrum reported in Section S7 and Table S5 are completely meaningless. The authors should consult an expert of vibrational spectroscopy before embarking in such tasks. The computer can only give wrong answers to wrong questions. I don't know from where the symmetry labeling reported in Column 4 of Table 5 is coming from, I would not have allowed one of my student to pass the exam on this basis.

e) I do not agree with the interpretation of the phase transition of 2F-S4. The uncertainty in the values of δ obtained from X-ray do not allow to establish such an increase of ionicity, nor this fact justifies the small increase (if any... the authors should use an internal standard and measure the areas to confirm the intensity increase of the e-mv bands if Fig. 5b). I believe the transtion is more likely a disorder to order.

In summary, I do not think the paper is suitable for publication in Nature Communications. After the authors have addressed the points above, the paper can be considered for another journal of the Nature portfolio, like Nature Chemistry (after all, the major achievement of the paper is the synthesis of new strong electron donors)

We would like to express our gratitude for the professional comments provided by the reviewer. Initially, we thought that 2S-F₄ exhibited the π -dimerization perturbation at room temperature and even more so at low temperatures, without actually inducing structural π -dimerization. However, we took into account this reviewer's comments and decided to conduct further experiments and calculations. Our new XRD data indicate that 2S-F₄ displayed structural fluctuation at a high temperature of 300 K, while it may show a π -dimerized structure at 200 K. These data led us to consider that the high-temperature phase of 2S-F₄ with π -dimerization fluctuation may be regarded as a disordering state, and the disorder-order-like transition occurs at 282 K accompanied by structural changes, as suggested by this reviewer. The detailed results are shown below.

1) We additionally performed the synchrotron radiation experiments at 300 K and observed significant X-ray diffuse scattering (XDS) along the c^* -axis (Supplementary Fig. 14). This XDS may be induced by the π -dimerization fluctuation between donor and acceptor along the a -axis (i.e., the π -stacking direction) in columns, possibly resulting in the intercolumnar interactions to fluctuate the molecular arrangement along the c -axis. The scattering may be a precursor phenomenon for the one-dimensional nature as identified for TTF-CA (Buron-Le Cointe, M., Lemée-Cailleau, M. H., Cailleau, H., Ravy, S., Bézar, J. F., Rouzière, S., Elkaïm, E. & Collet, E. *Phys. Rev. Lett.* **96**, 205503 (2006)). The possible π -dimerization fluctuation is consistent with the observation of the electron-molecular vibration (EMV)

coupling-based signals in the polarized infrared reflectivity spectra at room temperature, as discussed later.

2) Upon cooling to 200 K, the XDS along the c^* -axis disappeared, possibly due to the π -dimerization. Considering the magnetic characteristics that showed paramagnetic to non-magnetic transition at 282 K, this disappearance may be related to spin-Peierls-like π -dimerization. The XDS disappearance supports the XDS observed at 300 K is a precursor phenomenon prior to π -dimerization. Furthermore, the XRD at 200 K showed a superlattice with dimensions of $a \times 2b \times 2c$. The superlattice peaks are observed only at $k + l = 2n$ in the $a \times 2b \times 2c$ lattice. These peaks correspond to the reflection conditions of the A-faced center. The observation enabled us to propose a possible dimer model for the molecular arrangement, as shown in Supplementary Fig. 17.

3) We additionally performed the Raman spectroscopy of single-crystal **2S-F₄** at room temperature. The Raman spectrum and the optical conductivity spectrum along the π -stacking direction, which was derived from the polarized reflectivity spectrum, likely exhibited the same modes. Considering the inversion center of the single-crystal structure in a $P2_1/n$ space group with uniform π -stacked donors and acceptors along the $C2$ glide symmetry, the Raman-active modes should not be visible in IR, while the IR-active modes should not be visible in Raman. The coincidence of the shapes of the observed IR and Raman spectra indicates the dynamic fluctuating of donor–acceptor- π -stacking dimerization. The optical activation of the a_g mode along the π -stacking direction may be ascribed to the EMV couplings.

4) We conducted a computational study to confirm that the EMV coupling effect is responsible for these signals. We performed numerical simulations for the optical activation of a_g modes based on the donor–acceptor dimer model proposed by Painelli, A. and Girlando, A. (*J. Chem. Phys.* **84**, 5655–5671 (1986)). The simulation well explained the optical activation of the a_g mode due to EMV coupling, implying that the one-dimensional molecular stacking is not uniform, contradicting the uniform columnar structure predicted by X-ray structural analysis. The activation of the EMV modes hints at dynamic deviations from the average structure within the molecular columns, supporting the dynamic π -dimerization fluctuation.

Based on these data, we revised the text and Supplementary Information. The structural information for the newly obtained XRD single-crystal structures mixed-stack complexes, in which **2O-F₂** showed a positional disorder, and the calculation data based on the structures in the text and Supplementary

Information were updated as follows.

(p. 8–9 in text)

The fluorine atoms in **2O–F₂** and **2S–F₂** exhibited positional disorder, with an occupancy of 94:6 and 86:14, respectively (Supplementary Fig. 9b,d). The donors and acceptors are uniformly and alternately π -stacked in columns along the *a*-axis. The donor–acceptor interplanar distances depend on the molecular size of the donor (i.e., **2O** < **2S**) and acceptor (i.e., **F₂** < **F₄**); the distances increased in the order **2O–F₂** (3.32938 Å) < **2O–F₄** (3.36158 Å) < **2S–F₂** (3.39887 Å) < **2S–F₄** (3.406 Å) (Table 1). Significant intracolumnar short contacts were not observed, suggesting the existence of 1D electronic structures favorable for strong intermolecular interactions. ~~The fluorine atoms in **2S–F₂** exhibited positional disorder, with an occupancy of 86:14 (Supplementary Fig. 9d,e).~~ Notably, X-ray diffuse scattering was observed, reminiscent of structural fluctuation (Supplementary Note 6, Supplementary Fig. 14).

(p. 10 in text)

The δ values of **2O–F₄**, **2O–F₂**, **2S–F₄**, and **2S–F₂** were determined to be 0.7984(23), 0.713(43), 0.693(23), and 0.462(32), respectively (Table 1).

(p. 10–11 in text)

The band structures had small or negligible energy gaps (E_g) in proximity to the Fermi level (0.02–0.054 eV for **2O–F₄** and **2S–F₂** and <0.01 eV for **2O–F₂** and **2S–F₄**; ...

(p. 11 in text)

The density of states (DOS) was calculated to determine the bandwidth (W) of the complexes in the major occupancies, the values of which were 0.864 eV (**2O–F₄**), 0.898 eV (**2O–F₂**), 0.887 eV (**2S–F₄**), and 0.8994 eV (**2S–F₂**) (Table 1 and Supplementary Fig. 173), ...

We next quantified the intracolumnar transfer integrals (t_{DA}) between a donor and the nearest six neighboring molecules in the single-crystal structures...

These calculations also identified prominent intracolumnar donor–acceptor interactions ($t_{DA} \approx 0.21$ eV in Table 1; t_1 and t_4 in Supplementary Table 3 and Supplementary Fig. 27), ...

The W and intracolumnar t_{DA} values follow nearly consistent trends: F_4 complexes < F_2 complexes, ...

(p. 12–13 in text)

The shape of this spectrum reflects the neutral-to-N-I boundary state⁴³ ($\delta = 0.46(3)0.42(2)$) ...

(p. 14–15 in text)

It is also notable that the polarized infrared reflectivity spectra of the complexes along the π -stacking direction exhibited multiple sharp peaks in the low-energy region of 0.14–0.19 eV, except for nearly neutral $2S-F_2$ (Fig. 5b, and Supplementary Figs. 34, and 35a 25). These peaks indicate electron molecular vibration (EMV) coupling,⁴⁹ based on the broken symmetry along the π -stacking direction, which is possibly triggered by donor-acceptor π -dimerization^{43,44} supported by the DFT phonon calculations. These signals are intrinsically infrared-silent based on the uniformly π -stacked donors and acceptors along the $C2$ glide symmetry direction (Supplementary Figs. 26, 27, Supplementary Tables 5, and 13–16) apparent in the single-crystal XRD pattern (Supplementary Figs. 9, 10, and Supplementary Table 2). The appearance of the EMV-coupling-based signals indicates that spatiotemporal fluctuation could occur without inducing drastic structural changes detectable by XRD. The optical conductivity spectrum of $2S-F_4$ (Supplementary Fig. 35b) derived from the infrared spectrum showed a coincidence of the shapes to the Raman spectrum at 293K (Supplementary Fig. 35d), with varying intensities. Considering the inversion center of the single-crystal structure in a $P2_1/n$ space group with uniform π -stacked donors and acceptors along the $C2$ glide symmetry, the Raman-active modes should not be visible in IR, while the IR-active modes should not be visible in Raman. Considering the inversion center of the single-crystal structure in a $P2_1/n$ space group with uniform π -stacked donors and acceptors along the $C2$ glide symmetry, the Raman-active modes should not be visible in IR, while the IR-active modes should not be visible in Raman. The coincidence of the spectral shapes indicates the dynamic fluctuating of donor-acceptor- π -stacking dimerization based on the electron-molecular vibration (EMV) couplings.^{43,44} We performed a computational study by numerical simulations using the donor-acceptor dimer model,⁴⁹ confirming that the EMV coupling effect is responsible for these signals (Supplementary Note 8, Supplementary Fig 35c, Supplementary Tables 4, and 5). These results suggest that there is inhomogeneity in the one-dimensional molecular stacking, contradicting the uniform columnar structure

predicted by single-crystal XRD (Supplementary Table 2 and Supplementary Fig. 10). This inhomogeneity likely indicates dynamic fluctuations from the average structure, which is supported by the X-ray diffuse scattering observed in XRD at 300 K (Supplementary Fig. 14).

(p. 16 in text)

Except for the nearly neutral $2S-F_2$, the trend of σ_{it} ($2S-F_4 > 2O-F_2 > 2S-F_2 \approx 2O-F_4$) indicates that the closeness of these complexes to the N-I boundary ...

(p. 16–17 in text)

The trend is explained by the proximity of δ to the N-I boundary which induces degree of structural perturbation based on the π -dimerization fluctuation between the donor and acceptor, as is evident from the EMV coupling-based signals in the reflectivity spectra (Fig. 5b and Supplementary Fig. 25).

(p. 17–18 in text)

... E_a from 0.200(1) eV at high temperatures (288–340 K) to 0.277378(34) eV at low temperatures (228–273 K) without significant hysteresis upon cooling and heating (Fig. 6, 7a, and Table 1). The XRD pattern results in $P2_1/n$ space group at 300 K showed significant X-ray diffuse scattering (Supplementary Fig. 14), possibly due to the precursor phenomenon for the π -dimerization, which may result in EMV coupling-based signals at room temperature. On the other hand, the scattering disappeared (Supplementary Figs. 15 and 16) at 200 K, and the XRD low temperatures showed superlattice patterns with dimensions of $a \times 2b \times 2c$ (Supplementary Figs. 13 and 17), the complex maintained the $P2_1/n$ space group without possibly implying the π -dimerization along the a -axis (Supplementary Note 6) (Supplementary Table 3). However, the bond length analyses of the donor and acceptor in the complex indicate that δ increases upon cooling (namely from "N-I boundary" to "more ionic" states; Supplementary Fig. 12), which were estimated to be 0.63(3) at 293 K, 0.76(3) at 240 K, 0.81(4) at 200 K, and 0.76(3) at 173 K from Kistenmacher's relationship³⁷ for the structures of acceptors. The increase in δ suggested the strengthened degree of π -dimerization,^{43,44} reflecting the structural fluctuations evident in the EMV coupling based signals in the reflectivity at room temperature (Fig. 5b and Supplementary Figs. 25–27). The broken symmetry upon the possible strengthened degree of π -dimerization of the complex

may induce was experimentally identified by the markedly intensified

This indicates that the electronic structures changed at 282 K without any major impact on the structures...

(p. 19 in text)

..., suggesting a spin-Peierls-like singlet-triplet transition based on the 1D electronic structure⁵¹ during the strengthening of the degree of π -dimerization.

(p. 20–21 in text)

These transitions were triggered by lattice π dimerization processes with hysteresis loops upon heating and cooling based on their first order nature.⁵² In contrast, The high T_c (282 K) and high σ_{rt} (0.1 S cm^{-1}) even at ambient pressures of 2S-F₄, for which δ is exactly at the N-I boundary ($\delta = 0.693(23)$), exhibited a transition at T_c of 282 K not in the lattice but in the electronic structure, leading to with non-hysterical changes in their electronic functionalities such as electrical conductivity, reflectivity, dielectricity, and magnetism. This transition may involve the unique electronic perturbation structure at the N-I boundary between the electron-itinerant and localized states, resulting in the high T_c (282 K) and high σ_{rt} even at ambient pressures.

... 2S-F₄ at $\delta = 0.693(23)$ is the highest value ..., reminiscent of the π -dimerization fluctuation strengthening that spatiotemporally breaks the symmetry ...

(p. 22 in text)

Single-crystal XRD measurements were performed using a Rigaku MercuryII CCD X-ray diffractometer (Mo K_{α} , $\lambda = 0.71073 \text{ \AA}$) equipped with a RIGAKU GN2-TS600 temperature controller and BL02B1 ($\lambda = 0.30960 \text{ \AA}$) at a synchrotron facility SPring-8 in Japan. A N₂-gas-blowing device was employed for the low-temperature measurements.

(p. 23 in text)

Raman spectroscopy was performed on a RENISHAW inVia Reflex.

(p. 29 in text)

... and perpendicular to the directions (Figs. 5, 7b, Supplementary Figs. 34, and 35a,b).

Raman spectroscopy of 2S-F₄ was performed using an excitation wavelength of 532 nm (Supplementary Fig. 35d).

(p. 31 in text)

Materials, synthesis procedures, characterization data, crystallographic data, CV spectra, polarized reflection spectra, Raman spectrum, electrical conductivity data, ...

(p. 41–42 in text):

Reference 49: Painelli, A. & Girlando, A. Electron–molecular vibration (e–mv) coupling in charge-transfer compounds and its consequences on the optical spectra: A theoretical framework. *J. Chem. Phys.* **84**, 5655–5671 (1986). Rice, M. J. Organic linear conductors as systems for the study of electron–phonon interactions in the organic solid state. *Phys. Rev. Lett.* **37**, 36–39 (1976).

Fig. 1 | Structures and physical properties of mixed-stack complexes. ... The δ values for

$2X-F_n$ are shown with error bars (s.d.).

Fig. 2 | Synthesis of mixed-stack complexes $2X-F_n$.

Fig. 3 | Electronic structure of donor 2S, acceptor F_4 , and the mixed-stack complex 2S– F_4 according to theoretical calculations.

Fig. 4 | Bond length analysis of mixed-stack complexes. a, Bond labels of 2O, 2S, F₄, and

F₂. b,c,d,e,f,g, Comparison of C–C and C–S bond lengths (d_{CC} and d_{CS} , respectively) of donors and acceptors in single crystals of mixed-stack complexes, with error bars (s.d.). As a reference, the bond lengths in single crystals of neutral donors or acceptors (i.e., $\delta = 0$) are shown with dotted lines and $1e^-$ -oxidized donors or $1e^-$ -reduced acceptors (i.e., $\delta = 1$) are shown with solid lines.

Table 1 | Structural information and physical properties of mixed-stack complexes at 293 K.

Donor (D)–acceptor (A)	2O–F ₄	2O–F ₂	2S–F ₄	2S–F ₂
Experimental data				
$\Delta E_{\text{REDOX}} (= E_{1/2}^1(\text{D}) - E_{1/2}^1(\text{A}))$ (V) ^a	–0.04	0.20	0.15	0.39
D–A interplanar distance (Å) ^b	3.36158	3.32938	3.406	3.39887 ^c
δ from bond length analyses in A	0.7981(23)	0.713(43)	0.693(23)	0.462(32) ^d
σ at 300 293 K (S cm ^{–1})	$4.91.0 \times 10^{-43}$	$1.41.6 \times 10^{-2}$	0.10	$6.93.2 \times 10^{-3}$
$h\nu_{\text{CT}}$ (eV)	0.83	0.73	0.64	0.50
E_a for high temperature region (eV) ^d	0.113427(1) (290257– 315293 K)	0.178(1) (259–337 K)	0.200(1) (288–340 K)	0.112420(1) (288257– 312303 K)
E_a for low temperature region (eV) ^d	0.215196(24) (238218– 258241 K)	0.2256(4) (219–231 K)	0.277378(34) (228–273 K)	0.090217(178) (221200– 244230 K)
Calculated data				
E_g (eV) ^e	0.054	< 0.01 ^f	< 0.01	0.02 ^c
W (eV) ^e	0.864	0.898 ^c	0.887	0.8991 ^c
t_{DA} (eV) ^f	0.2035	0.2097 ^c	0.208	0.2068 ^c
$U_{\text{eff}}(\text{D})$ (eV) ^f	2.31	2.2830 ^c	1.8990	1.898 ^c
$U_{\text{eff}}(\text{A})$ (eV) ^f	2.523	2.4851 ^c	2.1922	2.223 ^c

^aStructural data with major occupancy were used in the calculations for 2O–F₂ and 2S–F₂.

^bDetermined from ρ – T plots using the Arrhenius equation. ^{d,e}Calculated by OpenMX^{38,39} as a sum of dispersions for the bonding and antibonding bands between donor HOMO and acceptor LUMO (i.e., W_{B} and W_{A} , respectively) and E_g , if present (i.e., $W = W_{\text{B}} + W_{\text{A}} + E_g$).

^eStructural data with major occupancy were used in the calculations for 2O–F₂ and 2S–F₂.

See Supplementary Information for the data with the minor occupancies.

(p. 10 in Supplementary Information)

Supplementary Note 6: Single-crystal XRD measurements

The **in-house** single-crystal X-ray diffractometer (XRD) analyses of a neutral donor **2S** (Supplementary Fig. 7 and Supplementary Table 1) and a charge-transfer salt **2S•BF₄** (Supplementary Fig. 8 and Supplementary Table 1), ~~mixed stack complexes at 293 K~~ (Supplementary Figs. 9–11 and Supplementary Table 2) and low temperatures (for **2S•F₄**; Supplementary Table 3) were performed...

(p. 11 in Supplementary Information)

The synchrotron single-crystal XRD analyses of mixed-stack complexes at 300 K (Supplementary Figs. 9–12, 14, and Supplementary Table 2) and **2S•F₄** at 200 K (Supplementary Figs. 13 and 15) were performed and on BL02B1 (X-ray wavelength $\lambda = 0.30960 \text{ \AA}$) at a synchrotron facility SPring-8 in Japan.¹⁸ A N₂-gas-blowing device was employed for the low-temperature measurements. A two-dimensional detector CdTe PILATUS was used to record the diffraction pattern. The intensities of Bragg reflections were collected by CrysAlisPro program.¹⁹ Intensities of equivalent reflections were averaged, and the structural parameters were refined by using Jana2006.²⁰ Fluorine atoms of **2O•F₂** and **2S•F₂** were positionally disordered with the occupancy of 94:6 and 86:14 (Supplementary Fig. 9b,d,e). The bond lengths analyses are shown in Fig. 4 and Supplementary Fig. 12. In the synchrotron radiation experiments at 300 K, we observed significant X-ray diffuse scattering along the c^* -axis (Supplementary Fig. 14). This scattering can be induced by the π -dimerization fluctuation between donors and acceptors along the a -axis (i.e., the π -stacking direction) in columns, possibly resulting in the intercolumnar interactions to fluctuate the molecular arrangement along the c -axis. The scattering may be a precursor phenomenon for the one-dimensional nature as identified for TTF-CA.²¹ Upon cooling to 200 K, the X-ray diffuse scattering disappeared (Supplementary Figs. 15 and 16), possibly due to the π -dimerization. Considering the magnetic characteristics that showed paramagnetic to non-magnetic transition at 282 K, this disappearance may be related to spin-Peierls-like π -dimerization, similar to those observed in one-dimensional charge-transfer complexes.²² The disappearance supports the X-ray diffuse scattering occurring at 300 K is a precursor phenomenon prior to π -dimerization. The XRD at 200 K showed a superlattice with dimensions of $a \times 2b \times 2c$ (Supplementary Fig. 13). The superlattice peaks are observed only at $k + l = 2n$ in $a \times 2b \times 2c$ lattice. These peaks correspond to the reflection conditions of the

A-faced center. The observation enabled us to propose a possible dimer model for the molecular arrangement, as shown in Supplementary Fig. 17. The models suggest a stripe pattern of the π -dimerization, which should belong to a space group of $P1$ without an inversion center.

(Supplementary References)

18: Sugimoto, K., Ohsumi, H., Aoyagi, S., Nishibori, E., Moriyoshi, C., Kuroiwa, Y., Sawa, H. & Takata, M. Extremely high resolution single crystal diffractometry for orbital resolution using high energy synchrotron radiation at SPring-8. *AIP Conf. Proc.* **1234**, 887–890 (2010).

19: CrysAlisPro, Agilent Technologies Ltd, Yarnton (2014).

20: Petříček, V., Dušek, M. and Palatinus, L. Discontinuous modulation functions and their application for analysis of modulated structures with the computing system JANA2006. *Z. Kristallogr. Cryst. Mater.* **229**, 345–352 (2014).

21: Buron-Le Cointe, M., Lemée-Cailleau, M. H., Cailleau, H., Ravy, S., Bérrar, J. F., Rouzière, S., Elkāim, E. & Collet, E. One-dimensional fluctuating nanodomains in the charge-transfer molecular system TTF-CA. *Phys. Rev. Lett.* **96**, 205503 (2006).

Supplementary Table 2: Crystallographic data for single-crystal mixed-stack complexes.

Compounds	2O-F ₄	2O-F ₂	2S-F ₄	2S-F ₂
Temperature / K	300293	300293	300293	300293
Formula	C ₂₆ H ₁₄ F ₄ N ₄ O ₄ S ₄	C ₂₆ H ₁₆ F ₂ N ₄ O ₄ S ₄	C ₂₆ H ₁₄ F ₄ N ₄ S ₈	C ₂₆ H ₁₆ F ₂ N ₄ S ₈
Formula weight	650.65	614.67	714.89	676.91
Crystal system	monoclinic	monoclinic	monoclinic	monoclinic
Space group	$P2_1/n$ (#14)	$P2_1/n$ (#14)	$P2_1/n$ (#14)	$P2_1/n$ (#14)
$a / \text{Å}$	6.7240(2)	6.6638(9)	6.8223(2)	6.8082(2)
	6.7175(6)	6.6807(7)	6.8231(6)	6.7851(4)

$b / \text{\AA}$	20.9067(7)	20.668(3)	21.9312(7)	22.0498(7)
	20.8822(12)	20.6999(16)	21.8944(13)	22.0104(9)
$c / \text{\AA}$	9.7855(3)	9.5765(13)	9.9232(3)	9.9559(3)
	9.7751(7)	9.5876(8)	9.9263(7)	9.9374(5)
$\alpha / \text{deg.}$	90	90	90	90
$\beta / \text{deg.}$	105.654(8)	104.565(7)	107.541(8)	107.967(8)
	105.644(8)	104.561(9)	107.531(8)	107.831(6)
$\gamma / \text{deg.}$	90	90	90	90
$V / \text{\AA}^3$	1324.59(9)	1276.5(3)	1415.68(10)	1421.69(10)
	1320.42(18)	1283.3(2)	1413.99(19)	1412.79(13)
Z	2	2	2	2
$D_{\text{calc}} / \text{g cm}^{-3}$	1.9529	2.0264	1.6771	1.67
	1.637	1.591	1.679	1.596
R_{int}	0.0572	0.1011	0.0658	0.0646
	0.0402	0.0387	0.0459	0.0234
$R_1 (I > 2.00\sigma(I))$	0.0513	0.0503	0.0343	0.0394
	0.0391	0.0409	0.0459	0.0290
wR_2 (all reflections)	0.0677	0.0742	0.0554	0.0604
	0.0909	0.0919	0.1288	0.0719
GOF	2.47	1.98	2.05	2.00
	1.029	1.023	1.160	1.035
CCDC	2264325	2264326	2264327	2264331

Supplementary Table 3: Temperature dependent crystallographic data for a single crystal 2S-F₄

Compounds	2S-F ₄	2S-F ₄	2S-F ₄	2S-F ₄
Temperature / K	293	240	200	173
Formula	C ₂₆ H ₁₄ F ₄ N ₄ S ₈	C ₂₆ H ₁₄ F ₄ N ₄ S ₈	C ₂₆ H ₁₄ F ₄ N ₄ S ₈	C ₂₆ H ₁₄ F ₄ N ₄ S ₈
Formula weight	714.89	714.89	714.89	714.89
Crystal system	monoclinic	monoclinic	monoclinic	monoclinic
Space group	P2₁/n (#14)	P2₁/n (#14)	P2₁/n (#14)	P2₁/n (#14)
a / Å	6.8231(6)	6.7621(8)	6.7419(11)	6.7239(8)
b / Å	21.8944(13)	21.8061(19)	21.771(2)	21.7495(12)
c / Å	9.9263(7)	9.8834(11)	9.8871(19)	9.8770(10)
α / deg.	90	90	90	90
β / deg.	107.531(8)	107.462(13)	107.358(18)	107.285(12)
γ / deg.	90	90	90	90
V / Å ³	1413.99(19)	1390.2(3)	1385.1(4)	1379.2(2)
Z	2	2	2	2
D _{calc} / g cm ⁻³	1.679	1.708	1.714	1.721
R _{int}	0.0459	0.0511	0.0832	0.0449
R ₁ (I > 2.00σ(I))	0.0459	0.0441	0.0547	0.0402
wR ₂ (all reflections)	0.1288	0.1145	0.1460	0.1054
GOF	1.160	1.075	1.065	1.075
CCDC	2264327	2264328	2264329	2264330

Supplementary Fig. 9: Single-crystal structures of mixed-stack complexes at **300 293** K.

a, 2O-F₄. b, 2O-F₂. c, 2S-F₄. d, e, 2S-F₂. Locationally disordered fluorine atoms in 2O-F₂ and 2S-F₂ were colored in aqua yellowish green (d,e). Other atoms were colored as follows; yellow: sulfur; red: oxygen; gray: carbon; blue: nitrogen; yellowish green aqua: fluorine. Hydrogens were omitted for clarity. ORTEP (50% thermal ellipsoid) and wire drawing for donor and acceptor, respectively (a,b,c,d). Wire and ORTEP (50% thermal ellipsoid) drawing for donor and acceptor, respectively (e).

Supplementary Fig. 10: Symmetry elements in the single-crystal structure of mixed-stack complexes at 300 293 K. a, 2O-F₄. b, 2O-F₂ (major occupancy). c, 2S-F₄. d, 2S-F₂ (major occupancy). The structures are displayed in ORTEP (donors, 50% thermal ellipsoid) and wire drawing for donor and acceptor, respectively. Hydrogens were omitted for clarity.

Atoms were colored as follows; yellow: sulfur; red: oxygen; gray: carbon; blue: nitrogen; yellowish green **aqua**: fluorine. Disordered atoms and hydrogens were omitted for clarity.

Supplementary Fig. 11: Single-crystal structures of donors in mixed-stack complexes at **300 293** K.

Supplementary Fig. 12: Single-crystal X-ray diffraction precession image of the $0 KL$ plane in the reciprocal lattice of $2S-F_4$ at 300 K.

Supplementary Fig. 13: Single-crystal X-ray diffraction precession image of the $0 KL$ plane of $2S-F_4$ at 200 K. (a) The reciprocal lattice. (b) The projection of the reciprocal lattice points to the b^*c^* plane ($-1 \leq k, l \leq 1$, b). Superlattice peaks are observed only at $k + l = 2n$ in the $a \times 2b \times 2c$ lattice, which corresponds to reflection conditions of the A-faced center.

Supplementary Fig. 14: Single-crystal X-ray diffraction precession images of the *H 0 L* (a) and *3 K L* (b) planes in the reciprocal lattice of 2S-F₄ at 300 K. The X-ray diffuse scattering extends along the c^* -axis and b^*+c^* -direction (perpendicular to the a^* axis).

Supplementary Fig. 15: Single-crystal X-ray diffraction precession image of the *H 0 L* plane in the reciprocal lattice of 2S-F₄ at 200 K. No apparent the X-ray diffuse scattering was detected.

Supplementary Fig. 16: Temperature-dependence of single-crystal X-ray diffraction precession. (a) Precession image of 2S-F₄ at 319 K and 200 K. (b) Temperature-dependent intensity of the signals for $(-3, -1/2, 3/2)$. The X-ray diffuse scattering along the b^*+c^* -direction disappeared and the superlattice peaks at $k+l=2n$ appeared at approximately 282 K.

Supplementary Fig. 17: Possible Models proposed for the molecular arrangement of 2S-F₄ superlattice observed at 200 K. a, $x=0$. b, $x=1/2$. The A/D molecules squared in the solid and dashed lines move in the opposite direction along the a -axis, leading to a stripe-

pattern charge-density wave arrangement in the bc plane. The model should belong to the space group of $P1$.

(p. 24–25 in Supplementary Information)

6. Theoretical calculations for band structures

Supplementary Note 7: Band and crystal orbital calculations

Mixed-stack complexes, particularly those at the N–I boundary, may exhibit fluctuation in π -dimerization at room temperatures before undergoing the structural changes. This was suggested by X-ray diffuse scattering that disappeared when cooled to 200 K, as well as EMV coupling (which will be discussed later). As have been analyzed for one-dimensional charge-transfer complexes display uniformly π -stacked single-crystal structures with dynamic fluctuations prior to the spin-Peierls-like π -dimerization,²² we performed theoretical calculations based on the average single-crystal XRD structures. In the calculations, the locational disordering of **2O–F₂** and **2S–F₂** (Supplementary Fig. 9b,d) may have impacts on the electronic structures and physical properties. To get an insight into the possible impacts, we performed the calculations not only for the major but also for the minor occupancies. The calculation conditions are shown in Methods section of the text. The results showed that complexes consistently exhibited half-filled 1D electronic structures (Fig. 3c and Supplementary Fig. 18). The real parts of the HOCO and lowest-unoccupied crystal orbitals (LUCO) at the Γ point (0, 0, 0) were visualized by VESTA²³ (Fig. 3d and Supplementary Fig. 19). The transfer integrals between a donor and the six neighboring molecules (donors and acceptors) were calculated (Supplementary Fig. 27 and Supplementary Table 3) to confirm the dominant intracolumnar interactions (t_1 and t_4).

Supplementary Fig. 183: Band structures of mixed-stack complexes. **a**, **2O-F₄**. **b**, **2O-F₂** (major occupancy). **c**, **2O-F₂** (minor occupancy). **d**, **2S-F₄**. **e**, **2S-F₂** (major occupancy). **f**, **2S-F₂** (minor occupancy). In the calculations for **2S-F₂**, the geometry of the major occupancies was used. Γ (0, 0, 0), X (0.5, 0, 0), Y (0, 0.5, 0), N (0, 0.5, 0.5), Z (0, 0, 0.5), P (-0.5, 0, 0.5), Q (-0.5, 0.5, 0.5). The complexes consistently exhibited half filled 1D electronic structures (Fig. 3c and Supplementary Fig. 13). The real parts of the HOCO and lowest unoccupied crystal orbitals (LUCO) at the Γ point (0, 0, 0) were visualized by VESTA²⁵ (Fig. 3d and Supplementary Fig. 14). The Fermi levels (E_F) are determined by occupying electrons according to the Fermi distribution function.

Supplementary Fig. 194: Crystal orbitals of mixed-stack complexes. **a,b,c,d,e**, The LUCO shapes of **2O-F₄** (**a**), **2O-F₂** (**b**, major occupancy), **2O-F₂** (**c**, minor occupancy), and **2S-F₂** (**d**, major occupancy), and **2S-F₂** (**e**, minor occupancy). **f,g,h,i,j**, The HOCO shapes of **2O-F₄** (**f**), **2O-F₂** (**g**, major occupancy), **2O-F₂** (**h**, minor occupancy), and **2S-F₂** (**i**, major occupancy), and **2S-F₂** (**j**, minor occupancy). In the calculations for **2S-F₂**, the

geometry of the major occupancies was used for the calculation. Orbitals were visualized by VESTA.²⁵¹⁸ Atoms were colored as follows; yellow: sulfur; red: oxygen; gray: carbon; blue: nitrogen; yellowish green-aqua: fluorine.

Supplementary Fig. 205: Wannier interpolation bands (shown in green squares) and band dispersion (shown in black solid lines). a, $2O-F_4$. b, $2O-F_2$ (major occupancy). c, $2O-F_2$ (minor occupancy). d, $2S-F_4$. e, $2S-F_2$ (major occupancy). f, $2S-F_2$ (minor occupancy).

Supplementary Fig. 2116: The maximaly localized Wannier function of 2O–F₄ in a cell. Atoms surrounding the molecules were omitted for clarity. Atoms were colored as follows; yellow: sulfur; red: oxygen; gray: carbon; blue: nitrogen; yellowish green-aqua: fluorine. **a**, 2O (0.5, 0, 0.5). **b**, 2O (0, 0.5, 0). **c**, F₄ (0, 0, 0.5). **d**, F₄ (0.5, 0.5, 0).

Supplementary Fig. 2217: The maximaly localized Wannier function of 2O–F₂ (major occupancy) in a cell. Atoms surrounding the molecules were omitted for clarity. Atoms were colored as follows; yellow: sulfur; red: oxygen; gray: carbon; blue: nitrogen; yellowish green-aqua: fluorine. **a**, 2O (0.5, 0, 0.5). **b**, 2O (0, 0.5, 0). **c**, F₂ (0, 0, 0.5). **d**, F₂ (0.5, 0.5, 0).

Supplementary Fig. 2347: The maximally localized Wannier function of 2O–F₂ (minor occupancy) in a cell. Atoms surrounding the molecules were omitted for clarity. Atoms were colored as follows; yellow: sulfur; red: oxygen; gray: carbon; blue: nitrogen; yellowish green: fluorine. **a**, 2O (0.5, 0, 0.5). **b**, 2O (0, 0.5, 0). **c**, F₂ (0, 0, 0.5). **d**, F₂ (0.5, 0.5, 0).

Supplementary Fig. 2418: The maximally localized Wannier function of 2S–F₄ in a cell. Atoms surrounding the molecules were omitted for clarity. Atoms were colored as follows; yellow: sulfur; red: oxygen; gray: carbon; blue: nitrogen; yellowish green-aqua: fluorine. **a**, 2S (0.5, 0, 0.5). **b**, 2S (0, 0.5, 0). **c**, F₄ (0, 0, 0.5). **d**, F₄ (0.5, 0.5, 0).

Supplementary Fig. 2519: The maximaly localized Wannier function of 2S-F₂ (**major occupancy**) in a cell. Atoms surrounding the molecules were omitted for clarity. Atoms were colored as follows; yellow: sulfur; red: oxygen; gray: carbon; blue: nitrogen; yellowish green aqua: fluorine. **a**, 2S (0.5, 0, 0.5). **b**, 2S (0, 0.5, 0). **c**, F₂ (0, 0, 0.5). **d**, F₂ (0.5, 0.5, 0).

Supplementary Fig. 26: The maximaly localized Wannier function of 2S-F₂ (**minor occupancy**) in a cell. Atoms surrounding the molecules were omitted for clarity. Atoms were colored as follows; yellow: sulfur; red: oxygen; gray: carbon; blue: nitrogen; yellowish green: fluorine. **a**, 2S (0.5, 0, 0.5). **b**, 2S (0, 0.5, 0). **c**, F₂ (0, 0, 0.5). **d**, F₂ (0.5, 0.5, 0).

Supplementary Fig. 270: Labels for t values for mixed-stack complexes. a, 2O-F₄. b, 2O-F₂ (major occupancy). c, 2O-F₂ (minor occupancy). de, 2S-F₄. ed, 2S-F₂ (major

occupancy). **f**, **2S–F2** (minor occupancy). The values were summarized in Supplementary Table 34. **2O**, **2S**, **F4**, and **F2** were colored with a blue, red, green, and light green background, respectively for clarity. Atoms were colored as follows; yellow: sulfur; red: oxygen; gray: carbon; blue: nitrogen; yellowish green aqua: fluorine.

Supplementary Table 34: Transfer integrals for mixed-stack complexes. The intracolumnar values (t_1 and t_4) are shown as t_{DA} in Table 1.

Complex	2O–F4	2O–F2 (major occupancy)	2O–F2 (minor occupancy)	2S–F4	2S–F2 (major occupancy)	2S–F2 (minor occupancy)
t_1 (eV)	0.2035	0.2097	0.197	0.208	0.2068	0.187
t_2 (eV)	-0.004976	-0.0042950	-0.00395	0.01832	-0.018792	0.0182
t_3 (eV)	0.00190117	-0.002290	-0.00200	-0.000301879	0.000517435	0.000759
t_4 (eV)	0.2035	0.2097	0.197	0.208	0.2068	0.187
t_5 (eV)	-0.00495501	-0.0042241	-0.00400	0.01832	-0.018792	0.0182
t_6 (eV)	-0.01210186	0.014137	0.0134	-0.000953276	-0.0025264	0.00190

Supplementary Fig. 281: Effective direct Coulomb interactions in a cell of 2O-F₄. Values of donors (a,b), and acceptors (c,d).

Supplementary Fig. 292: Effective direct Coulomb interactions in a cell of 2O-F₂ in the major occupancy. Values of donors (a,b), and acceptors (c,d).

Supplementary Fig. 30: Effective direct Coulomb interactions in a cell of 2O-F₂ in the minor occupancy. Values of donors (a,b), and acceptors (c,d).

Supplementary Fig. 3123: Effective direct Coulomb interactions in a cell of 2S-F₄. Values of donors (a,b), and acceptors (c,d).

Supplementary Fig. 3224: Effective direct Coulomb interactions in a cell of 2S-F₂ in the major occupancy. Values of donors (a,b), and acceptors (c,d).

Supplementary Fig. 33: Effective direct Coulomb interactions in a cell of 2S-F₂ in the minor occupancy. Values of donors (a,b), and acceptors (c,d).

These modifications led us to revise the subsequent numbers of Figures, Tables, Supplementary Figures, Supplementary Tables, and Supplementary References.

In addition, we replaced Reference 37 in the text to estimate the bond-length analysis as follows.

37. Miyasaka, H., Motokawa, N., Matsunaga, S., Yamashita, M., Sugimoto, K., Mori, T., Toyota, N. & Dunbar, K. R. Control of charge transfer in a series of Ru₂^{II,II}/TCNQ two-dimensional networks by tuning the electron affinity of TCNQ units: a route to synergistic magnetic/conducting materials. *J. Am. Chem. Soc.* **132**, 1532–1544 (2010).

37. Kistenmacher, T. J., Emge, T. J., Bloch, A. N. & Cowan, D. O. Structure of the red, semiconducting form of 4,4',5,5'-tetramethyl $\Delta^{2+2'}$ -bi-1,3-diselenole-7,7,8,8-tetracyano-*p*-quinodimethane. TMTSF-TCNQ. *Acta Crystallogr.* **B38**, 1193–1199 (1982).

For the contributions to the single-crystal structural analyses, we added Dr. Shunsuke Kitou (The Univ. of Tokyo), Prof. Taka-hisa Arima (The Univ. of Tokyo and Rigaku Corp.), Dr. Hiroyasu Sato (Rigaku Corp.), Prof. Hiroshi Sawa (Nagoya University), and Dr. Yuiga Nakamura (JASRI) as the authors. We also updated the "**Acknowledgment**" and "**Authors contributions**" for the revisions as follows.

(p. 31 in text)

Supplementary Information is available as follows. Materials, synthesis procedures, characterization data, crystallographic data, CV spectra, polarized reflection spectra, **Raman spectrum**, electrical conductivity data, ...

(p. 32 in text)

..., **Dr. Yusuke Tsutsui (Kyoto University)**, and Dr. Kazuma Nakamura (Kyushu Institute of Technology) for the helpful discussions related to theoretical calculations; **Prof. Hidefumi Akiyama and Dr. Keisei Shibata (The University of Tokyo)** for Raman spectroscopy; **Dr. Hiroyasu Sato (Rigaku Co., Ltd.)** for the single-crystal structural analyses; **Ms. Yuka Ito (The University of Tokyo)** for technical support in electrical conductivity measurements; ...

The synchrotron radiation experiments were performed at SPring-8 with the approval of the **Japan Synchrotron Radiation Research Institute (JASRI) (Proposal No. 2023B0304)**.

(p. 33 in text)

T.F. and **K.Y.oshimi** performed the theoretical calculations **for the band structures and U_{eff}** . **T.F. and H.Sato** performed in-house single-crystal XRD analyses; **S.K., A.T., H.Sawa, and**

Y.N. performed the synchrotron single-crystal XRD analyses.

c) Infrared spectroscopy is much more sensitive than X-ray to the absence of inversion center, thanks to the presence of strong e-mv induced bands with polarization along the stack: As shown by Fig. 5 and Fig. S25, all the CT crystals, except perhaps 2S-F2 are dimerized at room temperature. All the CT crystals then are band semiconductors, and the band structure calculations, that are based on the average structure containing the inversion center, cannot reflect this. There is no need to invoke U_{eff} (page 12 and following) or in any case it has a minor role.

We thank this reviewer's important comments. The single-crystal XRD of 2S-F₄ at 300 K displayed uniformly and alternately π -stacked donors and acceptors as the average structure. The XRD also showed significant XDS along the c^* -axis, suggestive of dynamical fluctuations prior to the π -dimerization. The XDS disappearance at 200 K may support that the fluctuation is a precursor phenomenon prior to π -dimerization. Such a phenomenon has often been shown in one-dimensional charge-transfer complexes that exhibit spin-Peierls-like dimerization (the examples are shown in reference 2). For theoretical calculations of the mixed-stack complexes in this manuscript (band calculations, crystal orbitals, transfer integrals, and estimations for U_{eff}), we followed the examples of typical 1D complexes and used the average single-crystal structure with dynamical fluctuation prior to the structural transition, as often performed for these 1D complexes. We have provided a detailed explanation for the calculations in the text and Supplementary Information as follows.

(p. 10 in text)

Based on the single-crystal structures of 2X-F_n as the average structures, we then obtained the band structures using first-principles calculations (OpenMX software), ...

(p. 13 in text)

..., we considered the electronic structures of 2X-F_n to be like homogeneous donor- or acceptor-stacked charge-transfer salts, with the average structure.²... by combining first-principles calculations (QE^{40,41}/RESPACK⁴² packages) based on the average single-crystal structures.

(p. 26 in text)

All the periodic DFT calculations were performed with the single-crystal structures of

mixed-stack complexes as the average structures at 293 K. The calculations were performed by the OpenMX software, ...

(p. 24–25 in Supplementary Information)

6. Theoretical calculations for band structures

Supplementary Note 7: Band and crystal orbital calculations

Mixed-stack complexes, particularly those at the N–I boundary, may exhibit fluctuation in π -dimerization at room temperatures before undergoing the structural changes. This was suggested by X-ray diffuse scattering that disappeared when cooled to 200 K, as well as EMV coupling (which will be discussed later). As have been analyzed for one-dimensional charge-transfer complexes display uniformly π -stacked single-crystal structures with dynamic fluctuations prior to the spin-Peierls-like π -dimerization,²² we performed theoretical calculations based on the average single-crystal XRD structures. In the calculations, the locational disordering of **2O**–**F**₂ and **2S**–**F**₂ (Supplementary Fig. 9) may have impacts on the electronic structures and physical properties. To perform theoretical calculations, we used the average structure before the structural transition, similar to the approach used for these 1D complexes. The calculation conditions are shown in Methods section of the text. The results showed that complexes consistently exhibited half-filled 1D electronic structures (Fig. 3c and Supplementary Fig. 18). The real parts of the HOCO and lowest-unoccupied crystal orbitals (LUCO) at the Γ point (0, 0, 0) were visualized by VESTA²⁵ (Fig. 3d and Supplementary Fig. 19). The transfer integrals between a donor and the six neighboring molecules (donors and acceptors) were calculated (Supplementary Fig. 27 and Supplementary Table 3) to confirm the dominant intracolumnar interactions (**t**₁ and **t**₄).

d) The calculation and "assignment" of the IR spectrum reported in Section S7 and Table S5 are completely meaningless. The authors should consult an expert of vibrational spectroscopy before embarking in such tasks. The computer can only give wrong answers to wrong questions. I don't know from where the symmetry labeling reported in Column 4 of Table 5 is coming from, I would not have allowed one of my student to pass the exam on this basis.

We sincerely appreciate this reviewer's professional comments. We consulted experts, Prof. Kaoru Yamamoto (Okayama University of Science) and Prof. Akira Takahashi (Nagoya Institute of

Technology) and collaborated to calculate the EMV coupling. We did not use **DAD** and **ADA** triads for the calculations to avoid arbitrariness, as suggested by the reviewer, and deleted the Supplementary Tables 13–16. We performed numerical simulations for the activation of a_g modes using the donor–acceptor dimer model proposed by Painelli, A. and Girlando, A. (*J. Chem. Phys.* **84**, 5655–5671 (1986)). The calculation requires the frequencies (ω) of the molecular normal modes (Q) and the corresponding g -values defined as $g = \sqrt{2\omega}d\epsilon/dQ$. We calculated these values for both the neutral and monovalent ionic states of the donor and acceptor and interpolated for the relevant ionicity states ($+\delta$ for the donor and $-\delta$ for the acceptor; δ : degree of charge transfer), according to the method outlined in the paper by Painelli, A. and Girlando, A. Using these values as initial parameters, calculations of the dimer model were conducted by adjusting the calculated values of ω and g for each vibrational mode within a reasonable range. The results showed a good agreement with experimental value as shown in Supplementary Fig. 35b,c, affirming that the dimer model accurately reproduced the experimental spectrum. We have included these findings and comments in the text and Supplementary Information as follows.

Fig. 5 | Polarized reflectivity of mixed-stack complexes. ... b, Temperature-dependent reflectivity of **2S–F₄** in the electric field of light parallel ($//$) and perpendicular (\perp) to the π -stacking directions for the low-energy region. **Arrows indicate an increase in the intensity of peaks assigned to the a_g and b_{3g} modes.**

(p. 41–44 in Supplementary Information)

Supplementary Note **87**: Optical reflection spectroscopy measurements

The mixed-stack complexes showed multiple sharp peaks in the **optical reflectivity** spectra **along the π -stacking direction** in the low-energy region of 0.14–0.19 eV at 293 K (Fig. 5b and Supplementary Fig. **3425**), except for the nearly neutral **2S–F₂**. **These peaks indicate**

electron-molecular vibration (EMV) coupling^{19,20} based on broken symmetry along the π -stacking direction possibly triggered by donor-acceptor π -dimerization.^{21,22} These signals are intrinsically IR-silent, as predicted by the single-crystal XRD that identified the structures categorized to be $P2_1/n$ with uniformly π -stacked donors and acceptors along the $C2$ -glide symmetry. However, these signals are IR-active in the case of π -dimerized structures, which was supported by the DFT-phonon calculations based on the single-crystal structure (Supplementary Figs. 26, 27, and Supplementary Table 5). The observation of EMV-couplings-based signals confirmed the structural perturbation of these complexes for the π -dimerization. It is noted that **2S-F₂**, with the fluorine atom positional disorder, did not generate EMV-coupling-based signals; the couplings can be quenched due to disordering or the nearly neutral state ($\delta=0.42(2)$).²¹

The EMV-coupling-based signals in polarized reflectivity spectra were assigned by Gaussian 16 program²³ to simulate phonon vibrations of triad molecules at the density functional theory (DFT) level with the M062X functional. We utilized the 6-31G(d) split valence plus polarization basis set was used.⁹⁻¹¹ For the IR signal simulation, we used the geometries of **DAD** and **ADA** triads, which are extracted from the single-crystal structure **2S-F₄** and composed of the uniformly stacked **2S-F₄**, and **2S-(DAD)** (Supplementary Fig. 26a and Supplementary Table 13), and **F₄-2S**, and **F₄-(ADA)** (Supplementary Fig. 26c and Supplementary Table 15). Additionally, we performed a simulation for the symmetry-broken electronic structure models formed by π -dimerization between the donor and acceptor. Thus, we simulated the signals for the asymmetric **D-AD** and **A-DA** triads by modifying uniformly stacked **DAD** and **ADA** triads by moving a donor or acceptor far away along the a -axis by 0.2 Å from the facing molecules (**D-AD**: Supplementary Fig. 26b and Supplementary Table 14; **A-DA**: Supplementary Fig. 26d and Supplementary Table 16). The simulated IR signals are shown in Supplementary Fig. 27 and the assignments were listed in Supplementary Table 5.

The optical conductivity spectrum of **2S-F₄** along the π -stacking direction, which was derived from the polarized infrared reflectivity spectrum, and Raman spectrum at the excitation wavelength of 532 nm at 293 K are shown in Supplementary Fig. 35b,d. The optical conductivity and Raman spectra likely exhibited the same modes. Considering the inversion center of the single-crystal structure of **2S-F₄** categorized to be $P2_1/n$ with uniform π -stacked donors and acceptors along the $C2$ -glide symmetry, the Raman-active modes

should not be visible in IR, while the IR-active modes should not be visible in Raman. The coincidence of the shapes of the observed IR and Raman spectra indicates the dynamic fluctuating of donor–acceptor- π -stacking dimerization.^{24,25} The optical activation of the a_g mode along the π -stacking direction can be ascribed to the electron-molecular vibration (EMV) couplings.^{26,27}

To confirm that the EMV coupling effect is responsible for these signals, we performed numerical simulations of the optical conductivity spectrum using the donor–acceptor dimer model proposed by Painelli, A. and Girlando, A.²⁸ The calculation requires the frequencies (ω) of the molecular normal modes (Q) and the corresponding g -values defined as $g = \sqrt{2\omega}d\epsilon/dQ$. Initially, we performed DFT calculations for the isolated states of the donor and acceptor molecules to ascertain the initial parameters for the simulation. The eigenfrequencies of molecular vibrations were determined through standard DFT vibrational analysis. The g -value, which are the derivatives of the radical electronic level ϵ_{HOMO} (or ϵ_{LUMO} for neutral acceptor) in terms of Q , are estimated by slightly deforming the molecular structure along Q and calculating the energy shifts of the HOMO or LUMO levels. The quantum chemical calculation package Gaussian16²⁹ was used for the DFT calculations. The B3LYP function was used for the density functional, with the basis sets 6-311G(d)^{30,31,32} for the donor molecule and 6-311G+(d) for the acceptor molecule. Noting that B3LYP tends to overestimate eigenfrequencies ω , the calculated frequencies were corrected by applying the standard scaling factor of 0.96. The values of ω and g vary based on the valence state of the molecule. In the present study, we calculated these values for both the neutral and monovalent ionic states of the donor and acceptor and interpolated for the relevant ionicity states ($+\delta$ for the donor and $-\delta$ for the acceptor; δ : degree of charge transfer; Supplementary Table 4), according to the method outlined in the paper by Painelli, A. and Girlando, A.²⁸ Using these values as initial parameters, calculations of the dimer model were conducted. The specific parameters used for the calculation are detailed in Supplementary Table 5. By adjusting the calculated values of ω and g for each vibrational mode as listed in Supplementary Table 4, we achieved the good agreement with experimental values as shown in Supplementary Fig. 35b,c. The adjustment of parameters falls within a reasonable range, affirming that the dimer model accurately reproduced the experimental spectrum.

The optical activation of the a_g mode due to EMV coupling implies that the one-

dimensional molecular stacking is not uniform, contradicting the uniform columnar structure predicted by X-ray structural analysis (Supplementary Table 2 and Supplementary Fig. 10). The activation of the EMV modes hints at dynamic deviations from the average structure within the molecular columns, supporting the dynamic π -dimerization fluctuation. In the unit cell of this crystal, there are two sets of the donor–acceptor pairs, which are linked via inversion symmetry. The transition moments generated by the EMV coupling within a pair couple in either the same direction or in the opposite direction with another pair, resulting in both infrared active and Raman active EMV modes. This explains the coincidence of the shapes of the observed IR and Raman spectra. The slight difference in peak position between the IR and Raman signals can be understood as a Davydov splitting due to the dipole interaction between the pairs.

Supplementary Fig. 26: Molecular structures of triad molecules used for the simulation of polarized reflectivity. The uniformly stacked **DAD** (a) and **ADA** (c) triads. The symmetry-broken **D-AD** (b) and **A-DA** (d) triads. The distances between the molecular center of gravity for **2S** and **F₄** were shown. The centers of gravity were calculated to be the centroids of ten C and S atoms of thiophenes for **2S** and twelve C atoms for **F₄**, respectively.

Supplementary Table 5: Signal assignment of vibration modes for polarized reflectivity spectra. The appearance of the listed peaks for **D-AD** and **A-DA** triad (Supplementary Fig. 26b,d) was simulated by DFT calculation (Gaussian, M062x/6-31G(d)).

Entry	Photon energy (eV)		Vibration mode	Possible assignment
	D-AD	A-DA		
1	0.143	0.143	D-b_{3g}	ν_1
2	0.145		D-b_{1tt}	
3	0.149	0.149	D-b_{1tt}	ν_2
4	0.152	0.152	A-a_g	ν_3
5	0.157	0.156	D-a_g	ν_5
6	0.159	0.160	A-b_{1tt}	ν_6
7	0.159		A-b_{1tt}	ν_7
8	0.161	0.161	D-b_{1tt}	ν_8
9	0.162	0.162	D-a_g	ν_9
10	0.164	0.164	D-b_{2tt}	
11	0.165	0.164	A-b_{1tt}	
12	0.166	0.167	D-a_g	ν_{10}
13	0.173	0.177	A-a_g	ν_{11}
14		0.181	A-b_{1tt}	
15	0.183	0.183	A-b_{2tt}	
16	0.192	0.191	A-b_{2tt}	ν_{12}
17	0.203	0.204	A-a_g	ν_{13}

Supplementary Fig. 35: Temperature-dependent polarized reflectivity spectra. **a**, The spectra were obtained by applying an electric field along the π -stacking direction (solid lines) and perpendicular to the π -stacking direction (a dashed line). **b**, Optical conductivity spectrum derived from the polarized infrared spectrum at 293 K along the π -stacking direction through the Kramers-Kronig transformation. **c**, Simulated EMV coupling-based spectrum. The values of ω and g for both the neutral and monovalent ionic states of the donor and acceptor were calculated and interpolated for the relevant ionicity according to the method outlined in the paper by Painelli, A. and Girlando, A.,²⁸ as listed in Supplementary Table 4. Using these values as initial parameters, calculations of the dimer model were conducted. The specific parameters in the calculations are listed in Supplementary Table 5. **d**, Raman spectrum at 293 K (Excitation wavelength: 532 nm).

Supplementary Table 4: The fundamental molecular vibration modes of donor and acceptor molecules estimated in the relevant ionicity calculated from their neutral and ionic states.

Entry	Molecule	Wavenumber (cm ⁻¹)	g value (meV)
1	F ₄	1608.28	43.85
2	2S	1435.71	5.84
3	F ₄	1405.09	126.67
4	2S	1380.18	12.46
5	2S	1335.17	21.54
6	2S	1312.94	27.06
7	2S	1295.49	18.23
8	F ₄	1235.33	47.93
9	2S	1156.45	25.70
10	2S	1116.31	6.30

Supplementary Table 5: Parameters used in the calculation of the vibronic spectrum of 2S–F₄.

Entry	Molecule	Wavenumber (cm ⁻¹)	g value (meV)	linewidth (cm ⁻¹)
1	F ₄	1650	20	10
2	2S	1435	10	10
3	F ₄	1395	30	10
4	2S	1320	30	10
5	2S	1295	50	10
6	2S	1260	40	10
7	2S	1250	10	10
8	F ₄	1245	40	10
9	2S	1215	60	10
10	2S	1170	30	10

Supplementary Table 13: The geometry of optimized structure for (DAD)⁺⁺

(Supplementary Fig. 26a). The calculation was performed by Gaussian (M062x/6-

31G+(d)) with setting total charge to be +1.

S1	1.8719	-0.8716	6.7626
S2	2.117	-4.7036	4.9624
S3	2.0312	-2.0693	2.6386
S4	1.8468	-3.6883	7.9307
C2	2.0052	-2.9933	5.288
C3	1.9872	-1.9144	4.3726
C4	1.9265	-0.654	5.0175
C1	1.9194	-2.5813	6.6133
C5	2.6124	-4.7124	3.2376
H5A	3.5311	-4.4082	3.1686
H5B	2.5736	-5.6208	2.8996
C6	1.7348	-3.8314	2.3889
H6A	0.8064	-4.0252	2.5908
H6B	1.885	-4.0454	1.4547
C7	1.8673	-2.6277	9.3749
H7A	1.0989	-2.0526	9.3605
H7B	2.6659	-2.0948	9.3691
H7C	1.8485	-3.1674	10.1685
S1	1.9499	0.8716	2.6984
S2	1.7048	4.7036	4.4986
S3	1.7906	2.0693	6.8224
S4	1.9751	3.6883	1.5303
C2	1.8167	2.9933	4.1731
C3	1.8346	1.9144	5.0884
C4	1.8953	0.654	4.4436
C1	1.9024	2.5813	2.8478
C5	1.2094	4.7124	6.2235
H5A	0.2908	4.4082	6.2924
H5B	1.2482	5.6208	6.5614
C6	2.087	3.8314	7.0721

H6A	3.0154	4.0252	6.8702
H6B	1.9368	4.0454	8.0063
C7	1.9545	2.6277	0.0861
H7A	2.723	2.0526	0.1005
H7B	1.1559	2.0948	0.092
H7C	1.9733	3.1674	-0.7075
S1	8.6842	-0.8716	6.7626
S2	8.9293	-4.7036	4.9624
S3	8.8435	-2.0693	2.6386
S4	8.6591	-3.6883	7.9307
C2	8.8175	-2.9933	5.288
C3	8.7995	-1.9144	4.3726
C4	8.7388	-0.654	5.0175
C1	8.7317	-2.5813	6.6133
C5	9.4247	-4.7124	3.2376
H5A	10.3434	-4.4082	3.1686
H5B	9.3859	-5.6208	2.8996
C6	8.5471	-3.8314	2.3889
H6A	7.6187	-4.0252	2.5908
H6B	8.6973	-4.0454	1.4547
C7	8.6796	-2.6277	9.3749
H7A	7.9112	-2.0526	9.3605
H7B	9.4782	-2.0948	9.3691
H7C	8.6608	-3.1674	10.1685
S1	8.7622	0.8716	2.6984
S2	8.5171	4.7036	4.4986
S3	8.6029	2.0693	6.8224
S4	8.7874	3.6883	1.5303
C2	8.629	2.9933	4.1731
C3	8.6469	1.9144	5.0884
C4	8.7076	0.654	4.4436
C1	8.7147	2.5813	2.8478
C5	8.0217	4.7124	6.2235

H5A	7.1031	4.4082	6.2924
H5B	8.0605	5.6208	6.5614
C6	8.8993	3.8314	7.0721
H6A	9.8277	4.0252	6.8702
H6B	8.7491	4.0454	8.0063
C7	8.7668	2.6277	0.0861
H7A	9.5353	2.0526	0.1005
H7B	7.9682	2.0948	0.092
H7C	8.7856	3.1674	-0.7075
F2	5.1776	2.3433	6.0856
F1	5.2516	0.1041	7.4356
N2	5.5699	-4.811	5.2064
N1	5.2656	-2.6521	8.7533
C8	5.3541	-1.2459	5.4798
C11	5.39	-2.4535	6.2045
C10	5.2443	1.1914	5.4004
C9	5.2796	0.0338	6.1012
C13	5.4961	-3.7387	5.6038
C12	5.3202	-2.509	7.6237
F2	5.4565	-2.3433	3.3754
F1	5.3825	-0.1041	2.0254
N2	5.0642	4.811	4.2546
N1	5.3685	2.6521	0.7077
C8	5.28	1.2459	3.9812
C11	5.2441	2.4535	3.2565
C10	5.3898	-1.1914	4.0607
C9	5.3545	-0.0338	3.3598
C13	5.138	3.7387	3.8573
C12	5.3139	2.509	1.8373

Supplementary Table 14: The geometry of optimized structure for (D-AD)⁺⁺ (Supplementary Fig. 26b). The calculation was performed by Gaussian (M062x/6-31G+(d)) with setting total charge to be +1.

S1	1.8719	-0.8716	6.7626
S2	2.117	-4.7036	4.9624
S3	2.0312	-2.0693	2.6386
S4	1.8468	-3.6883	7.9307
C2	2.0052	-2.9933	5.288
C3	1.9872	-1.9144	4.3726
C4	1.9265	-0.654	5.0175
C1	1.9194	-2.5813	6.6133
C5	2.6124	-4.7124	3.2376
H5A	3.5311	-4.4082	3.1686
H5B	2.5736	-5.6208	2.8996
C6	1.7348	-3.8314	2.3889
H6A	0.8064	-4.0252	2.5908
H6B	1.885	-4.0454	1.4547
C7	1.8673	-2.6277	9.3749
H7A	1.0989	-2.0526	9.3605
H7B	2.6659	-2.0948	9.3691
H7C	1.8485	-3.1674	10.1685
S1	1.9499	0.8716	2.6984
S2	1.7048	4.7036	4.4986
S3	1.7906	2.0693	6.8224
S4	1.9751	3.6883	1.5303
C2	1.8167	2.9933	4.1731
C3	1.8346	1.9144	5.0884
C4	1.8953	0.654	4.4436
C1	1.9024	2.5813	2.8478
C5	1.2094	4.7124	6.2235
H5A	0.2908	4.4082	6.2924
H5B	1.2482	5.6208	6.5614

C6	2.087	3.8314	7.0721
H6A	3.0154	4.0252	6.8702
H6B	1.9368	4.0454	8.0063
C7	1.9545	2.6277	0.0861
H7A	2.723	2.0526	0.1005
H7B	1.1559	2.0948	0.092
H7C	1.9733	3.1674	-0.7075
S1	8.6842	-0.8716	6.7626
S2	8.9293	-4.7036	4.9624
S3	8.8435	-2.0693	2.6386
S4	8.6591	-3.6883	7.9307
C2	8.8175	-2.9933	5.288
C3	8.7995	-1.9144	4.3726
C4	8.7388	-0.654	5.0175
C1	8.7317	-2.5813	6.6133
C5	9.4247	-4.7124	3.2376
H5A	10.3434	-4.4082	3.1686
H5B	9.3859	-5.6208	2.8996
C6	8.5471	-3.8314	2.3889
H6A	7.6187	-4.0252	2.5908
H6B	8.6973	-4.0454	1.4547
C7	8.6796	-2.6277	9.3749
H7A	7.9112	-2.0526	9.3605
H7B	9.4782	-2.0948	9.3691
H7C	8.6608	-3.1674	10.1685
S1	8.7622	0.8716	2.6984
S2	8.5171	4.7036	4.4986
S3	8.6029	2.0693	6.8224
S4	8.7874	3.6883	1.5303
C2	8.629	2.9933	4.1731
C3	8.6469	1.9144	5.0884
C4	8.7076	0.654	4.4436
C1	8.7147	2.5813	2.8478

C5	8.0217	4.7124	6.2235
H5A	7.1031	4.4082	6.2924
H5B	8.0605	5.6208	6.5614
C6	8.8993	3.8314	7.0721
H6A	9.8277	4.0252	6.8702
H6B	8.7491	4.0454	8.0063
C7	8.7668	2.6277	0.0861
H7A	9.5353	2.0526	0.1005
H7B	7.9682	2.0948	0.092
H7C	8.7856	3.1674	-0.7075
F2	5.2776	2.3433	6.0856
F1	5.3516	0.1041	7.4356
N2	5.6699	-4.811	5.2064
N1	5.3656	-2.6521	8.7533
C8	5.4541	-1.2459	5.4798
C11	5.49	-2.4535	6.2045
C10	5.3443	1.1914	5.4004
C9	5.3796	0.0338	6.1012
C13	5.5961	-3.7387	5.6038
C12	5.4202	-2.509	7.6237
F2	5.5565	-2.3433	3.3754
F1	5.4825	-0.1041	2.0254
N2	5.1642	4.811	4.2546
N1	5.4685	2.6521	0.7077
C8	5.38	1.2459	3.9812
C11	5.3441	2.4535	3.2565
C10	5.4898	-1.1914	4.0607
C9	5.4545	-0.0338	3.3598
C13	5.238	3.7387	3.8573
C12	5.4139	2.509	1.8373

Supplementary Table 15: The geometry of optimized structure for (ADA)⁻ (Supplementary Fig. 26c). The calculation was performed by Gaussian (M062x/6-31G+(d)) with setting total charge to be -1.

S1	1.8719	-0.8716	6.7626
S2	2.117	-4.7036	4.9624
S3	2.0312	-2.0693	2.6386
S4	1.8468	-3.6883	7.9307
C2	2.0052	-2.9933	5.288
C3	1.9872	-1.9144	4.3726
C4	1.9265	-0.654	5.0175
C1	1.9194	-2.5813	6.6133
C5	2.6124	-4.7124	3.2376
H5A	3.5311	-4.4082	3.1686
H5B	2.5736	-5.6208	2.8996
C6	1.7348	-3.8314	2.3889
H6A	0.8064	-4.0252	2.5908
H6B	1.885	-4.0454	1.4547
C7	1.8673	-2.6277	9.3749
H7A	1.0989	-2.0526	9.3605
H7B	2.6659	-2.0948	9.3691
H7C	1.8485	-3.1674	10.1685
S1	1.9499	0.8716	2.6984
S2	1.7048	4.7036	4.4986
S3	1.7906	2.0693	6.8224
S4	1.9751	3.6883	1.5303
C2	1.8167	2.9933	4.1731
C3	1.8346	1.9144	5.0884
C4	1.8953	0.654	4.4436
C1	1.9024	2.5813	2.8478
C5	1.2094	4.7124	6.2235
H5A	0.2908	4.4082	6.2924
H5B	1.2482	5.6208	6.5614
C6	2.087	3.8314	7.0721

H6A	3.0154	4.0252	6.8702
H6B	1.9368	4.0454	8.0063
C7	1.9545	2.6277	0.0861
H7A	2.723	2.0526	0.1005
H7B	1.1559	2.0948	0.092
H7C	1.9733	3.1674	-0.7075
F2	-1.6347	2.3433	6.0856
F1	-1.5607	0.1041	7.4356
N2	-1.2424	-4.811	5.2064
N1	-1.5467	-2.6521	8.7533
C8	-1.4582	-1.2459	5.4798
C11	-1.4223	-2.4535	6.2045
C10	-1.568	1.1914	5.4004
C9	-1.5327	0.0338	6.1012
C13	-1.3162	-3.7387	5.6038
C12	-1.4921	-2.509	7.6237
F2	-1.3558	-2.3433	3.3754
F1	-1.4298	-0.1041	2.0254
N2	-1.7481	4.811	4.2546
N1	-1.4438	2.6521	0.7077
C8	-1.5323	1.2459	3.9812
C11	-1.5682	2.4535	3.2565
C10	-1.4225	-1.1914	4.0607
C9	-1.4578	-0.0338	3.3598
C13	-1.6743	3.7387	3.8573
C12	-1.4984	2.509	1.8373
F2	5.1776	2.3433	6.0856
F1	5.2516	0.1041	7.4356
N2	5.5699	-4.811	5.2064
N1	5.2656	-2.6521	8.7533
C8	5.3541	-1.2459	5.4798
C11	5.39	-2.4535	6.2045
C10	5.2443	1.1914	5.4004

C9	5.2796	0.0338	6.1012
C13	5.4961	-3.7387	5.6038
C12	5.3202	-2.509	7.6237
F2	5.4565	-2.3433	3.3754
F1	5.3825	-0.1041	2.0254
N2	5.0642	4.811	4.2546
N1	5.3685	2.6521	0.7077
C8	5.28	1.2459	3.9812
C11	5.2441	2.4535	3.2565
C10	5.3898	-1.1914	4.0607
C9	5.3545	-0.0338	3.3598
C13	5.138	3.7387	3.8573
C12	5.3139	2.509	1.8373

Supplementary Table 16: The geometry of optimized structure for (A-DA)⁻ (Supplementary Fig. 26d). The calculation was performed by Gaussian (M062x/6-31G+(d)) with setting total charge to be -1.

S1	1.9719	-0.8716	6.7626
S2	2.217	-4.7036	4.9624
S3	2.1312	-2.0693	2.6386
S4	1.9468	-3.6883	7.9307
C2	2.1052	-2.9933	5.288
C3	2.0872	-1.9144	4.3726
C4	2.0265	-0.654	5.0175
C1	2.0194	-2.5813	6.6133
C5	2.7124	-4.7124	3.2376
H5A	3.6311	-4.4082	3.1686
H5B	2.6736	-5.6208	2.8996
C6	1.8348	-3.8314	2.3889
H6A	0.9064	-4.0252	2.5908
H6B	1.985	-4.0454	1.4547
C7	1.9673	-2.6277	9.3749

H7A	1.1989	-2.0526	9.3605
H7B	2.7659	-2.0948	9.3691
H7C	1.9485	-3.1674	10.1685
S1	2.0499	0.8716	2.6984
S2	1.8048	4.7036	4.4986
S3	1.8906	2.0693	6.8224
S4	2.0751	3.6883	1.5303
C2	1.9167	2.9933	4.1731
C3	1.9346	1.9144	5.0884
C4	1.9953	0.654	4.4436
C1	2.0024	2.5813	2.8478
C5	1.3094	4.7124	6.2235
H5A	0.3908	4.4082	6.2924
H5B	1.3482	5.6208	6.5614
C6	2.187	3.8314	7.0721
H6A	3.1154	4.0252	6.8702
H6B	2.0368	4.0454	8.0063
C7	2.0545	2.6277	0.0861
H7A	2.823	2.0526	0.1005
H7B	1.2559	2.0948	0.092
H7C	2.0733	3.1674	-0.7075
F2	-1.6347	2.3433	6.0856
F1	-1.5607	0.1041	7.4356
N2	-1.2424	-4.811	5.2064
N1	-1.5467	-2.6521	8.7533
C8	-1.4582	-1.2459	5.4798
C11	-1.4223	-2.4535	6.2045
C10	-1.568	1.1914	5.4004
C9	-1.5327	0.0338	6.1012
C13	-1.3162	-3.7387	5.6038
C12	-1.4921	-2.509	7.6237
F2	-1.3558	-2.3433	3.3754
F1	-1.4298	-0.1041	2.0254

N2	-1.7481	4.811	4.2546
N1	-1.4438	2.6521	0.7077
C8	-1.5323	1.2459	3.9812
C11	-1.5682	2.4535	3.2565
C10	-1.4225	-1.1914	4.0607
C9	-1.4578	-0.0338	3.3598
C13	-1.6743	3.7387	3.8573
C12	-1.4984	2.509	1.8373
F2	5.1776	2.3433	6.0856
F1	5.2516	0.1041	7.4356
N2	5.5699	-4.811	5.2064
N1	5.2656	-2.6521	8.7533
C8	5.3541	-1.2459	5.4798
C11	5.39	-2.4535	6.2045
C10	5.2443	1.1914	5.4004
C9	5.2796	0.0338	6.1012
C13	5.4961	-3.7387	5.6038
C12	5.3202	-2.509	7.6237
F2	5.4565	-2.3433	3.3754
F1	5.3825	-0.1041	2.0254
N2	5.0642	4.811	4.2546
N1	5.3685	2.6521	0.7077
C8	5.28	1.2459	3.9812
C11	5.2441	2.4535	3.2565
C10	5.3898	-1.1914	4.0607
C9	5.3545	-0.0338	3.3598
C13	5.138	3.7387	3.8573
C12	5.3139	2.509	1.8373

(Supplementary Reference)

28: Painelli, A. & Girlando, A. Electron–molecular vibration (e–mv) coupling in charge-transfer compounds and its consequences on the optical spectra: A theoretical framework. *J. Chem. Phys.* **84**, 5655–5671 (1986).

30. Binning Jr. R. C. & Curtiss, L. A. Compact contracted basis sets for third-row atoms: Ga–Kr. *J. Comp. Chem.* **11**, 1206–1216 (1990).

31. Rassolov, V. A., Pople, J. A., Ratner, M. A. & Windus, T. L. 6-31G* basis set for atoms K through Zn. *J. Chem. Phys.* **109**, 1223–1229 (1998).

32. Rassolov, V. A., Ratner, M. A., Pople, J. A., Redfern, P. C. & Curtiss, L. A. 6-31G* basis set for third-row atoms. *J. Comp. Chem.* **22**, 976–984 (2001).

For the contributions to the calculation study for the vibrational modes, we added Kaoru Yamamoto (Okayama Univ. of Sci.) and Akira Takahashi (Nagoya Institute of Technology) as the authors. We also updated the "**Acknowledgment**" and "**Authors contributions**" as follows.

(p. 32 in text)

The computation for simulating EMV coupling was performed using Research Center for Computational Science, Okazaki, Japan (Project No. 23-IMS-C276).

(p. 33 in text)

... K.Yamamoto, T.F., and A.T. performed simulation for molecular vibrations.

e) I do not agree with the interpretation of the phase transition of 2F-S4. The uncertainty in the values of δ obtained from X-ray do not allow to establish such an increase of ionicity, nor this fact justifies the small increase (if any... the authors should use an internal standard and measure the areas to confirm the intensity increase of the e-mv bands if Fig. 5b). I believe the transtion is more likely a disorder to order.

We thank this reviewer's important comments. We were unable to identify the molecular structure at the atomic level nor estimate the value of δ from the bond length analyses for the large-sized superlattice structure of the newly obtained XRD patterns in the low-temperature phase. We acknowledge the reviewer's comment regarding the uncertainty in the value of δ and have revised the corresponding comments in the text and deleted Supplementary Fig. 12 and Supplementary Reference 23 and 24 as follows.

(p. 17–18 in text)

The XRD pattern results in $P2_1/n$ space group at 300 K showed significant X-ray diffuse scattering (Supplementary Fig. 14), possibly due to the precursor phenomenon for the π -

dimerization, which may result in EMV coupling-based signals at room temperature. On the other hand, the scattering disappeared (Supplementary Figs. 15 and 16) at 200 K, and the XRD low-temperatures showed superlattice patterns with dimensions of $a \times 2b \times 2c$ (Supplementary Figs. 13 and 17), the complex maintained the $P2_1/n$ space group without possibly implying the π -dimerization along the a -axis (Supplementary Note 6) (Supplementary Table 3). However, the bond length analyses of the donor and acceptor in the complex indicate that δ increases upon cooling (namely from "N-I boundary" to "more-ionic" states; Supplementary Fig. 12), which were estimated to be 0.63(3) at 293 K, 0.76(3) at 240 K, 0.81(4) at 200 K, and 0.76(3) at 173 K from Kistenmacher's relationship³⁷ for the structures of acceptors. The increase in δ suggested the strengthened degree of π -dimerization,^{43,44} reflecting the structural fluctuations evident in the EMV coupling based signals in the reflectivity at room temperature (Fig. 5b and Supplementary Figs. 25–27). The broken symmetry upon the possible strengthened degree of π -dimerization of the complex may induce was experimentally identified by the markedly intensified

Supplementary Fig. 12: Temperature dependence of bond lengths of mixed-stack complexes. a, b, c, Temperature dependent d_{CC} and d_{CS} of donors and acceptors in **2S-F₄** with labels for the bonds. Observed bond lengths of d_{CC} of **2S** (a), d_{CS} of **2S** (b), and d_{CC} of **F₄** (c)

in the single crystal. $i = (i^+ + i^-)/2$. As the references, the bond lengths in single crystals of neutral **2S** ($\delta=0$), **F₄** ($\delta=0$), $1e^-$ oxidized **2S** ($\delta=1$), and $1e^-$ reduced **F₄** ($\delta=1$) were shown. **d,e,f**, Simulated bond lengths of d_{CC} of **2S** (**d**), d_{CS} of **2S** (**e**), and d_{CC} of **F₄^{+6,17-}** (**f**) in their optimized structures by Gaussian (Supplementary Figs. 4 and 5).

(Supplementary Reference)

23. Emge, T. J., Maxfield, M., Cowan, D. O. & Kistenmacher, T. J. Solution and solid state studies of tetrafluoro-7,7,8,8-tetracyano-*p*-quinodimethane, TCNQF₄. Evidence for long-range amphoteric intermolecular interactions and low dimensionality in the solid state structure. *Mol. Cryst. Liq. Cryst.* **65**, 161–178 (1981).

24. O’Kane, S. A., Clérac, R., Zhao, H., Ouyang, X., Galán Mascarós, J. R., Heintz, R. & Dunbar, K. R. New crystalline polymers of Ag(TCNQ) and Ag(TCNQF₄): structures and magnetic properties. *J. Solid State Chem.* **152**, 159–173 (2000).

Additionally, based on the feedback of this reviewer, we have replaced Reference 49 in the text with more suitable literature. This literature was also inserted as Supplementary Reference 28 as follows.

(Reference in text)

49: Painelli, A. & Girlando, A. Electron–molecular vibration (e–mv) coupling in charge-transfer compounds and its consequences on the optical spectra: A theoretical framework. *J. Chem. Phys.* **84**, 5655–5671 (1986). Rice, M. J. Organic linear conductors as systems for the study of electron-phonon interactions in the organic solid state. *Phys. Rev. Lett.* **37**, 36–39 (1976).

(Supplementary Reference)

28: Painelli, A. & Girlando, A. Electron–molecular vibration (e–mv) coupling in charge-transfer compounds and its consequences on the optical spectra: A theoretical framework. *J. Chem. Phys.* **84**, 5655–5671 (1986).

Our reply to the Comments from Reviewer #2

We sincerely appreciate the favorable comments and important suggestions. The comments are shown in blue, and our responses are shown in black. The revised portions in the text and Supplementary information are highlighted in yellow.

This work by Fujino, Mori et al. deals with molecular charge-transfer complexes with alternated stacks, usually found in an ionic (I) or neutral (N) states, with in both cases low conductivity, at variance with the very rare examples reported here where mixed-stacks complexes with a charge transfer close to the N-I boundary favors a high conductivity. The main originality of the work is therefore to be found in (i) the ability to engineer CT salts at the N-I boundary by adapting the redox potentials of both partners, (ii) the adaptation of HOMO and LUMO symmetry to favor the best possible overlap within the stacks, using original bis(thiophene) derivatives whose HOMO symmetry differs from that of classical TTFs. These combined approaches allow for the isolation of conducting salts with RT conductivity one order of magnitude higher than those found earlier (and not several orders or magnitude as stated page 13). Another interesting point is that the most conducting material exhibits the largest band gap, a consequence of increased π -dimerization tendency, an observation that perhaps limits the approach toward even more conducting (or metallic) materials.

Besides, the experimental work is complete and the analyses of the data are of excellent quality. The methodology is sound, the experiments well described to be reproduced and the supplementary material very helpful.

Altogether, I recommend publication in Nature Comm. Such materials at the N-I boundary are really hard to find (and only really identified under pressure) and their high conductivity, high solubility and air stability provide opportunities for both fundamental new physics and interesting application in electronics.

As minor modifications, I suggest

1) It should be specified (top of page 6) that the synthesis of 2S reported here complements another route to 2S described by the same authors in a recent patent (WO2020262443).

We would like to express our gratitude for this reviewer's important suggestion. We added the description to specify the synthetic revision in the text and Supplementary Note 3 and added the patent as the reference in the Supplementary Information as follows.

(p. 6 in text)

First, **2S** with non-bulky methylthio groups was synthesized in 52% yield after three-step transformations from 2,2'-bi(3,4-ethylenedithiophene) **1**³² (Fig. 2a, **Supplementary Note 3**, Supplementary Figs. 7, **4133**, **4234**, and Supplementary Table 1).

(p. 5 in Supplementary Information)

Supplementary Note 3: Synthesis of donor 2S and charge-transfer salt 2S•BF₄

A donor **2S** was designed to have methylthio groups at its ends. These groups are small enough not to interfere with the π -stacking of the molecule during crystallization, similar to **2O**.¹ Initially, we attempted to dimerize unsubstituted or dibrominated monomers, followed by lithiation and methylthio substitution.^{1,3} However, these attempts resulted in low yields. Therefore, we modified the synthesis route by starting with bromination of unsubstituted dimer **1**, followed by lithiation and methyl thiolation, as shown in Fig. 2a and Methods section in the text.

(Supplementary Reference)

3: Seino, Y., Nakamura, H., Mori, H., Fujino, T., Dekura, S. & Kameyama, R. Conductive oligomer, conductive composition, conductive aid, and condenser electrode, transparent electrode, battery electrode, or capacitor electrode formed using said conductive composition. *WO2020262443*, Publication Date 30/12/2020.

2) Page 6, lines 12 and following. The text should detail and give references to the concept of pitch for the node pattern of the frontier orbitals.

We appreciate this reviewer's important question. The average periodicity of horizontally nodal frontier orbitals in donors and acceptors were calculated using the S–S distances of methylthio groups for donors and N–N distances of the cyano groups for acceptors by dividing the number of the nodes. We added the explanation on the estimation of the periodicity in the text, Supplementary Note 5, Supplementary Fig. 5, and reference 15 as follows.

(p. 7 in text)

... consistent horizontally nodal node patterns with an average periodicity pitch of 2.0 Å that correspond well with those of 1e⁻-reduced F₄/F₂ with an average periodicity pitch of 1.7–1.8

Å (Fig. 3b, Supplementary Fig. 5, and Supplementary Tables 8, 10, and 8–12).

(p. 50 in text: Caption for Fig. 3b)

These orbitals have horizontally nodal patterns with an average periodicity pitch of 1.8–2.0 Å...

(p. 8 in Supplementary Information)

The average periodicity of horizontally nodal orbitals¹⁵ of $2O^{+}$, $2S^{+}$, F_4^{-} , and F_2^{-} were calculated using the S–S distances of methylthio groups for $2O^{+}$ and $2S^{+}$ and N–N distances of the cyano groups for F_4^{-} for F_2^{-} by dividing the number of the nodes.

Supplementary Fig. 5: Calculated SOMO shapes of donors in a radical cation form and acceptors in a radical anion form. a, Radical cation $2O^{+}$.¹ b, Radical cation $2S^{+}$. c, Radical anion F_4^{-} . d, Radical anion F_2^{-} . Atoms were colored as follows; red: oxygen; yellow: sulfur; gray: carbon; aqua: fluorine; white: hydrogen. The calculated S–S and N–N distances and the average periodicity of horizontally nodal orbitals were shown.

(Supplementary Reference)

15: Kato, Y., Matsumoto, H. & Mori, T. Absence of HOMO/LUMO Transition in charge-transfer complexes of thienoacenes. *J. Phys. Chem. A* **125**, 146–153 (2021).

3) Correct the over-exaggerated statement page 13 last line

We thank this reviewer's kind comments. We followed the comments and modified the description as follows.

(p. 16 in text)

...were remarkably superior by a few ~~several~~ orders of magnitude to those of previously reported typical mixed-stack complexes with σ_{rt} (300 K) below 10^{-4} S cm⁻¹: ... The σ_{rt} of **2S-F₄** (0.10 S cm⁻¹) is the highest value reported to date for a single crystal of a 1D mixed-stack complex, and it is an order of magnitude higher than the previously highest value of 1.0×10^{-2} S cm⁻¹.²¹

4) Figure 4 is really hard to read. I suggest that the authors adopt for Fig 4b,c,e,f the same color code used in fig4d,g, i.e. the $\delta=0$ and $\delta=1$ text in black as the related points. In the fig 4 caption, it should be specified that full lines are for $\delta=1$ and dotted lines for $\delta=0$.

We appreciate this reviewer's kind comment. We modified the color codes for Fig. 4b,c,e,f to those used in the Fig. 4d,g. We also modified the color and type of lines; we showed the reference values for neutral donor and acceptor ($\delta = 0$) with the dotted black lines, while $1e^-$ -oxidized donors or $1e^-$ -reduced acceptors ($\delta = 1$) with the black solid lines, which was specified in the caption as follows.

Fig. 4 | Bond length analysis of mixed-stack complexes. **a**, Bond labels of 2O, 2S, F₄, and F₂. **b,c,d,e,f,g**, Comparison of C–C and C–S bond lengths (d_{CC} and d_{CS} , respectively) of

donors and acceptors in single crystals of mixed-stack complexes, with error bars (s.d.). As a reference, the bond lengths in single crystals of neutral donors or acceptors (i.e., $\delta = 0$) are shown with dotted lines and $1e^-$ -oxidized donors or $1e^-$ -reduced acceptors (i.e., $\delta = 1$) are shown with solid lines. Observed d_{CC} of **2O**³⁰ (**b**), d_{CC} of **2S** (**c**), d_{CC} of **F4**^{33,35} (**d**), d_{CS} of **2O**³⁰ (**e**), d_{CS} of **2S** (**f**), and d_{CC} of **F2**^{34,36} (**g**). $i = (i^1 + i^2)/2$.

5) In ref 26 correct spelling is: Torrance.

We appreciate this reviewer's kind comments. We modified the spelling in reference 26 as follows.

26. Torrance Torrence, J. B. An overview of organic charge-transfer solids: insulators, metals, and the neutral-ionic transition. *Mol. Cryst. Liq. Cryst.* **126**, 55–67 (1985).

Our reply to the Comments from Reviewer #3

We sincerely appreciate the favorable comments and important suggestions. The comments are shown in blue, and our responses are shown in black. The revised portions in the text and Supplementary information are highlighted in yellow.

In this work the authors extend their previous investigations on the conducting crystalline materials based on 2,2'-bi(3,4-ethylenedithiophene) and 2,2'-bi(3,4-ethylenedioxythiophene) capped here with methylthio groups, i.e. donors 2S and 2O, respectively. Here they obtained crystalline charge transfer complexes of both donors with the acceptors TCNQ-F4 and TCNQ-F2. Single crystal conductivity measurements of these charge transfer materials, where the donors and acceptors are alternated within 1D stacks, show semiconducting behaviour with relatively high conductivity values ($10^{-3} - 0.1 \text{ S cm}^{-1}$) for such type of alternated D-A compounds. State-of-the-art characterizations backed up with band structure and DFT calculations of the energy levels of donors and acceptors have been performed, confirming the good match between the electron donor and electron acceptor abilities of the partners, at the threshold of neutral-ionic boundary. The paper reads nicely and the results are of interest for the community. However, in spite of the use of the term "Topological fusion of donor and acceptor" in the title, which is completely misleading as one could think that the donors and the acceptors are chemically fused, while the authors refer to a mixing of the HOMO of the donor and the LUMO of the acceptor in the CT complex, in my opinion there is no striking novelty in the manuscript which could qualify it for the Nature journals portfolio. The conductivity of the CT compounds is certainly higher compared to other D-A alternated materials but still remains activated. I think some modulation of the conducting properties by light irradiation could maybe add interest to the paper.

We appreciate this reviewer's kind comments. We agree that the term "topological fusion of donor and acceptor" is misleading to the readers and corrected the title as follows. We are interested in exploring the conductive properties induced by external stimuli such as light irradiation as the important subject of future research.

(Title in text)

Orbital hybridization ~~Topological fusion~~ of donor and acceptor to enhance the conductivity of ~~form highly-conducting~~ mixed-stack complexes

We also corrected the related terms in the text as follows.

(Abstract in text)

Surprisingly, the orbitals were **highly hybridized topologically fused** in the single-crystal complexes, ...

(p. 6 in text)

Surprisingly, the similar donor HOMO and acceptor LUMO with comparable energy levels and well-matched orbital symmetries were **highly hybridized topologically fused** in the complexes, ...

(p. 12 in text)

..., which identified that the HOMO of the donor and LUMO of the acceptor are **highly hybridized topologically fused for the first time**.

(p. 13 in text)

Given that the strong hybridization between the donor HOMO and acceptor LUMO **to form topologically fused orbitals** in the band structures., ...

(p. 21 in text)

These complexes have **highly hybridized topologically fused** orbitals between the donor and acceptor and exhibited high σ_{r} (10^{-3} to 0.1 S cm^{-1}) under ambient conditions; ...

(p. 50 in text; Caption for Fig. 3d)

d, Highly hybridized Topologically fused HOCO and LUCO between donor and acceptor ...

Some revisions to be taken into account for a submission in a more specialized journal:

1) The Abstract should be more specific, with the description of the donor and acceptor molecules involved in the study.

We thank this reviewer's kind comments. We added the detailed description for the molecules as follows.

(Abstract in text)

In this study, mixed-stack complexes that uniquely exist at the neutral–ionic boundary were synthesized by combining donors **(bis(3,4-ethylenedithiophene))** and acceptors **(fluorinated tetracyanoquinodimethanes)**...

2) How are estimated the pitch values in Fig. 3b?

We sincerely appreciate this reviewer's important question. The average periodicities of horizontally nodal orbitals in donors and acceptors were calculated using the S–S distances of methylthio groups for donors and N–N distances of the cyano groups for acceptors by dividing the number of the nodes, as shown in Supplementary Fig. 5. We added the detailed description for this in the Supplementary Information and Supplementary Reference 15 as follows.

(p. 8 in Supplementary Information)

The average periodicity of horizontally nodal orbitals¹⁵ of $2O^{+}$, $2S^{+}$, F_4^{-} , and F_2^{-} were calculated using the S–S distances of methylthio groups for $2O^{+}$ and $2S^{+}$ and N–N distances of the cyano groups for F_4^{-} for F_2^{-} by dividing the number of the nodes.

Supplementary Fig. 5: Calculated SOMO shapes of donors in a radical cation form and acceptors in a radical anion form. a, Radical cation $2O^{+\bullet}$. b, Radical cation $2S^{+\bullet}$. c, Radical anion $F_4^{\bullet-}$. d, Radical anion $F_2^{\bullet-}$. Atoms were colored as follows; red: oxygen; yellow: sulfur; gray: carbon; aqua: fluorine; white: hydrogen. The calculated S-S and N-N distances and the average periodicity of nodal orbitals are shown.

(Supplementary Reference)

15. Kato, Y., Matsumoto, H. & Mori, T. Absence of HOMO/LUMO Transition in charge-transfer complexes of thienoacenes. *J. Phys. Chem. A* **125**, 146–153 (2021).

Our reply to the Comments from Reviewer #4

We sincerely appreciate the favorable comments and important suggestions. The comments are shown in blue, and our responses are shown in black. The revised portions in the text and Supplementary information are highlighted in yellow.

This is a very relevant and significant contribution by H. Mori and co-workers, reporting a cleverly designed study of a series of charge transfer salts near the border of the neutral to ionic transition. The results clearly show for the first time that near the border of this transition, in spite of the mixed stacking arrangement of donor and acceptor molecules, high electrical conductivity can be achieved due to a favourable combination (topological fusion) of donor and acceptor molecular orbitals.

The class of molecular materials displaying neutral to ionic transitions have been since the early days of molecular conducting materials, key compounds for understanding fundamental aspects of the electronic properties of molecular materials, attracting the attention of a wide scientific community in solid state physics and molecular materials science. These results of this study provide not only new compounds with a breaking record of electrical conductivity among this type of salts, but also pave the way to a new route to prepare highly conducting materials based on neutral species.

This study is well designed, combining a rational choice of new molecular units/compounds with a comprehensive physical characterisation of electronic and magnetic properties complemented by theoretical electronic structure quantum calculations. The main conclusions are overall well supported by the experimental results and theoretical calculations using state of the art techniques.

In spite of not being a native English speaker I feel that the manuscript would benefit from a throughout revision of the English style. Some points of the phrasing are not entirely clear.

Any way in view of the relevance and significance of the results I consider this work as certainly deserving publication in Nature Communications after some minor revisions and authors addressing the following secondary aspects.

One point that does not become clear in the discussion is the role of disorder in different compounds. Some of the structures present variable degrees of disorder that should not be neglected, as in 2S-F2 where there is Fluorine occupational disorder. Possible disorder effects should be taken into account

when comparing with other compounds. As mentioned in Fig S13 calculations were done considering only the geometry of the largest occupancy. The authors should make more clear and discuss here possible effects of disorder and for instance in the theoretical calculations consider possible differences for other geometries.

We would like to express our gratitude for the important comments from this reviewer. We agree with this reviewer's suggestions that the positional disorders of fluorine atoms in **2O-F₂**, which is newly analyzed in the revision, and **2S-F₂** in the 94:16 and 86:14 occupations, respectively, may influence the electronic structures that contribute to the optical and conductive nature. Therefore, we added descriptions of the possible effects of the positional disorders on the conductivity as follows.

(p. 13 in text)

The significant positional disordering of fluorine atoms in **2S-F₂** may also affect the shape of the spectrum.

(p. 14 in text)

..., conferring superior electrical conductivities⁴⁵⁻⁴⁸ upon the **2O** complexes, ~~although~~ **However**, it is not negligible that nearly neutral **2S-F₂** ~~may have~~ insufficient conductive carriers and positional disorders of fluorine atoms, which may impact the electronic structures that contribute to the conductivity.

Furthermore, we performed the additional calculation studies based on the structures in the minor occupancies to get an insight into the possible impacts. These data showed different electronic structures compared to those in the major occupancies, as per the reviewer's suggestion. While the U_{eff} values remained unaffected, there were significant impacts on the band structures, leading to a wider gap in the minor occupancies (0.07 eV for **2O-F₂** and 0.05 eV for **2S-F₂**) compared to those for the **2O-F₂** and **2S-F₂** in the major occupancies (less than or equal to 0.02 eV). This suggests that a band insulating state coexists in the minor occupancies. It is noted that various factors that affect the conductivities of mixed-stack complexes, including band dispersions (bandwidth W), carrier-to-carrier Coulomb repulsion (U_{eff} that includes effects of molecular orbital shapes, conjugate area sizes, and charge-transfer degrees), the proximity of the charge-transfer degrees to the neutral-ionic boundary, disorder-order fluctuations, and positional disorders of atoms. Although it is difficult to attribute the effects of positional disorders solely to the conductivities, the calculations helped in providing insights into the electronic structures that may

affect their conductivities. The results are presented in Supplementary Figs. 18c,f, 19c,e,h,j, 20c,f, 23, 26, 27c,f, 30, 33, and Supplementary Table 3 and the explanations are shown in the text and Supplementary Information as follows.

(p. 56 in text; Caption *e* for Table 1)

^eStructural data with major occupancy were used in the calculations for **2O–F₂** and **2S–F₂**. See Supplementary Information for the data with the minor occupancies.

(p. 25 in Supplementary Information)

...To get an insight into the possible impacts, we performed the calculations not only for the major but also for the minor occupancies...

Supplementary Fig. 183: Band structures of mixed-stack complexes. **a**, **2O-F₄**. **b**, **2O-F₂** (major occupancy). **c**, **2O-F₂** (minor occupancy). **d**, **2S-F₄**. **e**, **2S-F₂** (major occupancy). **f**, **2S-F₂** (minor occupancy). In the calculations for **2S-F₂**, the geometry of the major occupancies was used. Γ (0, 0, 0), X (0.5, 0, 0), Y (0, 0.5, 0), N (0, 0.5, 0.5), Z (0, 0, 0.5), P (-0.5, 0, 0.5), Q (-0.5, 0.5, 0.5). The complexes consistently exhibited half-filled 1D electronic structures (Fig. 3c and Supplementary Fig. 13). The real parts of the HOCO and lowest unoccupied crystal orbitals (LUCO) at the Γ point (0, 0, 0) were visualized by VESTA²⁵ (Fig. 3d and Supplementary Fig. 14). The Fermi levels (E_F) are determined by occupying electrons according to the Fermi distribution function.

Supplementary Fig. 194: Crystal orbitals of mixed-stack complexes. **a,b,c,d,e**, The LUCO shapes of **2O-F₄** (**a**), **2O-F₂** (**b**, major occupancy), **2O-F₂** (**c**, minor occupancy), and **2S-F₂** (**d**, major occupancy), and **2S-F₂** (**e**, minor occupancy). **d,f,g,h,i,j**, The HOCO shapes of **2O-F₄** (**fd**), **2O-F₂** (**ge**, major occupancy), **2O-F₂** (**h**, minor occupancy), and **2S-F₂** (**if**, major occupancy), and **2S-F₂** (**j**, minor occupancy). In the calculations for **2S-F₂**, the geometry of the major occupancies was used for the calculation. Orbitals were visualized by VESTA.²⁵¹⁸ Atoms were colored as follows; yellow: sulfur; red: oxygen; gray: carbon; blue:

nitrogen; yellowish green-aqua: fluorine.

Supplementary Fig. 205: Wannier interpolation bands (shown in green squares) and band dispersion (shown in black solid lines). a, $2O-F_4$. b, $2O-F_2$ (major occupancy). c, $2O-F_2$ (minor occupancy). d, $2S-F_4$. e, $2S-F_2$ (major occupancy). f, $2S-F_2$ (minor occupancy).

Supplementary Fig. 2347: The maximally localized Wannier function of 2O–F₂ (minor occupancy) in a cell. Atoms surrounding the molecules were omitted for clarity. Atoms were colored as follows; yellow: sulfur; red: oxygen; gray: carbon; blue: nitrogen; yellowish green: fluorine. **a**, 2O (0.5, 0, 0.5). **b**, 2O (0, 0.5, 0). **c**, F₂ (0, 0, 0.5). **d**, F₂ (0.5, 0.5, 0).

Supplementary Fig. 26: The maximally localized Wannier function of 2S–F₂ (minor occupancy) in a cell. Atoms surrounding the molecules were omitted for clarity. Atoms were colored as follows; yellow: sulfur; red: oxygen; gray: carbon; blue: nitrogen; yellowish green: fluorine. **a**, 2S (0.5, 0, 0.5). **b**, 2S (0, 0.5, 0). **c**, F₂ (0, 0, 0.5). **d**, F₂ (0.5, 0.5, 0).

Supplementary Fig. 270: Labels for t values for mixed-stack complexes. a, 2O-F₄. b, 2O-F₂ (major occupancy). c, 2O-F₂ (minor occupancy). de, 2S-F₄. ed, 2S-F₂ (major

occupancy). **f**, **2S-F₂** (minor occupancy). The values were summarized in Supplementary Table 34. **2O**, **2S**, **F₄**, and **F₂** were colored with a blue, red, green, and light green background, respectively for clarity. Atoms were colored as follows; yellow: sulfur; red: oxygen; gray: carbon; blue: nitrogen; yellowish green aqua: fluorine.

Supplementary Table 34: Transfer integrals for mixed-stack complexes. The intracolumnar values (t_1 and t_4) are shown as t_{DA} in Table 1.

Complex	2O-F₄	2O-F₂	2O-F₂	2S-F₄	2S-F₂	2S-F₂
		(major occupancy)	(minor occupancy)		(major occupancy)	(minor occupancy)
t_1 (eV)	0.2035	0.2097	0.197	0.208	0.2068	0.187
t_2 (eV)	-0.004976	-0.0042950	-0.00395	0.01832	-0.018792	0.0182
t_3 (eV)	0.00190117	-0.002290	-0.00200	-0.000301879	0.000517435	0.000759
t_4 (eV)	0.2035	0.2097	0.197	0.208	0.2068	0.187
t_5 (eV)	-0.00495501	-0.0042241	-0.00400	0.01832	-0.018792	0.0182
t_6 (eV)	-0.01210186	0.014137	0.0134	-0.000953276	-0.0025264	0.00190

Supplementary Fig. 30: Effective direct Coulomb interactions in a cell of **2O-F₂ in the minor occupancy. Values of donors (a,b), and acceptors (c,d).**

Supplementary Fig. 33: Effective direct Coulomb interactions in a cell of 2S–F₂ in the minor occupancy. Values of donors (a,b), and acceptors (c,d).

Another point to be taken into account are the values for electrical conductivity that were obtained either by 4-probe or only 2-probe techniques. The 2 probe values underestimate the intrinsic conductivity. This seems ignored in comparing the different compounds (first line of page 14 of the manuscript) although it probably does not significantly change conclusions.

We appreciate this reviewer's professional suggestions. We performed additional conductivity measurements with a four-probe method for 2O–F₄ and 2S–F₂, which were previously measured using the two-probe methods. The measurements showed comparable to higher room-temperature conductivities and comparable activation energies. These revised data have been included in the text, Table 1, and Fig. 6, and Supplementary Note 9 as follows.

(p. 16 in text)

The σ_{rt} values (293 K) determined via the direct-current method along the π -stacking direction (the a -axis) were remarkably superior by a few several orders of magnitude to those of previously reported typical mixed-stack complexes with σ_{rt} below 10^{-4} S cm⁻¹: $4.9 \pm 0.4 \times 10^{-43}$ S cm⁻¹ (2O–F₄), 1.46×10^{-2} S cm⁻¹ (2O–F₂), 0.10 S cm⁻¹ (2S–F₄), and $6.9 \pm 0.2 \times 10^{-3}$ S cm⁻¹ (2S–F₂) (Figs. 1b, 6, 7a, and Table 1) within the ohmic region (Supplementary Fig. 3628). ...

The ρ - T plots are indicative of semiconducting behavior with relatively small E_a values: 0.11327(1) eV (**2O-F₄**), 0.178(1) eV (**2O-F₂**), 0.200(1) eV (**2S-F₄**), and 0.11229(1) eV (**2S-F₂**) around room temperature (Figs. 6, 7a, and Table 1), ...

(p. 30 in text)

Electrical resistivity (ρ) measurements of the single-crystal mixed stack complexes **2O-F₄/2S-F₂** and **2O-F₂/2S-F₄** single crystals were performed by the conventional a two- and four-probe method, respectively,

(p. 49 in Supplementary Information)

Supplementary Note 98: Electrical resistivity measurements

The current-voltage (I - V) characteristics by a two-probe method at room temperature from -10 to 10 V for **2O-F₄/2S-F₂**, -3 to 3 V for **2S-F₂**, and -1.2 to 1.5 V for **2S-F₄** (Supplementary Fig. 3628) confirmed the ohmic behaviors for the V ranges. Within the ohmic regions, the temperature (T)-dependent resistance (R) of the sample was measured by a four-probe method at a constant direct current voltage (1 V) for **2O-F₄** and **2S-F₂** upon cooling the electrode from 293 to 200 K and subsequent heating to 293 K (ca. 1.5 K/min; Fig. 6). During the measurements, the temperature of the sample was monitored by the Cernox (Lake Shore) thermometer. For highly conducting single crystal **2O-F₂** and **2S-F₄**, the temperature dependent R was measured at a constant ac current (1 μ A) upon cooling the electrode from 340 to 200 K and subsequent heating to 340 K (by 1.0-1.5 K min⁻¹ for 200-300 K; 1.0 K/min for 300-340 K; (Figs. 6 and 7a).

Table 1 | Structural information and physical properties of mixed-stack complexes at 293 K.

Donor (D)–acceptor (A)	2O–F ₄	2O–F ₂	2S–F ₄	2S–F ₂
Experimental data				
$\Delta E_{\text{REDOX}} (= E_{1/2}^1(\text{D}) - E_{1/2}^1(\text{A}))$ (V) ^a	-0.04	0.20	0.15	0.39
D–A interplanar distance (Å) ^b	3.36158	3.32938	3.406	3.39887 ^c
δ from bond length analyses in A	0.7981(23)	0.713(43)	0.693(23)	0.462(32) ^d
σ at 300 293 K (S cm ⁻¹)	4.91×10^{-42}	1.41×10^{-2}	0.10	6.93×10^{-3}
$h\nu_{\text{CT}}$ (eV)	0.83	0.73	0.64	0.50
E_a for high temperature region (eV) ^d	0.113427(1)	0.178(1)	0.200(1)	0.112420(1)
	(290257–315293 K)	(259–337 K)	(288–340 K)	(288257–312303 K)
E_a for low temperature region (eV) ^d	0.215496(24)	0.2256(4)	0.277378(34)	0.090247(178)
	(238218–258244 K)	(219–231 K)	(228–273 K)	(221200–244230 K)
Calculated data				
E_g (eV) ^e	0.054	< 0.01 ^f	< 0.01	0.02 ^c
W (eV) ^e	0.864	0.898 ^c	0.887	0.8991 ^c
t_{DA} (eV) ^f	0.2035	0.2097 ^c	0.208	0.2068 ^c
$U_{\text{eff}}(\text{D})$ (eV) ^f	2.31	2.2830 ^c	1.8990	1.898 ^c
$U_{\text{eff}}(\text{A})$ (eV) ^f	2.523	2.4851 ^c	2.1922	2.223 ^c

Fig. 6 | Temperature-dependent electrical conductivities of mixed-stack complexes. ρ – T plots obtained by a four-probe method for cooling (circles) and heating (plus marks) processes are shown.

The following are some minor points that authors should also address:

Page 2 Abstract, first line : “Mixed-stack complexes which comprise alternating layers of donors and acceptors are ...” . Here layers should be omitted and instead it is suggested “Mixed-stack complexes which comprise columns of alternating donors and acceptor molecules are ...”

We appreciate this reviewer's kind comments. As per this suggestion, we corrected the text as follows.

(Abstract in text)

Mixed-stack complexes which comprise columns of alternating layers of donors and acceptors are organic conductors ...

Page 3, line 14, where it reads “... , although they are more frequently constructed, possibly owing to the Madelung energy gain between charged donor and acceptor in the intermediates.”. Here the word “intermediate” has no clear meaning.

We sincerely appreciate this reviewer's appropriate comments. We deleted the description "in the intermediates" as follows.

(p. 4 in text)

..., although they are more frequently constructed, possibly owing to the Madelung energy gain between charged donor and acceptor ~~in the intermediates~~.

Page 5, line 5, where it reads “ ...fulfills the two requirements for electronic structures” Here the requirements are not clear. Please specify.

We appreciate this reviewer's kind comments. We specified the requirements as follows.

(p. 5 in text)

Thus far, a few mixed-stack complexes have partly fulfilled these two requirements: 1) similar energy levels between the donor HOMO and the acceptor LUMO, with the appropriate energy gaps of approximately 0.2 eV, and 2) a consistent orbital symmetry between them; ...

Page 7, lines 13-14, please define angles theta 1 and theta 2.

We thank this reviewer's kind comments. We defined the θ_1 and θ_2 in the Supplementary Fig. 11 and deleted the comments in the text as follows.

(p. 8 in text)

The **2O** and **2S** donors in single-crystal complexes are nearly planar, with $|\theta_1| \approx 180^\circ$ and $|\theta_2| \approx 176^\circ$, respectively (Supplementary Fig. 11), ...

Supplementary Fig. 11: Single-crystal structures of donors in mixed-stack complexes at 300 K.

Page 16, line 4, when it reads "... helped us to address the underlying mechanism" Here which mechanism is not clear. Please specify.

We appreciate this reviewer's important suggestions. The ESR measurements clarified the electronic structures of the single-crystal mixed-stack complexes, including the characteristics of the spins for the π -electrons and the temperature dependence. We specified these matters as follows.

(p. 18 in text)

The magnetic characteristics revealed via temperature-dependent electron spin resonance (ESR) measurements helped us to address the insights into the electronic structures of the

mixed-stack complexes (Supplementary Note 10) ~~underlying mechanism~~.

[Additional corrections]

The above-mentioned updates in the Figures, Tables, Supplementary Figures, Supplementary Tables, and Supplementary Notes required us to change the numbers for the following items. Additionally, we have made some corrections to the text for a typo and missing information as follows.

(p. 6 in text)

Surprisingly, the ~~similar~~ donor HOMO and acceptor LUMO with comparable energy levels

(p. 8 in text)

The salts had an astonishingly high solubility in organic solvents such as ~~acetonitrile dichloromethane and chloroform~~ ...

(p. 12 in text)

The spectra obtained for the π -stacking direction of the single crystals have peak energies based on the charge-transfer band ($h\nu_{CT}$) at 0.50 eV (**2S-F₂**) < 0.64 eV (**2S-F₄**) < 0.73 eV (**2O-F₂**) < 0.83 eV (**2O-F₄**) (Table 1 and Fig. 5a; ~~complexes~~ are located exactly at the N-I boundary, whereas **2S-F₂** is nearly neutral).

(p. 19 in text)

..., suggesting a spin-Peierls-like singlet ~~triplet transition formation~~ based on the 1D electronic structure ...

(p. 23–24 in text)

Synthesis of 5,5'-bis(methylthio)-2,2'-bi(3,4-ethylenedithiophene) 2S

~~General synthetic procedure and materials sources are shown in Supplementary Notes 1 and 2. ... a solution of N-bromosuccinimide (NBS; 238 mg, 1.34 mmol) ...~~

(p. 31–32 in text)

~~The crystallographic data (CIF files) for the structures reported in this Article have been deposited with the Cambridge Crystallographic Data Centre (CCDC), under deposition~~

numbers 2264341 (**2S**), 2264342 (**2S•BF₄**), 2264325 (**2O–F₄**), 2264326 (**2O–F₂**), 2264327 (**2S–F₄**), and 2264331 (**2S–F₂**). Crystallographic data for **2S**, **2S•BF₄**, **2O–F₄**, **2O–F₂**, **2S–F₄**, and **2S–F₂** (CIF).

(p. 50 in text; Caption for Fig. 7d,e)

d, Normalized χ_{spin} determined from the calculation using the intensity and ΔB_{pp} of ESR signals at $g \approx 2.003$. The values are normalized by the value at 290 K, with error bars (s.d.). See Supplementary Information for details. **e**, ΔB_{pp} of the ESR signals at $g \approx 2.003$, with error bars (s.d.).

REVIEWERS' COMMENTS

Reviewer #2 (Remarks to the Author):

Two main scientific questions were raised by the Referees concerning (i) the presence/absence of inversion centers in the structures related to stack dimerization and (ii) the assignments of IR spectra. The first point has been very seriously addressed with new structural data collected on synchrotron radiation on all complexes, , confirming for the latter the n-dimerization fluctuation above the phase transition, in accordance with the observation of the electron–molecular vibration (EMV) coupling-based signals in the polarized infrared reflectivity spectra. Based on these new structures, the degree of charge transfer was revised, as well as all band structures calculations. Calculations of IR spectra were also fully revised, using a dimer model as reported in the literature (Ref 49 in the text, ref 28 in ESI). Besides new electrical conductivity data obtained by 4-probe techniques for all complexes were also provided. SI were accordingly extensively modified. All these efforts were possible through the involvement of new contributors (7 people), the role of each of them has been clearly established to justify their addition as co-author.

Besides, concerning now the writing of the paper, as requested by the referees, all obscure or exaggerated sentences have been removed or corrected. For example the term “highly hybridized” replaces the term “topologically fused”. Questions on the pitch of nodal planes of frontier orbitals raised by two referees were also carefully answered. Altogether the paper is much easier to read. This revised version is therefore based now on much more solid experimental data and discussion, allowing for publication in Nature Comm.

Note: since crystal structure data were modified with new data collections at synchrotron, please check if the cif files deposited at CSD were replaced by the new ones.

Reviewer #3 (Remarks to the Author):

The authors substantially improved the quality of the manuscript by addressing as much as possible the reviewers' requests. I recommend acceptance of the manuscript.

Reviewer #4 (Remarks to the Author):

I consider that in this revised version the authors have addressed positively previous points raised by the reviewers and I have no objection in advising publication of this version of the manuscript.

RESPONSE TO REVIEWERS' COMMENTS

Our reply to the Comments from Reviewer #2

We sincerely appreciate the favorable comments and important suggestions. The comments are shown in blue, and our responses are shown in black.

Two main scientific questions were raised by the Referees concerning (i) the presence/absence of inversion centers in the structures related to stack dimerization and (ii) the assignments of IR spectra. The first point has been very seriously addressed with new structural data collected on synchrotron radiation on all complexes, confirming for the latter the π -dimerization fluctuation above the phase transition, in accordance with the observation of the electron–molecular vibration (EMV) coupling-based signals in the polarized infrared reflectivity spectra. Based on these new structures, the degree of charge transfer was revised, as well as all band structures calculations. Calculations of IR spectra were also fully revised, using a dimer model as reported in the literature (Ref 49 in the text, ref 28 in ESI). Besides new electrical conductivity data obtained by 4-probe techniques for all complexes were also provided. SI were accordingly extensively modified. All these efforts were possible through the involvement of new contributors (7 people), the role of each of them has been clearly established to justify their addition as co-author.

Besides, concerning now the writing of the paper, as requested by the referees, all obscure or exaggerated sentences have been removed or corrected. For example the term “highly hybridized” replaces the term “topologically fused”. Questions on the pitch of nodal planes of frontier orbitals raised by two referees

were also carefully answered. Altogether the paper is much easier to read.

This revised version is therefore based now on much more solid experimental data and discussion, allowing for publication in Nature Comm.

Note: since crystal structure data were modified with new data collections at synchrotron, please check if the cif files deposited at CSD were replaced by the new ones.

We would like to express our gratitude for the professional comments provided by the reviewer. We also thank the kind suggestion. We confirmed that the deposited cif files were correctly replaced.

Our reply to the Comments from Reviewer #3

We sincerely appreciate the favorable comments and important suggestions. The comments are shown in blue, and our responses are shown in black.

The authors substantially improved the quality of the manuscript by addressing as much as possible the reviewers' requests. I recommend acceptance of the manuscript.

We would like to express our gratitude for the reviewer's kind comments. We also appreciate the reviewer's previous suggestions and have taken them into consideration.

Our reply to the Comments from Reviewer #4

We sincerely appreciate the favorable comments and important suggestions.

I consider that in this revised version the authors have addressed positively previous points raised by the reviewers and I have no objection in advising publication of this version of the manuscript.

We appreciate the valuable feedback provided by the reviewer and have taken their previous suggestions into account.